# On the normative advantages of dopamine and striatal opponency for learning and choice

Alana Jaskir*, Michael J Frank*

Department of Cognitive, Linguistic and Psychological Sciences, Carney Institute for Brain Science, Brown University, Providence, United States

**Abstract** The basal ganglia (BG) contribute to reinforcement learning (RL) and decision-making, but unlike artificial RL agents, it relies on complex circuitry and dynamic dopamine modulation of opponent striatal pathways to do so. We develop the OpAL* model to assess the normative advantages of this circuitry. In OpAL*, learning induces opponent pathways to differentially emphasize the history of positive or negative outcomes for each action. Dynamic DA modulation then amplifies the pathway most tuned for the task environment. This efficient coding mechanism avoids a vexing explore–exploit tradeoff that plagues traditional RL models in sparse reward environments. OpAL* exhibits robust advantages over alternative models, particularly in environments with sparse reward and large action spaces. These advantages depend on opponent and nonlinear Hebbian plasticity mechanisms previously thought to be pathological. Finally, OpAL* captures risky choice patterns arising from DA and environmental manipulations across species, suggesting that they result from a normative biological mechanism.

## Editor's evaluation

This paper provides a formal analysis of the normative advantage of the opponent pathways of the basal ganglia circuit for cost-benefit decision-making. Specifically, a previously introduced Hebbian nonlinearity is combined with reward-based DA modulation to optimize exploration across lean and rich environments, and across a range of pharmacological and contextual manipulations. The scope of the model, its biological plausibility, and its normative and descriptive aspects are likely to have a significant impact.

*For correspondence:
alana_jaskir@brown.edu (AJ);
Michael_Frank@brown.edu (MJF)

## Introduction

Everyday choices involve integrating and comparing the subjective values of alternative actions. Moreover, the degree to which one prioritizes the benefits or costs in forming subjective preferences may vary between and even within individuals. For example, one may typically use food preference to guide their choice of restaurant, but be more likely to minimize costs (e.g., speed, distance, price) when only low-quality options are available (only fast-food restaurants are open). In this article, we evaluate the computational advantages of such context-dependent choice strategies and how they may arise from biological properties within the basal ganglia (BG) and dopamine (DA) system.

In ecological settings, there are often multiple available actions, and rewards are sparse. In machine learning, this combination is particularly vexing for reinforcement learning (RL) agents due to a difficult exploration/exploitation tradeoff (*Sutton and Barto, 2018*), and approaches to confront this problem typically require prior task-specific knowledge (*Riedmiller et al., 2018*). We set out to study how the architecture of biological RL might additionally circumvent this problem. We find that biological

properties within this system – specifically, the presence of opponent striatal pathways, nonlinear Hebbian plasticity, and dynamic changes in dopamine as a function of reward history – confer decision-making advantages relative to canonical RL models lacking these properties. In so doing, this analysis provides a new lens into various findings regarding how learning and decision-making is altered across species as a function of manipulations of (or individual differences within) the BG and DA systems.

To begin, we focus on bandit learning tasks, where an agent learns to identify and reliably select the option which yields the highest rate of probabilistic reward. We consider how biological properties with the BG allow an agent to effectively explore early (sample options that are currently estimated as unfavorable but are possibly more rewarding), and subsequently better exploit (reliably select the most rewarding action). As we shall see, this entails (1) learning separate 'actors' that magnify the relative benefits of alternative options in highly rewarding environments or the relative costs in sparsely rewarding environments and (2) dynamically shifting the contribution of these actors to govern action selection, depending on which is more specialized for the context. We then show how this dynamic biological mechanism can be recruited for risky decision-making, where increased dopamine amplifies the contribution of benefits over costs, leading to riskier choice; lowered dopamine alternatively amplifies the costs over the benefits.

In neural network models of such circuitry, the cortex 'proposes' candidate actions available for consideration, and the BG facilitates those that are most likely to maximize reward and minimize cost (*Frank, 2005*; *Ratcliff and Frank, 2012*; *Franklin and Frank, 2015*; *Gurney et al., 2015*; *Dunovan and Verstynen, 2016*). These models are based on the BG architecture in which striatal medium spiny neurons (MSNs) are subdivided into two major populations that respond in opponent ways to DA (due to differential expression of D1 and D2 receptors; *Gerfen, 1992*; *Burke et al., 2017*). Phasic DA signals convey reward prediction errors (*Montague et al., 1996*; *Schultz et al., 1997*), amplifying both activity and synaptic learning in D1 neurons, thereby promoting action selection based on reward. Conversely, when DA levels drop, activity is amplified in D2 neurons, promoting learning and choice that minimizes disappointment (*Frank, 2005*; *Iino et al., 2020*). See *Figure 1A* for a visual summary of this opponency.

Empirically, the BG and DA have been strongly implicated in such motivated action selection and RL across species. For example, in perceptual decisions, striatal D1 and D2 neurons combine information about veridical perceptual data with internal preferences based on potential reward, causally influencing choice toward the more rewarding options (*Doi et al., 2020*; *Bolkan et al., 2022*). Further, striatal DA manipulations influence RL (*Yttri and Dudman, 2016*; *Frank et al., 2004*; *Pessiglione et al., 2006*), motivational vigor (*Niv et al., 2007*; *Beeler et al., 2012*; *Hamid et al., 2016*), cost–benefit decisions about physical effort (*Salamone et al., 2018*), and risky decision-making. Indeed, as striatal DA levels rise, humans and animals are more likely to select riskier options that offer greater potential payout than those with certain but smaller rewards (*St Onge and Floresco, 2009*; *Zalocusky et al., 2016*; *Rutledge et al., 2015*), an effect that has been causally linked to striatal D2 receptor-containing subpopulations (*Zalocusky et al., 2016*).

However, for the large part, this literature has focused on the findings *that* DA has opponent effects on D1 and D2 populations and behavioral patterns, and not what the computational advantage of this scheme might be (i.e., *why*). For example, the Opponent Actor Learning (OpAL) model (*Collins and Frank, 2014*) summarizes the core functionality of the BG neural network models in algorithmic form, capturing a wide variety of findings of DA and D1 vs. D2 manipulations across species (for review, *Collins and Frank, 2014*; *Maia and Frank, 2017*). Two distinguishing features of OpAL (and its neural network inspiration), compared to more traditional RL models, are that (1) it relies on opponent D1/D2 actors that separately learn benefits and costs of actions rather than a single expected reward value for each action and (2) learning in such populations is acquired through nonlinear dynamics, mimicking three-factor Hebbian plasticity rules. This nonlinearity causes the two populations to evolve to specialize in discriminating between options of high or low reward value, respectively (*Collins and Frank, 2014*), as seen in *Figure 1B*. It is also needed to explain pathological conditions such as learned Parkinsonism, whereby low DA states induce hyperexcitability in D2 MSNs, driving aberrant plasticity and, in turn, progression of symptoms (*Wiecki et al., 2009*; *Beeler et al., 2012*).

But why would the brain develop this nonlinear opponent mechanism for action selection and learning, and how could (healthy) DA levels be adapted to capitalize on it? A clue to this question lies in the observation that standard (nonbiological) RL models typically perform worse at selecting

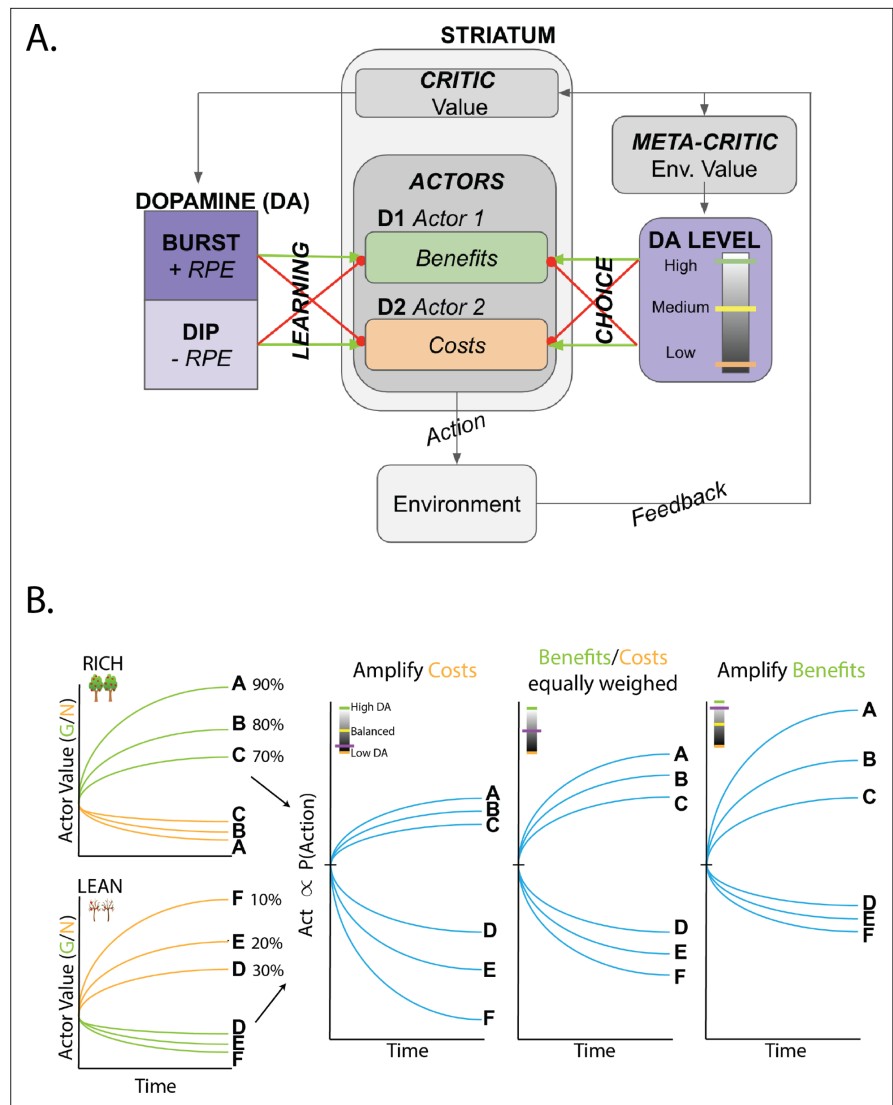

**Figure 1.** Overview of OpAL* and dynamics of three-factor Hebbian term. (**A**) OpAL* architecture. Akin to the original OpAL model (**Collins and Frank, 2014**), OpAL* is a modified dual actor-critic model where the critic learns action values and generates reward prediction errors (RPEs); the actors use these RPEs to directly learn a policy (i.e., how to behave). For each action, the representation according to one actor (representing the D1 pathway) is strengthened by positive RPEs and weakened by negative RPEs (encoded by dopamine burst and dips, respectively). In contrast, positive RPEs weaken and negative RPEs strengthen the second actor's action representations (representing the D2 pathway). Uniquely, OpAL* modulates dopamine levels at the time of choice according to a 'meta-critic,' which tracks the value or 'richness' of the overall environment according to the agent's reward history agnostic to action history. OpAL* also introduces additional features, such as annealing and normalization, that provide OpAL* with robustness and flexibility but preserve key properties of the OpAL model necessary for capturing empirical data. (**B**) Schematic of OpAL dynamics with three-factor Hebbian term. Nonlinear weight updates due to Hebbian factor lead to increasing discrimination between high reward probability options in the $G$ actor and between low reward probability options in the $N$ actor. For intermediate dopamine states ($G$ and $N$ actors are balanced), there is equal sensitivity to differences in reward probability across the range of rich and lean environments. For high dopamine states ($\beta_g > \beta_n$), the action policy emphasizes differences in benefits (as represented in the D1/"G" weights), whereas in low dopamine states ($\beta_g < \beta_n$), the action policy emphasizes differences in costs (as represented in the D2/"N" weights). Changes in dopaminergic state (represented by the purple indicators) affect the policy of OpAL due to its nonlinear and opponent dynamics. OpAL* hypothesizes that modulating dopaminergic state by environmental richness is a normative mechanism for flexible weighting of these representations.

The online version of this article includes the following figure supplement(s) for figure 1:

**Figure supplement 1.** Dynamics of G/N/Act without three-factor Hebbian dynamics.

the optimal action in 'lean environments' with sparse rewards than they do in 'rich environments' with plentiful rewards (*Collins and Frank, 2014*). This asymmetry results from a difference in exploration/exploitation tradeoffs across such environments. In rich environments, an agent can benefit from overall higher levels of exploitation: once the optimal action is discovered, an agent can stop sampling alternative actions as it is not important to know their precise values. In contrast, in lean environments, choosing the optimal action typically lowers its value (due to sparse rewards), to the point that it can drop below those of even more suboptimal actions. This causes stochastic switching between options until the worst actions are reliably identified and avoided in the long run. Moreover, while in machine learning applications one can simply tune hyperparameters of an RL model to optimize performance for a given environment, biological agents do not have the luxury as they cannot know whether they are in a rich or lean environment in advance and cannot modify hyperparameters accordingly.

In this article, we investigate the utility of nonlinear BG opponency for adaptive behavior in rich and lean environments. We propose a new model, OpAL*, which dynamically adapts its dopaminergic state online as a function of learned reward history (as observed empirically; *Hamid et al., 2016*; *Mohebi et al., 2019*). Specifically, OpAL* dynamically modulates its dopaminergic states in proportion to its estimates of 'environmental richness,' leading to high striatal DA motivational states in rich environments and lower DA states in lean environments with sparse rewards. To do so, it relies on a 'meta-critic' that evaluates the richness/sparseness of the environment as a whole. Initially, low confidence in the meta-critic leads the agent to rely equally on both actors, with more stochastic choice as they learn to specialize. Thereafter, OpAL*'s opponent and nonlinear representations serve to directly and quickly optimize the model's policy. In contrast, standard RL models that focus on learning the expected values of actions are slow to converge on the best policy, particularly as the number of alternative actions grows. In this article, we demonstrate that the specialization of D1 and D2 pathways in OpAL* for discriminating between low rewarding and high rewarding options, rather than estimating veridical reward statistics, allows OpAL* to better equate performance in rich and lean environments. This dynamic modulation amplifies the D1 or D2 actor most well suited to discriminate amongst benefits or costs of choice options for the given environment, akin to an 'efficient coding' strategy typically studied in the domain of perception (*Barlow, 2012*; *Laughlin, 1981*; *Chalk et al., 2018*). We compared the performance of OpAL* to alternative BG models and to several alternative models typically used in machine learning (Q-learning and upper confidence bound models, the latter of which includes an explicit mechanism intended to optimize exploration). We find that OpAL*, across a wide range of parameter settings, exhibits robust advantages over these alternatives across a range of environments with varying reward rates and complexity levels. This advantage depends on opponency, nonlinearity, and adaptive DA modulation and is most prominent in lean environments with large action spaces, an ecologically probable environment which requires more adaptive navigation of explore–exploit as outlined above. OpAL* also addresses limitations of the original OpAL model highlighted by *Möller and Bogacz, 2019*, while retaining key properties needed to capture a range of empirical data and afford the normative advantages.

Finally, we apply OpAL* to capture a range of empirical data across species, including how risk preference changes as a function of D2 MSN activity and manipulations that are not explainable by monolithic RL systems even when made sensitive to risk (*Zalocusky et al., 2016*). In humans, we show that OpAL* can reproduce patterns in which dopaminergic drug administration selectively increases risky choices for gambles with potential gains (*Rutledge et al., 2015*). Moreover, we show that even in the absence of biological manipulations, OpAL* also accounts for recently described economic choice patterns as a function of environmental richness. In particular, we simulate data showing that when offered the very same safe and risky choice option, humans are more likely to gamble when that offer had been presented in the context of a richer reward distribution (*Frydman and Jin, 2021*). Similarly, we show that the normative objective for policy optimization in OpAL*, while in general facilitating adaptive behavior and transitive preferences, can lead to irrational preferences when options appear in novel contexts differing in reward richness of initial learning, as observed empirically (*Palminteri et al., 2015*). Taken together, our simulations provide a clue as to the normative function of the biology of RL which differs from that assumed by standard models and gives rise to variations in risky decision-making.

## OpAL overview

Before introducing OpAL*, we first provide an overview of the original OpAL model (**Collins and Frank, 2014**), an algorithmic model of the BG whose dynamics mimic the differential effects of dopamine in the D1/D2 pathways described above. OpAL is a modified 'actor-critic' architecture (**Sutton and Barto, 2018**). In the standard actor-critic, the critic learns the expected value of an action from rewards and punishments and reinforces the actor to select those actions that maximize rewards. Specifically, after selecting an action ($a$), the agent experiences a reward prediction error ($\delta$) signaling the difference between the reward received ($R$) and the critic's learned expected value of the action ($V_t(a)$) at time $t$:

$$\delta_t = R_t - V_t(a) \tag{1}$$
$$V_{t+1}(a) = V_t(a) + \alpha_c \times \delta_t, \tag{2}$$

where $\alpha_c$ is the critic learning rate. The prediction error generated by the critic is then also used to train the actors. OpAL is distinguished from a standard actor-critic in two critical ways, motivated by the biology summarized above. First, it has two separate opponent actors: one promoting selection ('Go') of an action $a$ in proportion to its relative benefit over alternatives, and the other suppressing selection of that action ('No Go') in proportion to its relative cost (or disappointment). (See Supplemental note 1 in Appendix 2). Second, the update rule in each of these actors contains a *three-factor Hebbian rule* such that weight updating is proportional not only to learning rates and RPEs (as in standard RL) but is also scaled by $G_t$ and $N_t$ themselves. In particular, positive RPEs conveyed by phasic DA bursts strengthen the $G$ (D1) actor and weaken the $N$ (D2) actor, whereas negative RPEs weaken the D1 actor and strengthen the D2 actor.

$$G_{t+1}(a) = G_t(a) + \alpha_G G_t(a) \times \delta_t \tag{3}$$
$$N_{t+1}(a) = N_t(a) + \alpha_N N_t(a) \times -\delta_t \tag{4}$$

where $\alpha_G$ and $\alpha_N$ are learning rates controlling the degree to which D1 and D2 neurons adjust their synaptic weights with each RPE. We will refer to these $G_t$ and $N_t$ terms that multiply the RPE in the update as the 'Hebbian term' because weight changes grow with activity in the corresponding $G$ and $N$ units. As such, the $G$ weights grow to represent the benefits of candidate actions (those that yield positive RPEs more often, thereby making them yet more eligible for learning), whereas the $N$ weights grow to represent the costs or likelihood of disappointment (those that yield negative RPEs more often).

The resulting nonlinear dynamics capture biological plasticity rules in neural networks, where learning depends on dopamine ($\delta_t$), presynaptic activation in the cortex (the proposed action $a$ is selectively updated), and postsynaptic activation in the striatum ($G_t$ or $N_t$) (**Frank, 2005**; **Wiecki et al., 2009**; **Beeler et al., 2012**; **Gurney et al., 2015**; **Frémaux and Gerstner, 2015**; **Reynolds and Wickens, 2002**). Incorporation of this Hebbian term prevents redundancy in the D1 vs. D2 actors and confers additional flexibility, as described in the next section. It is also necessary for capturing a variety of behavioral data, including those associated with pathological aberrant learning in DA-elevated and depleted states, whereby heightened striatal activity in either pathway amplifies learning that escalates over experience (**Wiecki et al., 2009**; **Beeler et al., 2012**; **Collins and Frank, 2014**). As we shall see in the 'Mechanism' section below, this same property allows actors to better represent the probabilistic history of outcomes at the low and high ranges.

For action selection (decision-making), OpAL combines together $G_t(a)$ and $N_t(a)$ into a single action value, $Act(a)$, but where the contributions of each opponent actor are weighted by corresponding gains $\beta_g$ and $\beta_n$.

$$Act_t(a) = \beta_g G_t(a) - \beta_n N_t(a) \tag{5}$$
$$\beta_g = \beta(1 + \rho) \tag{6}$$
$$\beta_n = \beta(1 - \rho) \tag{7}$$

Here, $\rho$ reflects the *dopaminergic state* controlling the relative weighting of $\beta_g$ and $\beta_n$, and $\beta$ is the overall softmax temperature. Higher $\beta$ values correspond to higher exploitation, while $\beta = 0$ would

generate random choice independent of learned values. When $\rho = 0$, the dopaminergic state is 'balanced' and the two actors $G$ and $N$ (and hence, learned benefits and costs) are equally weighted during choice. If $\rho > 0$, benefits are weighted more than costs, and vice versa if $\rho < 0$. While the original OpAL model assumed a fixed, static $\rho$ per simulated agent to capture individual differences or pharmacological manipulations, below we augmented it to include the contributions of dynamic changes in dopaminergic state, so that $\rho$ can evolve over the course of learning to optimize choice.

The actor then selects actions based on their relative action propensities, using a softmax decision rule, such that the agent selects those actions that yield the most frequent positive RPEs:

$$p(a) = \frac{e^{Act_t(a)}}{\sum_{i \in A} e^{Act_t(i)}}, \tag{8}$$

## Nonlinear OpAL dynamics support amplification of action-value differences

After learning, $G$ and $N$ weights correlate positively and negatively with expected reward, with appropriate ordinal rankings of each action preserved in the combined action value $Act$ (**Collins and Frank, 2014**). However, with extensive learning (particularly after the critic converges), the Hebbian term induces instability and decay in the G and N representations, such that they eventually converge to zero (**Möller and Bogacz, 2019**). OpAL* addresses this issue by adjusting learning rates as a function of uncertainty, stabilizing learned actor weights while also preserving their ability to flexibly adapt to change points, and by normalizing the prediction error. See 'Normalization and annealing' in the next section for a full discussion. These adjustments enable us to preserve the Hebbian contribution, which was previously found to be a necessary component for capturing a range of empirical data (**Collins and Frank, 2014**). Importantly for these findings and for the findings in this article, the Hebbian term produces nonlinear dynamics in the two actors such that they are not redundant and instead specialize in discriminating between different reward probability ranges (**Figure 1B**). While the $G$ actor shows greater discrimination among frequently rewarded actions, the $N$ actor learns greater sensitivity among actions with sparse reward. Note that if $G$ and $N$ actors are weighted equally in the choice function ($\rho = 0$), the resultant choice preference is invariant to translations across levels of reward, exhibiting identical discrimination between a 90 and 80% option as it would between a 80% and 70% option. This 'balanced' OpAL model therefore effectively reduces to a standard nonopponent actor-critic RL model, but as such fails to capitalize on the underlying specialization of the actors ($G$ and $N$) in ongoing learning. We considered the possibility that such specialization could be leveraged dynamically to amplify a given actor's contribution when it is most sensitive, akin to an 'efficient coding' strategy applied to decision-making (**Frydman and Jin, 2021**).

## OpAL*

Given the differential specialization of $G$ vs. $N$ actors, we considered whether the agent's online estimation of environmental richness (reward rate) could be used to control dopaminergic states (as seen empirically; **Hamid et al., 2016**; **Mohebi et al., 2019**). Due to its opponent effects on D1 vs. D2 populations, such a mechanism would differentially and adaptively weight $G$ vs. $N$ actor contributions to the choice policy. To formalize this hypothesis, we constructed OpAL*, which uses an online estimation of environment richness to dynamically amplify the contribution of the actor theoretically best specialized for the environment type.

To provide a robust estimate of reward probability in a given environment, OpAL* uses a 'meta-critic,' so-named because it evaluates the reward value of the environment as a whole given the agent's overall choice history (i.e., policy to that point), rather than that of any particular state or action. The meta-critic summarizes the contributions of various inputs that may regulate the DA system, including those from cortical sources such as orbitofrontal cortex and anterior cingulate, regions which have access to not only the mean reward values but also their confidence (**Kepecs et al., 2008**). Notably, these regions also project to striatal cholinergic cells conveying information about environmental state (**Stalnaker et al., 2016**). These cholinergic cells in turn locally regulate striatal DA release (**Adrover et al., 2020**; **Threlfell et al., 2012**; **Reynolds et al., 2022**) in proportion to reward history (**Mohebi et al., 2019**), and may be sensitive to uncertainty (**Franklin and Frank, 2015**). As such, the meta-critic is represented as a beta distribution to estimate $\hat{p}_t(r)$ for the environment as a whole (i.e., over all states and actions) or 'context value.' This distribution can be updated by keeping a running count of

the outcomes (e.g., rewards and omissions) on each trial and adding them to the hyperparameters $\eta$ and $\gamma$, respectively.

$$\eta_{t+1}^c = \eta_t^c + R_t \tag{9}$$

$$\gamma_{t+1}^c = \gamma_t^c + (1 - R_t) \tag{10}$$

$$X \sim \text{Beta}(\eta_t^c, \gamma_t^c) \tag{11}$$

$$\hat{p}_t(r) = E[X] \tag{12}$$

The dopaminergic state $\rho$ is then increased when $\hat{p}_t(r) > .5$ (*rich environment*), and decreased when $\hat{p}_t(r) < .5$ (*lean environment*). To ensure that dopaminergic states accurately reflect environmental richness, we apply a conservative rule to modulate $\rho$ only when the meta-critic is sufficiently 'confident' that the reward rates are above or below 0.5, that is, we take into account not only the mean but also the variance of the beta distribution, parameterized by $\phi$ (**Equation 13**; for simplicity, we used $\phi = 1.0$ for all simulations). This process is akin to performing inference over the most likely environmental state to guide DA. (See Supplemental note 2 in Appendix 2). Lastly, a constant $k$ controls the strength of the modulation (**Equation 14**)

$$S = \begin{cases} 1 & \text{if } E[X] - \phi \, \text{std}(X) > .5 \\ 1 & \text{if } E[X] + \phi \, \text{std}(X) < .5 \\ 0 & \text{otherwise} \end{cases} \tag{13}$$

$$\rho_t = S \times (E[X] - .5) \times k, \qquad k \geq 0 \tag{14}$$

To illustrate the necessity of nonlinearity for dopamine modulation to be impactful, we plotted how *Act* values change as a function of reward probability and for different DA levels (represented as different colors, **Figure 2**). While *Act* values increase monotonically with reward probability, the convexity in the underlying $G$ and $N$ weights (**Figure 1B**) gives rise to stronger *Act* discrimination between more rewarding options (e.g., 80% vs. 70%) with higher dopamine levels. Conversely, *Act* discrimination between less rewarding options (e.g., 30% vs. 20%) is enhanced with lower dopamine levels. Thus, high DA amplifies the G actor's contributions to choice, increasing the action gap for high-probability options. Conversely, low DA amplifies the N actor's contributions, increasing the action gap for low-probability options. As the Bayesian meta-critic converges on an estimate of environmental richness, OpAL* can adapt its policy to dynamically emphasize the most discriminative actor and appropriately enhance the 'action gap' – *Act* difference between optimal and second best option – to optimize the policy (**Figure 2**, left). In contrast, a variant which lacks nonlinearity (No Hebb) induces redundancy in the $G$ and $N$ weights and thus essentially reduces to a standard actor-critic agent. As such, dopamine modulation does not change its discrimination performance across environments and the action gap for choice remains fixed (**Figure 2**, right; **Figure 1—figure supplement 1**).

## Choice

To accommodate varying levels of $k$ and maintain biological plausibility, the contribution of each actor is lower-bounded by zero – that is, $G$ and $N$ actors can be suppressed but cannot be inverted (firing rates cannot go below zero), while still allowing graded amplification of the other subpopulation.

$$Act_t(a) = \beta_g G_t(a) - \beta_n N_t(a) \tag{15}$$

$$\beta_g = \beta \max(0, 1 + \rho_t) \tag{16}$$

$$\beta_n = \beta \max(0, 1 - \rho_t) \tag{17}$$

## Normalization and annealing

The original three-factor Hebbian rule presented in **Collins and Frank, 2014** approximates the learning dynamics in the neural circuit models needed to capture the associated data and also confers flexibility as described above. However, it is also susceptible to instabilities, as highlighted by **Möller and Bogacz, 2019**. Specifically, because weight updating scales with the $G$ and $N$ values themselves, large reward magnitudes or oscillating prediction errors (due to critic convergence) can cause the weights to decay rapidly toward 0 (see Appendix 1 section 'Addressing'; **Möller and Bogacz, 2019**).

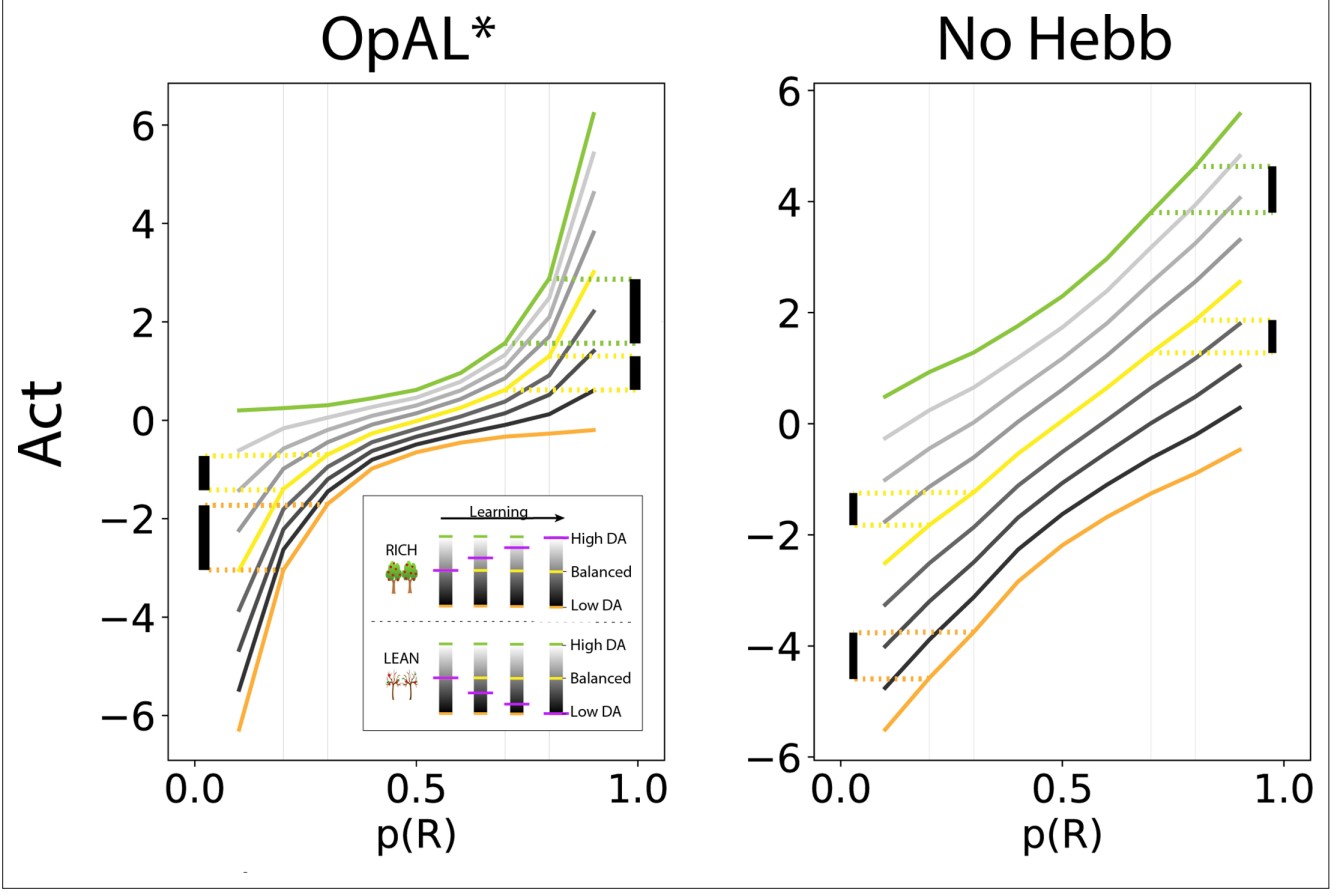

**Figure 2.** OpAL* capitalizes on convexity of actor weights induced by nonlinearity, allowing adaptation to different environments. *Act* values are generated by presenting each model with a bandit using a fixed reward probability for 100 trials; curves are averaged over 5000 simulations. Left: nonlinearity in OpAL* update rule induces convexity in *Act* values as a function of reward probability (due to stronger contributions of *G* weights with higher rewards, and stronger contributions of *N* weights with sparse reward). OpAL* dynamically adjusts its dopaminergic state over the course of learning as a function of its estimate of environmental richness (indicated by elongated, purple bars), allowing it to traverse different Act curves (high dopamine [DA] in green emphasizes the *G* actor, low DA in orange emphasizes the *N* actor). (Note that the agent's meta-critic first needs to be confident in its estimate of reward richness of the environment during in initial exploration for it to adjust DA to appropriately exploit convexity.) Thereafter, OpAL* can differentially leverage convexity in *G* or *N* weights, outperforming a 'balanced' OpAL+ model (in yellow), which equally weighs the two actors (due to static DA). Vertical bars show discrimination (i.e., action gap) between 80% and 70% actions is enhanced with high DA state, whereas discrimination between 20% and 30% actions is amplified for low DA. Right: due to redundancy in the No Hebb representations, policies are largely invariant to dopaminergic modulation during the course of learning.

To address this issue, OpAL* introduces two additional modifications based on both functional and biological considerations. First, we apply a transformation to the actor prediction errors such that they are normalized by the range of available reward values (see *Tobler et al., 2005* for evidence of such normalization in dopaminergic signals and *Bavard et al., 2018* for evidence that humans use such range adaptation). Secondly, as is common in machine learning (*Darken and Moody, 1990*; *Bengio, 2012*), the actor learning rate is annealed with experience. Rather than simply decreasing the learning rate with time (which would make the agent insensitive to subsequent volatility in task conditions), we instead anneal actor learning rates as a function of uncertainty within the OpAL* Bayesian meta-critic. (See Supplemental note 3 in Appendix 2). As the variance decreases, the agent is more certain about the environmental statistics, and the actor learning rate declines to stabilize learning (again see *Franklin and Frank, 2015* for mechanisms of such uncertainty-based actor learning modulation).

$$G_{t+1}(a) \quad = G_t(a) + \alpha_G(t)G_t(a)f(\delta_t) \tag{18}$$

$$N_{t+1}(a) \quad = N_t(a) + \alpha_N(t)N_t(a)f(-\delta_t) \tag{19}$$

$$\alpha_{G,N}(t) \quad = \frac{\alpha_{G,N}}{1+\frac{1}{T*var(X)}} \tag{20}$$

$$f(\delta) \quad = \frac{\delta}{R_{mag} - L_{mag}}, \tag{21}$$

where X refers to the beta distribution of the meta-critic (*Equation 11*) at time $t$, and T is a hyper-parameter scaling the degree of annealing linearly with the variance of the meta-critic (greater T values indicate more certainty needed for annealing). The reward prediction error $\delta$ is produced by a standard critic (*Equations 1 and 2*). The critic learning rate ($\alpha_c$) is constant across trials.

This mechanism preserves the agent's ability to remain flexible to potential switch points in reward contingencies (see *Appendix 1—figure 6*). These modifications improve robustness of OpAL* and ensure that the actor weights are better behaved (avoiding convergence to zero and maintaining ordinal rankings in the resulting Act values for 1000 trials; *Appendix 1—figure 4*), while still preserving the Hebbian mechanism that induces convexity in the weights (which, as shown below, are needed for its normative advantages). Of course, the stability–flexibility tradeoff remains, and the level of annealing could be further optimized dynamically, as shown specifically within OpAL (*Franklin and Frank, 2015*) and more broadly in the literature (*Nassar et al., 2012*; *Iglesias et al., 2013*). This stability–flexibility tradeoff is particularly apparent in drifting reward environments (*Appendix 1—figure 5*), but optimization for this tradeoff is largely orthogonal to the focus of the present work. For further discussion and exploration, please see Appendix 1 ('Addressing'; *Möller and Bogacz, 2019*) and *Franklin and Frank, 2015*.

## Results
### Robust advantages of adaptively modulated dopamine states

The main claim of this article is that endogenous changes in dopamine levels can leverage specialization afforded by opponent pathways under Hebbian plasticity, and accordingly optimize performance when environmental statistics are unknown. In this section, we therefore characterize the robustness of OpAL* advantages across a large range of parameter settings relative to variants omitting DA modulation (OpAL+) or the Hebbian term (No Hebb). We then explore how such advantages scale with complexity in environments with increasing number of choice alternatives.

To specifically assess the benefit of adaptive dopaminergic state modulation, we compared OpAL* to two control models to establish the utility of the adaptive dopamine modulation (which was not a feature of the original OpAL model) and to test its dependence on nonlinear Hebbian updates. More specifically, the OpAL+ model equally weights benefits and costs throughout learning (' $\rho = 0$ '); as such, any OpAL* improvement would indicate an advantage for dynamic dopaminergic modulation. (See Supplemental note 4 in Apprendix 2). The No Hebb model reinstates the dynamic dopaminergic modulation but omits the Hebbian term in the three-factor learning rule (*Equations 16 and 17*). This model therefore serves as a test as to whether any OpAL* improvements depend on the underlying nonlinear actor weights produced by the three-factor Hebbian rule. The No Hebb model also serves to compare OpAL* to more standard actor-critic RL models; removing the Hebbian term renders each actor redundant, effectively yielding a single-actor model (see the section 'OpAL*' for more details). Improvement of OpAL* relative to the No Hebb model would therefore suggest an advantage of OpAL* over standard actor-critic models (we also test OpAL* against a standard Q-learner later in this article). Importantly, models were equated for computational complexity, with modulation hyperparameters ($\phi$ and $k$) of dynamic DA models (OpAL* and No Hebb) held constant, and were compared using the same random seeds to best equate performance (see 'Materials and methods').

Following an initial comparison in the simplest two choice learning situation, we tested whether OpAL* advantages may be further amplified in more complex environments with multiple choice alternatives. We introduced additional complexity into the task by adding varying numbers of alternative suboptimal actions (e.g., an environment with four actions with probability of reward 80, 70, 70, and 70%). Results were similar for average learning curves and average reward curves; we focus on average learning curves as they are a more refined, asymptotically sound measure of normative behavior.

For each parameter setting for each model type, we calculated the average softmax probability of selecting the best option (80% in rich environments or 30% in lean environments) across 1000 simulations for 1000 trials. We then took the area under the curve (AUC) of this averaged learning curve for different time horizons (100, 250, 500, and 1000 trials). To statistically investigate where OpAL*

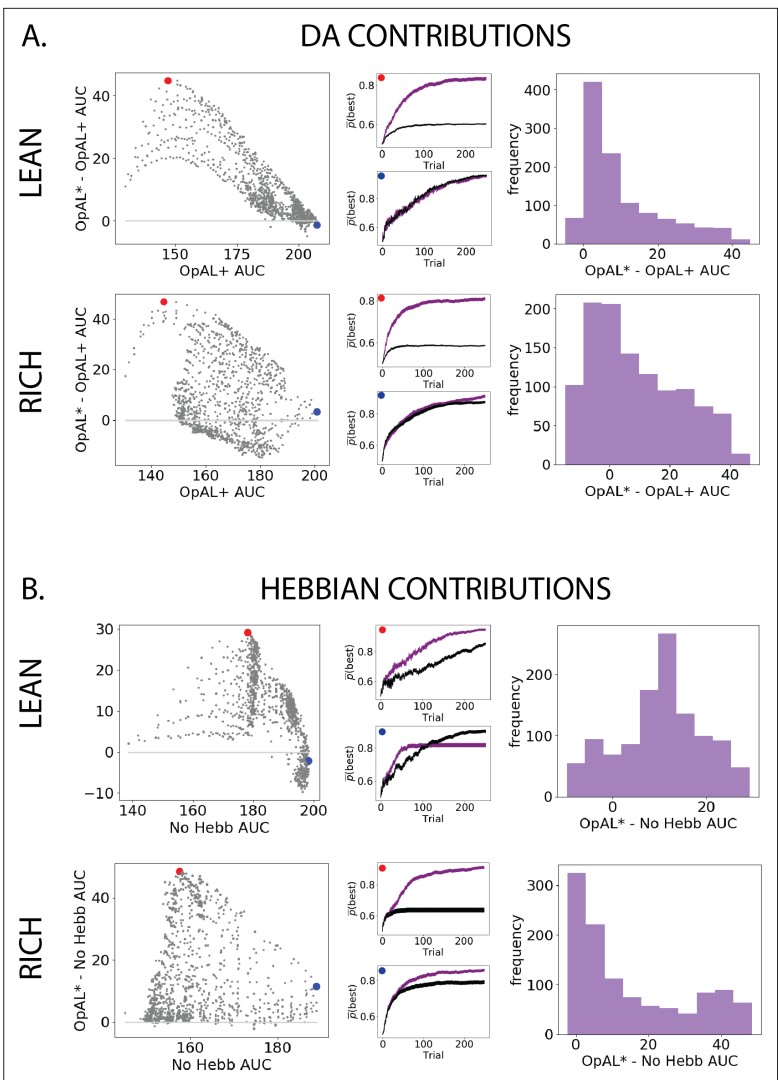

**Figure 3.** Parameter-level comparison of OpAL* to OpAL+ and OpAL* to the No Hebb model across a range of plausible parameters. Results of two-armed bandit environments – rich (80% vs. 70%) or lean (30% vs. 20%) – for 250 trials. Advantages over the OpAL+ model indicate the need for dynamic dopamine (DA) modulation. Advantages over the No Hebb model indicate the need for the nonlinear three-factor Hebbian rule (found in *Equations 18 and 19*). Together, advantages over both control models also indicate the need for opponency, particularly given redundancy in G and N weights in the No Hebb model. See 'Parameter grid search' for more details. (**A**) OpAL* improves upon a control model which lacks dynamic modulation (OpAL+, $\rho = 0$), with largest improvement for moderately performing parameters. Left: each point represents a single-parameter combination and its difference in learning curve areas under the curve (AUCs) in OpAL* compared to the OpAL+ model. Center: average learning curves of the parameter setting which demonstrates the best improvement of OpAL* over the OpAL+ model (indicated by the red dot) and the parameter setting with the best OpAL+ model performance (indicated by the blue dot). Error bars reflect standard error of the mean over 1000 simulations. Right: histogram of the difference in average learning curve AUCs of the two models with equated parameters. (**B**) Dynamic A modulation is insufficient to induce performance advantage without three-factor Hebbian learning (No Hebb). Comparison descriptions analogous to the above.

The online version of this article includes the following figure supplement(s) for figure 3:

**Figure supplement 1.** Dopamine (DA) contribution colored by parameter values, low complexity.

(three-factor Hebbian learning and dopaminergic modulation) was most advantageous, we performed one-sample $t$-tests where the null was zero on the difference between the AUC of OpAL* and each control model for every parameter combination over several time horizons (100, 250, 500, and 1000 trials). OpAL* outperformed its OpAL+ ($\rho = 0$) control and the non-Hebbian version across all time horizons ($p's < 1.0e-13$), except in the lean environment for 1000 trials when compared to the No Hebb model (no significant difference).

We can visualize these statistics plotted according to the AUC of the control model as well as the frequency of the AUC differences (*Figure 3*). Across parameter settings, OpAL* robustly outperforms comparison models in both environment types. Interestingly, OpAL* advantages in the lean environment over the OpAL+ model show an inverted-U relationship, whereby improvements are most prominent for mid-performing parameter combinations. These lean data also contain distinct sweeps and clustering, which relate to the learning rates of both the critic and the actor (see *Figure 3— figure supplement 1* for corresponding figures colored according to parameter values). Notably, the most prominent advantages of OpAL* relative to the No Hebb model occur in the lean environment, where the large majority of parameter combinations show advantages (peaks in AUC differences in the positive range). As explored in detail in the 'Mechanism' section below, the lean environment requires sophisticated explore–exploit tradeoffs that challenge standard RL models; the Hebbian term of OpAL* induces distortions in the N weights to quickly and preferentially avoid the most suboptimal actions. Moreover, dynamic DA modulation provides additional performance advantages (as evidenced by OpAL*'s performance over OpAL+), by exploiting the actor most suited for the environment, but relies on the Hebbian term to do so (as evident by OpAL*'s performance over No Hebb).

Overall, these results show an advantage for dynamic dopaminergic states as formulated in OpAL* when reward statistics of the environment are unknown. Moreover, note that these improvements over balanced ($\rho = 0$) OpAL+ provide a lower bound estimate on the advantages of adaptive modulation, given that using any other fixed $\rho \neq 0$ would perform worse across environments (see *Appendix 1— figure 1*). This advantage is particularly prominent in the lean (sparse reward) environment, which is computationally more challenging and ecologically more realistic than the rich environment, as we will discuss in the 'Mechanism' section. Crucially, dynamic dopaminergic state leverages the full potential

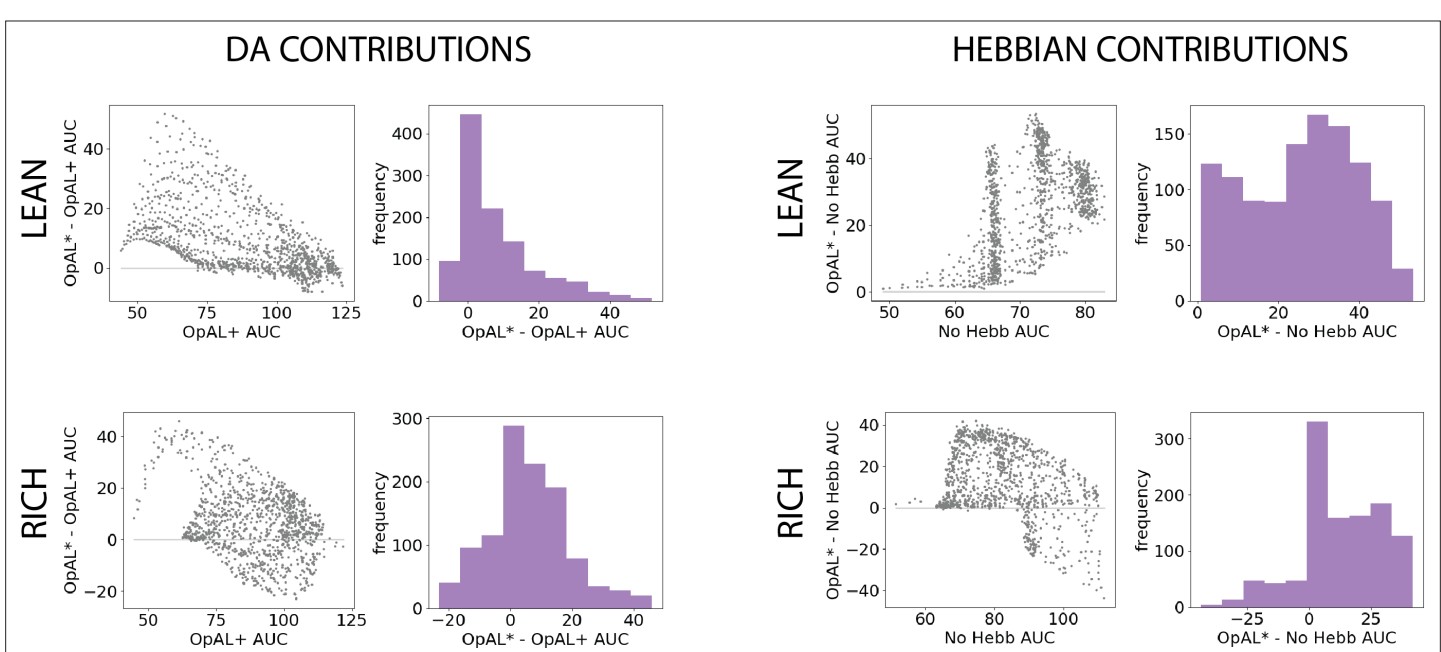

**Figure 4.** Parameter-level comparison of OpAL* to control models in high complexity. OpAL* robustly outperforms control models in high-complexity environments, with lean environments showing the greatest advantage. Models completed a six-armed bandit task (with only one optimal action) for 250 trials. See 'Parameter grid search' for detailed analysis methods.

The online version of this article includes the following figure supplement(s) for figure 4:

**Figure supplement 1.** Dopamine (DA) contribution colored by parameter values, high complexity.

of opponency *only* when combined with three-factor Hebbian learning rules, as demonstrated by OpAL*'s advantage over the No Hebb model.

## OpAL* advantages in sparse reward environments grow with complexity of action space

We next explored the advantages of dopamine modulation and Hebbian plasticity in progressively more complex environments by increasing the number of available choice alternatives, across several time horizons (100, 250, 500, and 1000 trials). Each complexity level introduced an additional suboptimal action to the rich or lean environment. For example, a complexity level of 4 for the lean environment consisted of four options: a higher rewarding option (30% probability of reward) and three equivalent lower rewarding options (20% probability of reward each).

OpAL* outperformed the OpAL+ model (differences in AUCs, $p's < 2.0e^{-}23$) across all time horizons and complexity levels. OpAL* also outperformed the non-Hebbian version ($p's < 1.0e^{-}13$), except for the lowest complexity lean environments after 1000 trials ($p's > 0.4$; OpAL* advantages were still significant for shorter time horizons).

We can again visualize these results as the AUC differences between matched parameters (*Figure 4*). We visualize the highest complexity here for simplicity. As in the two-option results, the benefits of OpAL* are most evident in the lean environment, particularly relative to the No Hebb model. OpAL* shows better performance across a range of parameters than control models. Notably, the OpAL* and OpAL+ models achieve roughly equivalent performance in rich and lean environments in this parameter range. As noted in the introduction, standard RL models typically suffer in lean environments due to greater demands on exploration (see below for comparisons to more traditional RL models); these simulations show that OpAL* and OpAL+ overcome this robustness limitation that the No Hebb model does not (maximum AUC around 80 for lean and 150 for rich), but OpAL*'s DA modulation contributes above and beyond the flexibility endowed by the opponent and nonlinearity of OpAL+. Again, in lean environments, OpAL* improvements over the OpAL+ model were most evident for mid- and low-performing parameter sets (upside-down bowl trend in the scatter plot). See *Figure 4—figure supplement 1* for DA contributions according to parameter values.

Finally, to specifically assess the advantage of dynamic dopamine modulation, we quantified the OpAL* improvement over the balanced OpAL+ model as a function of complexity levels. Notably, OpAL*'s advantages grow monotonically with complexity, roughly doubling from low- to high-complexity levels in lean environments (*Figure 5*). Relative to the OpAL+ model, OpAL* adaptively modulated its choice policy to increase dopamine levels ($\rho > 0$) in rich environments, but to decrease dopamine levels ($\rho < 0$) in lean environments (see *Figure 2*). Indeed, performance advantages are especially apparent in reward lean environments, providing a computational advantage for low dopamine levels that can accentuate differences between sparsely rewarded options. We see similar trends when comparing to the No Hebb model, with the performance advantage of OpAL* tripling from low to high complexity and the need for Hebbian dynamics most evident in the lean environment.

## OpAL* robustly and optimally outperforms benchmark models

In the previous section, we demonstrated that the combination of OpAL*'s components (opponency, nonlinearity, and dynamic dopamine modulation) confers adaptive flexibility especially when an agent does not know the statistics of a novel environment. However, these advantages were all shown within the context of an actor-critic model of BG circuitry. In this section, we sought to evaluate whether OpAL* exhibits similar advantages compared to standard alternatives in the reinforcement literature. We include Q-learning (*Watkins and Dayan, 1992*) as it is arguably the most commonly used learning algorithm in both machine learning and biological RL (*Sutton and Barto, 2018*; *Li, 2018*; *Palminteri et al., 2015*; *Pessiglione et al., 2006*; *Frank et al., 2007a*; *Niv et al., 2012*). As demonstrated in *Collins and Frank, 2014*, a standard Q-learner shows lowered performance in lean environments compared to rich environments due to different exploration/exploitation requirements (even when its parameters are optimized jointly across both environments). We thus consider the Upper Confidence Bound (UCB) algorithm, a more strategic approach to managing the exploration/exploitation tradeoff, which is also used in both machine learning and biological RL (*Gershman, 2018*). The UCB algorithm provides a means for 'directed exploration' to options that have not been well-explored and hence the agent is unconfident about its values (see 'Materials and methods'). UCB further presents a particularly

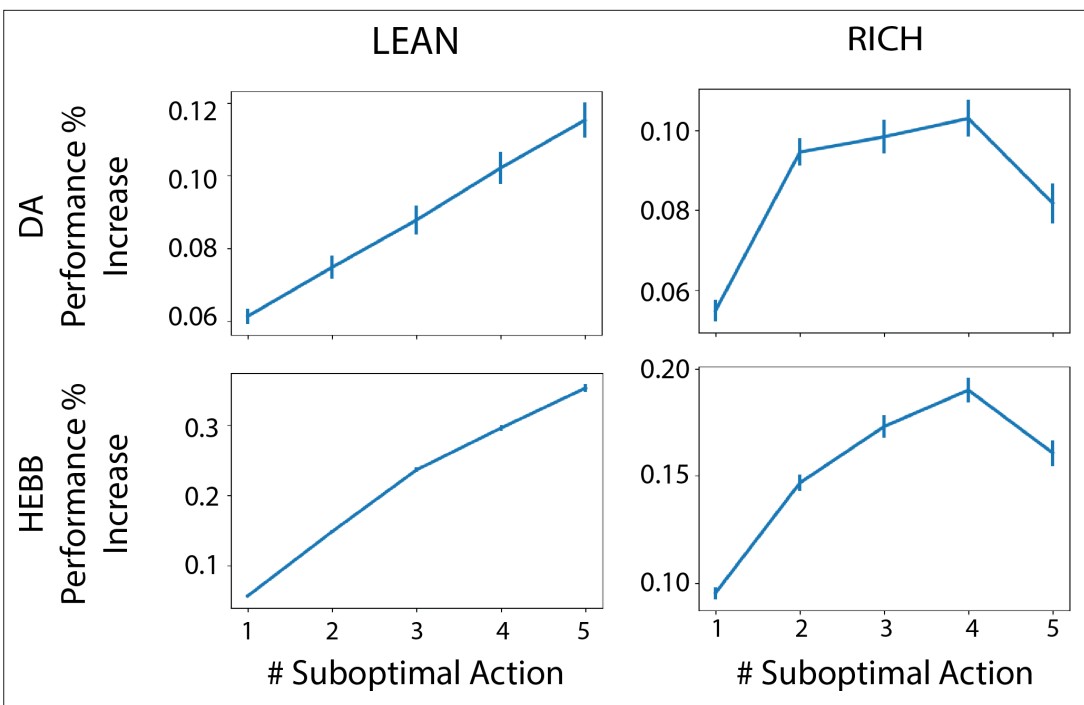

**Figure 5.** Advantage of dynamic dopaminergic modulation of OpAL* grows with complexity. Complexity corresponds to the number of bandits available in the environment (e.g., a two-armed bandit, which data point corresponds to *Figure 3*, or a six-armed bandit, which data point corresponds to *Figure 4*). Values reported are the average percentage increase of OpAL* learning curve area under the curve (AUC) compared to a OpAL+ model (top row) or No Hebb model (bottom tow) with equated parameters. That is, we computed the difference in AUC of OpAL* and OpAL+/No Hebb model learning curves for a fixed parameter normalized by the AUC of the balanced OpAL model. We then averaged this percentage increase over all parameters in the grid search. Results are shown for 250 trials of learning. Error bars reflect standard error of the mean.

challenging benchmark because it has access to the sample mean of experienced rewards for each option (i.e., it is an ideal observer), whereas RL models and OpAL only have direct access to the most recent RPEs for weight updates. In addition to serving as informative benchmarks, these models both lack the opponent characteristic of OpAL* (G and N); that is, they only learn a single decision value for each possible choice. We reasoned that OpAL* might outperform these established models: while they are guaranteed to converge to expected Q values for each option, such convergence requires repeated sampling of each alternative action, impeding the ability to maximize returns in the interim. Indeed, this limitation of RL algorithms based on action values is well known in machine learning, leading to the greater reliance on policy gradient methods in recent years. We explore these issues in more detail in the 'Mechanism' section below, showing that these issues are particularly pernicious in lean environments that are associated with reduced 'action gaps' in Q-learning agents, as well as in alternative BG opponency models that are based on value functions (*Mikhael and Bogacz, 2016*; *Möller and Bogacz, 2019*).

We compared UCB and Q-learning across a large range of parameter settings with that of OpAL*, as we did when comparing OpAL* to OpAL+ and No Hebb. For both the two-choice paradigm (80%/70% or 30%/20%) and six-choice paradigm (80%/70% × 5 or 30%/20% × 5), we conducted a grid search over the parameter space of each model and for each parameter combination calculated an average learning curve (as in *Figure 6A*). To ensure that the range of parameters was adequate for the comparison models, we verified that the optimal performing parameter set did not include the boundaries of the grid search, and we also found similar advantages when we only included parameters that were in the top 10% of performance in any given environment or complexity level (not shown). For each parameter combination, we calculated the AUC of the learning curves and then plotted histograms of these AUCs across all parameter sets (*Figure 6B*). We found overall the peak

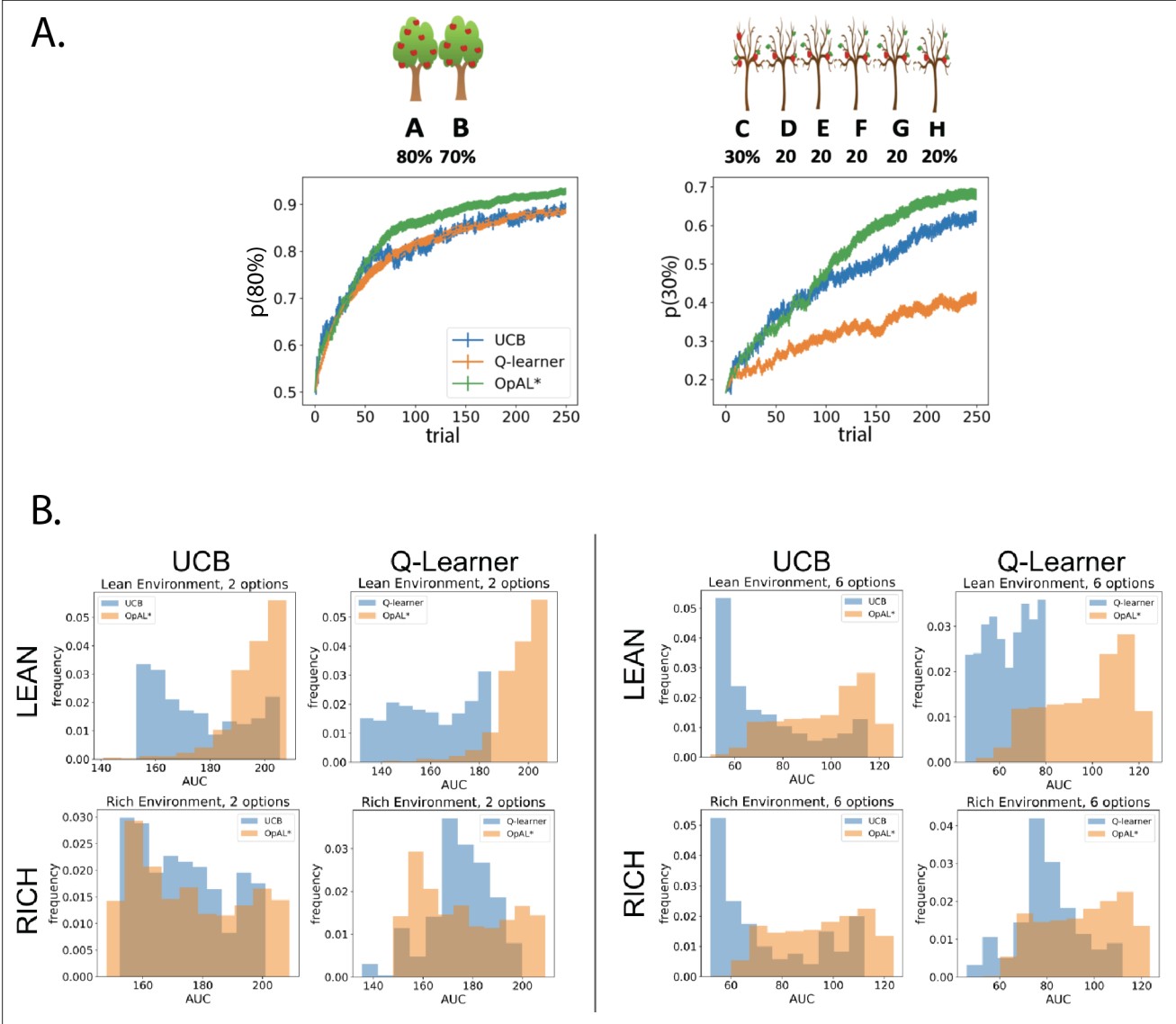

**Figure 6.** OpAL* demonstrates flexible performance across environments and complexities compared to standard-RL models, Q-learning and Upper Confidence Bound (UCB). UCB is specifically designed for improving explore–exploit tradeoffs like those prevalent in reward lean environments (see 'Mechanism' section below). While UCB demonstrates improved performance relative to Q-learning in lean environments, OpAL* outperforms both models robustly over all parameters, included the optimized set for each agent. (**A**) Biological OpAL* mechanisms outperform optimized control models across environments and complexity levels, shown here in a reward rich environment (80% vs. 70% two-armed bandit) and a lean, sparse reward environment (30% vs. 20% six-armed bandit). OpAL* outperforms a standard Q-learning and UCB in the computationally easiest scenario – reward rich environment with only two options – and, more prominently, in the computationally most difficult scenario – reward lean environment with six total options. Notably, OpAL* sustains its advantages in the difficult scenario which requires more exploration, even after extended learning (***Figure 6—figure supplement 1***). Curves averaged over 1000 simulations, error bars are SEM. Parameters correspond to the optimal performance for each environment according to a grid search, corresponding to the right tail of the histograms in (**B**). (**B**) Biological mechanisms incorporated in OpAL* support robust advantages over standard reinforcement learning models across parameter settings in a reward rich environment (80% vs. 70% two-armed bandit left, six-armed bandit right) and a lean, sparse reward environment (30% vs. 20% two-armed bandit left, six-armed bandit right). Average learning curves were produced for each parameter setting in a grid search over 1000 simulations for 250 trials, and the area under the curve (AUC) was calculated for each parameter. The AUCs of all parameters for a model are shown normalized to account for different sized parameter spaces across models.

The online version of this article includes the following figure supplement(s) for figure 6:

**Figure supplement 1.** Optimal learning curves for other complexities and time horizons.

and range of the histogram of OpAL* AUCs to be shifted rightward, with particularly notable advantages in lean environments and those with higher complexity.

We next sought to assess the best possible performance for each agent across environments, and thus allowed the parameters for that agent to be optimized. We selected the parameter set from the grid search that optimized performance for each model for various multi-bandit environments, including both reward rich (e.g., 80% vs. 70% reward probabilities) and lean (e.g., 30% vs. 20%) settings; for each agent, a different parameter combination was found to optimize performance in a given environment. As the explore–exploit dilemma becomes increasingly difficult with the number of available options, and to explore the generality of our findings, we also explored different complexity levels (two-armed bandit and six-armed bandit, where one of the options was best [e.g., 80%] and all the others were equivalent to each other [e.g., 70%]).

Notably, OpAL* outperformed both comparison agents in both the 'easiest' environment (rich, two options) and especially in the most difficult (lean, six options; *Figure 6A*). As expected, UCB demonstrated a clear improvement above Q-learning in the lean scenario (which taxes exploration–exploitation) but not the rich environment. Nevertheless, despite not having an explicit mechanism for exploring uncertain options, and despite the fact that UCB tracks the sample mean of the entire reward history for the options it has chosen, OpAL* still showed robust improvements over UCB in the lean environment. We will consider the reason OpAL* outperforms UCB in the 'Mechanism' section below.

## OpAL* adaptively modulates risk-taking

Although the above analyses focused on learning effects, the adaptive advantages conferred by dopaminergic contribution were mediated by changes in the choice function (weighting of learned benefits vs. costs) rather than learning parameters per se. We thus next sought to examine whether the same adaptive mechanism could also be leveraged for inferring when it is advantageous to make risky choices.

Models selected between a sure reward and a gamble of twice the value with unknown but stationary probability. The sure thing (ST) was considered the default reference point (*Kahneman and Tversky, 1979*), and gamble reward was encoded relative to that; that is, $R_{\text{mag}} = +1$ if gamble was won (gamble received an additional point relative to taking ST) or $L_{\text{mag}} = -1$ (loss of the ST). In high-probability gamble states, the probability of reward was drawn uniformly above 50%; in low-probability gamble states, probability of reward was drawn uniformly below 50%. Models were presented with the same gamble for 50 trials. The meta-critic tracked the $\hat{p}(r)$ of the gamble and modulated $\rho$ by its estimated expected value, as in *Equations 9–14*. G/N actors then tracked the action value of selecting the gamble. The probability of accepting the gamble was selected using the softmax choice function, such that accepting the gamble is more likely as the benefits (G) exceed the costs (N). *Act* definition can be found in *Equation 15*.

$$p(gamble) = \frac{1}{1+e^{-Act(a)}}$$

As expected, OpAL* dynamically updated its probability of gambling and improved performance in comparison to the balanced OpAL+, non-modulated model (*Figure 7*). In states with high probability (> 50%), value modulation helped the model infer that the gamble was advantageous. In low-probability gambles (< 50%), value modulation aided in avoiding the gamble, which was unfavorable in the limit. As observed in the bandit problems, DA modulation showed a larger benefit in the lean environment relative to the rich environment. By lowering its dopamine levels, OpAL* can leverage the specialization of N weights, which are more sample efficient in lean environments relative to standard RL; we elaborate in more detail in the next section the sampling tradeoff in rich and lean environments. This suggests that DA regulation is particularly helpful for avoiding risky decisions with low expected value but whose potential payoffs are larger than a guaranteed reward.

## Mechanism

How does OpAL* confer such an advantage across environments? Intuitively, by specializing on different regions of the reward probability space, OpAL* can leverage the appropriate actor that is most adept at distinguishing between low- or high-probability options. But this presumes that the actors already know the appropriate rankings, which of course they must learn. A satisfactory account

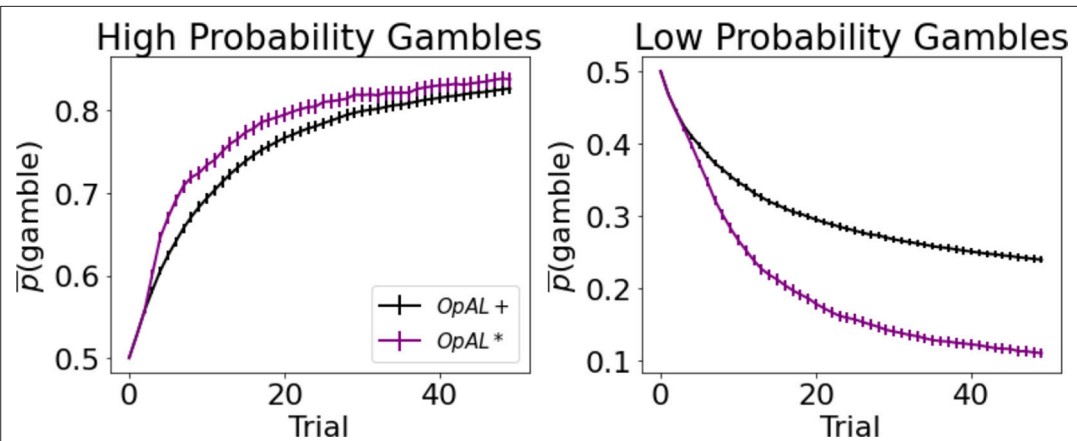

**Figure 7.** Dynamic dopamine modulation by estimated reward probability helps OpAL* decide when it is beneficial to gamble or to accept a sure reward. $\alpha_c = 0.05, \alpha_G = \alpha_N = 0.1, \beta = 5$, annealing parameter $T = 10$, and modulation parameters $k = 20$ and $\phi = 1.5$. Results were averaged over 1000 simulated states. Error bars are standard error of the mean. To limit variance, paired OpAL* and OpAL models were again given the same initial random seed.

of how OpAL* solves this problem must first address why standard RL agents suffer in lean environments. To do so, we discuss two objectives an algorithm may have: learning accurate expected values for each action or directly optimizing a policy (*Sutton and Barto, 2018*). A Q-learner is a prototypical example of the former objective; OpAL* belongs to the latter class. In this section, we contrast these objectives and their implications for cross-environment flexibility before moving on to the empirical implications of OpAL* in the next section (OpAL* captures alterations in learning and choice preference across species).

## Q learners show poor convergence and reductions in action gap with sparse reward

A key objective of a Q-learner, by construction, is that Q values converge to the expected reward for each option. However, before the algorithm converges, the policy selects actions that will necessarily be influenced by misestimation errors. Importantly, Q value convergence is impeded when the agent has to select between multiple options via a stochastic choice policy. Indeed, algorithms like Q-learning and UCB converge well when an option is well-sampled, but the speed and accuracy of this convergence are affected by stochastic sampling, leading to value estimation errors. This issue is known to weaken the 'action gap': the gap between the expected reward value for the optimal action and that of the next best option (*G. Bellemare et al., 2015*), which in turn impedes performance. In contrast, as a modified actor-critic, the actor propensities in OpAL* adjust to directly optimize performance, without representing the expected rewards for each action per se. Nevertheless, opponent actor weights retain ordinal (but nonlinear) rankings of action–outcome associations which can be used for action selection among novel pairs of actions (*Figure 1*; *Collins and Frank, 2014*). Below we show that value misestimation errors and the action gap are particularly challenged in lean environments, and that OpAL* remedies this difference.

To investigate estimation errors in algorithms that choose based on learned values, consider two metrics: the difference of a representation to the ground truth (e.g., difference between Q-value and 0.8 for an option that yields reward 80% of the time) and the action gap (the difference between Q-values of the best option and second best option). The former tracks convergence of true expected values and the latter reflects the effectiveness of the current policy (assuming a softmax-like policy, whereby the relative difference in value determines the probability of action selection). An algorithm that has not converged may still nonetheless have an effective policy.

Note that because an agent typically only learns about what it has chosen, convergence is hindered for those actions it has not selected. When an agent is given full counterfactual information about rewards for all actions irrespective of choice, the best and second best options in rich and lean

environments converge quickly to their true targets (0.8 vs. 0.7 or 0.3 vs. 0.2, respectively) and the action gap for both environments converges to the true value of 0.10 at similar rates (*Figure 8A*, left). In contrast, when an agent receives feedback only for actions it has chosen, both the action gap and convergence are impeded, with a stark difference in rich versus lean (these differences are further amplified as more actions become available, see *Figure 8—figure supplement 1*). Notably, the action gap in the rich environment is larger than that of the lean environment (and even than that under full information). This is because in the rich case, once the agent begins exploiting the optimal action, its value converges (black line) while that of the second best option's value remains underestimated (gray line plateaus). This lack of convergence for the suboptimal action (0.7) is actually helpful for an effective policy as its value remains closer to the initialization value and thereby increasing the action gap. Conversely, as options in the lean environment are sampled, their values *decrease*, and repeated sampling of the best option (0.30) could cause it to become estimated as undesirable relative to a less-explored (but truly suboptimal) action. Thus, misestimation errors are particularly pernicious in a lean environment, preventing exploitation of the best options until all have sufficiently converged. As such, the estimates of both the best and second best options actually converge more reliably on average (though remain slightly underestimated), but the policy suffers until this is the case.

While the above analysis focuses on average convergence for each individual run, another way to investigate the robustness of an algorithm's policy is to consider the proportion of simulations where the best action has an estimated Q-value greater than that of the second best (i.e., how often is the action gap positive). This 'action value ranking' can then be compared to the analogous rankings in OpAL*, wherein we assess the rankings of G weights on rich environments and N weights in lean environments (given that these weights dominate in their respective environment). Again, we observe that Q-learner shows a higher proportion of simulations in the rich environment with proper rankings than in lean. Importantly, and in contrast, OpAL* shows balanced action value rankings in rich and lean environments, which directly translates to similar rich and lean performance in terms of policy (*Figure 8A*, right). Because the rich versus lean asymmetry in the Q-learner stems from the stochastic policy, these effects are amplified further by increasing complexity via the number of suboptimal actions, whereas OpAL* preserves cross-environment performance in these cases (see *Figure 8—figure supplement 1*).

## Opponency and Hebbian nonlinearity allows OpAL* to optimize action gaps

Having characterized the divergent behavior in rich and lean environments for value-based agents, we now illustrate how opponency and nonlinearity allow OpAL* to overcome these differences. Let us first consider OpAL* dynamics given full information. Because of the nonlinearities in opponent actors, convexity in the G and N weights imply that the two actors differentially specialize in discriminating between high and low reward probability options, as shown in *Figure 1B*. Thus, increasing DA amplifies the G actor's contributions to choice, which increases the action gap for high-probability options. Conversely, lowering DA amplifies the N actor's contributions, which increases the action gap for low-probability options. As the Bayesian meta-critic converges on an estimate of environmental richness, OpAL* can adapt its policy to dynamically emphasize the most discriminative actor and appropriately enhance the action gap to optimize the policy (*Figure 2*, left). In contrast, a variant which lacks nonlinearity (No Hebb) induces redundancy in the $G$ and $N$ weights and thus essentially reduces to a standard actor-critic agent. As such, dopamine modulation does not change its discrimination performance across environments and the action gap for choice remains fixed (*Figure 1—figure supplement 1*).

However, the above logic depends on the agent having access to properly ranked actions, the question remains as to how OpAL* can leverage this action gap when full information is not provided (and therefore the action gap is not guaranteed to be positive during early learning). As highlighted above, sparse reward environments typically result in more stochastic sampling because repeated selection of the optimal action often leads to its value decreasing below that of suboptimal actions during early learning, causing the agent to switch to those suboptimal actions again until they become worse, and so on until convergence. This vacillation is evident in the comparison models (Q-learning and No Hebb), which are susceptible to substantial policy fluctuations in lean environments (*Figure 8B and*

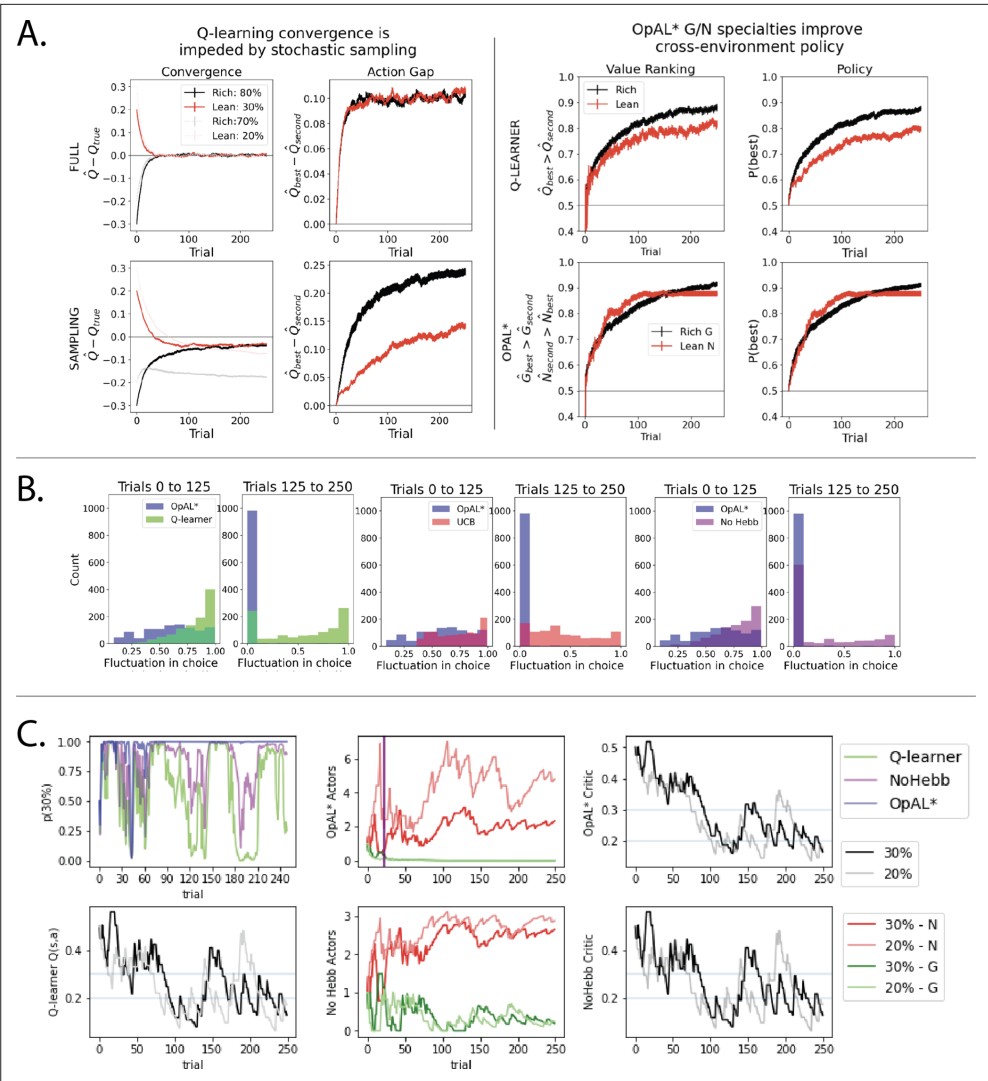

**Figure 8.** Overview of mechanisms contributing to performance differences. Simulations in a and b conducted using the cross-environment optimized parameters according to grid search and 1000 simulations. (**A**) Left: stochastic sampling of actions hinders a Q learner's convergence to true Q-values in lean environments (red lines) relative to rich environments (black/gray lines), causing reduced action gaps in lean environments. This difference is absent when full information is provided to the agent about reward outcomes for each alternative action (top row), implying that it is due to sampling (bottom row). Right: the proportion of simulations where the Q-value of the optimal action (80% or 30%) exceeds that of the suboptimal option (70% or 20%) differs in rich vs. lean, impeding the policy. In contrast, OpAL* capitalizes on the specialization in the N actor to discriminate between sparse reward outcomes, demonstrating comparable performance in both environments, as seen empirically in rodent behavior (*Hamid et al., 2016*). Similar patterns are seen in higher complexity environments (*Figure 8—figure supplement 1*). Learning curves reflect mean of 1000 simulations; error bars reflect standard error of the mean. (**B**) OpAL* exhibits reduced policy fluctuations in lean environments. Policy fluctuations are indexed by variability within simulations of the sign of $p(\text{choose } 30) - p(\text{choose } 20)$. Higher standard deviation of this metric indicates more fluctuations, which are common in comparison models due to convergence issues. (**C**) OpAL* avoids policy fluctuations due to nonlinear accumulation of reward prediction errors (RPEs). Here, each agent experienced a fixed policy and reward sequence for a two-armed bandit (30% vs. 20%) to visualize learning dynamics for a fixed sequence of events. Left, top: each agent's policy had they been able to freely choose. After initial exploration period, OpAL* reliably selects the optimal action; other agents again showed extended policy fluctuations. Middle: in OpAL*, N weights accumulate nonlinearly with RPEs for suboptimal 20% action, differentiating from 30%. This pattern is not present without Hebbian nonlinearity. Vertical bar indicates where dynamic DA is engaged as meta-critic determines the environment is lean, so OpAL* policy relies on the N actor. Similar patterns were observed for other random seeds.

*Figure 8 continued on next page*

*Figure 8 continued*

The online version of this article includes the following figure supplement(s) for figure 8:

**Figure supplement 1.** Q-learning convergence issues are amplified as the number of suboptimal actions increases.

**Figure supplement 2.** Upper Confidence Bound (UCB) convergence impacted by multiaction sampling.

**Figure supplement 3.** Alternative opponent model that lacks three-factor Hebbian dynamics (*Möller and Bogacz, 2019*) is impacted by multiaction sampling.

**Figure supplement 4.** Consistency of reward prediction error (RPE) valence in OpAL* induces convexity.

*C*). (Below we will consider policy fluctuations that also impede performance in UCB as the number of alternative actions grow, but that issue is not specific to lean.).

OpAL* overcomes this issue in the lean environment in two ways. Firstly, in the early stages of learning, neither actor dominates action selection (the weights have not yet accumulated, and dynamic DA has not yet engaged to preferentially select an actor). Thus the nondominant (here, $G$) actor contributes to the policy early during learning, thereby flattening initial discrimination and enhancing exploration. In this sense, the exploration/exploitation tradeoff in OpAL* operates at the meta-critic level: it needs sufficient exploratory experience early on to be confident that the environment is one that should be exploited primarily by one actor or the other, whereas the actual action is then selected as a function of actor weights therein.

Secondly, and more critically, OpAL* more quickly allows the N weights to discriminate between low-probability options. In early stages of learning, the Hebbian nonlinearity ensures that negative experiences induce disproportional distortions in $N$ weights (*Figure 2*), more rapidly increasing the action gap between optimal and suboptimal options with less stochastic sampling required.

To evaluate this claim systematically, we fixed both the policy and feedback across all agents so that they experienced the same choices and outcomes in a lean environment (*Figure 8C*). We then evaluated how such sequences of events translated into changes in the critic's evaluation and the resulting N weights/Q values. We also plot the softmax probability of selecting the optimal action had the agent been able to freely choose according to its (forced) experiences. Parameters selected were those which optimized performance across environments for each agent according to a grid search.

First, we note that OpAL* terminates its exploration earlier than alternative models (No Hebb and Q-learning). Second, we observe while Q-values continue to oscillate in their rankings throughout the trials, OpAL*'s N weights maintain a proper ranking after the first 50 trials. These dynamics importantly rely on the Hebbian term (the No Hebb model, while experiencing fewer fluctuations than the Q-learning model, has a smaller, and at times negative, action gap relative to OpAL*). The central mechanism by which OpAL* maintains consistent adaptive policy in lean environments is that the N weights nonlinearly accumulate with the history of negative RPEs induced by the critic (*Figure 8C*). Note that, given the fixed policy and outcomes, the critic itself closely follows the dynamics of both the No Hebb critic and the Q-learner. However, recall that in OpAL*, the update in $G$ and $N$ weights occurs in proportion to not only the critic RPEs, but to the prior $G$ or $N$ weight itself (i.e., *Equations 3 and 4*). Thus weight updates are influenced not only by the current RPE but also by previous RPEs that the $G$ or $N$ weights have accumulated.

Indeed, expanding the recursive actor weight update equations, resulting update in a given trial can be written as:

$$\Delta G_t(a) = \alpha_G \delta_t (1 + \sum_{i=1}^{t-1} \alpha_G \delta_i + \sum_{i=1}^{t-2} \sum_{j=i+1}^{t-1} \alpha_G^2 (\delta_i \delta_j) + \sum_{i=1}^{t-3} \sum_{j=i+1}^{t-2} \sum_{k=j+1}^{t-1} \alpha_G^3 (\delta_i \delta_j \delta_k) + \dots + \prod_{i=1}^{t-1} \alpha_G^{t-1} \delta_i)$$

(22)

See the section 'Derivation of OpAL actor weights as a function of RPE history' for full derivation.

The first term is just the standard actor update as a function of the RPE in the current trial. But one can see that the update is additionally influenced by the sum of all of the previous RPEs, each of them equally weighted by $\alpha_G$, and thus OpAL* updates implicitly have access to the entire history of RPEs. Moreover, updates are further scaled by higher order terms comprising each pair of previous RPEs, scaled by $\alpha_G^2$, and so on. As such, the actor weights grow nonlinearly with the *consistency* of RPE's: when the preponderance of RPEs is positive ($\delta > 0$), the second-order terms will lead to superlinear

weight changes, but when they are mostly negative, these higher order terms will pull weight changes in the opposite direction. The same form applies to updates of $N$ weights, but where each $\delta_i$ is replaced by $-\delta_i$.

Note that before the critic has converged, the sum of the accumulated negative RPEs (the first term above) is larger than that of positive RPEs for lean options, and vice versa for rich options. The nonlinear accumulation translates into disproportionately larger N weights for the most suboptimal actions (and conversely, larger G weights for optimal actions in a rich environment). Note also that, unlike standard RL, higher actor learning rates in this scheme do not imply that weight changes are primarily influenced by the current RPE; rather, here they imply more influence of the higher order terms, leading to more convex weights. (See Supplemental note 5 in Appendix 2). See *Equation 22* and 'Derivation of OpAL actor weights as a function of RPE history' for more details.

The net result is that OpAL* can optimize and stabilize its policy well before critic or Q value convergence (at which point the expectation over RPEs is zero, and higher order terms induce decay in actor weights, although this is largely mitigated by annealing). (See Supplemental note 6 in Appendix 2). This notably contrasts with Q-learning, where slow convergence to ground truth values is detrimental to performance. By adapting its policy by environmental richness (and confidence therein), OpAL* can dynamically leverage this specialization to quickly optimize performance, well before the critic converges, avoiding an explore–exploit tradeoff that is especially vexing in lean environments.

A particularly striking result is that OpAL* exhibits advantages over UCB, even though UCB has access to the sample mean of an action's reward history (i.e., perfect memory). But in order to obtain that sample mean, the agent has to sufficiently explore it. Akin to Q-learning, UCB's objective is to learn accurate value estimates. The main difference is that while a Q learner explores stochastically via softmax, UCB's exploration is directed toward those actions for which the values are most uncertain, allowing it to obtain high-certainty upper-bound estimates of the action value. As such, like Q-learning, UCB demonstrates slowed convergence when it has to choose among multiple actions, which is further impeded as the number of actions grows; (*Figure 8—figure supplement 2*). In contrast, OpAL* exploration serves to optimize a policy, without seeking precise value estimates. Indeed, while OpAL* does exhibit early exploration before switching to exploitation, this transition occurs earlier due to the growing action gap and dopamine modulation as described above. In contrast, UCB demonstrates a more gradual and prolonged exploration phase (*Figure 8B*) even when the correct action is well estimated (*Figure 8—figure supplement 2*, lean, 600–1000 trials, two options), thereby impeding performance. Notably, this tradeoff is particularly intensified as the number of options grows because the UCB bonus will serve to enhance exploration to all of these options. Accordingly, when we optimized UCB's exploration bonus parameter, we found that the best it could do is reduce the exploration bonus in high-complexity environments to counter this impediment (not shown; but see histograms in which UCB shows variable performance for different levels of this parameter across environments; *Figure 6*).

We conclude this discussion by considering whether OpAL* might simply induce a more efficient change from exploration to exploitation across learning independent of its specialized opponent actors. The favorable comparison to UCB (*Figure 6*) suggests this is not the case because effectively the UCB algorithm is designed to do just that in a more sophisticated way (exploration directed toward options that have not been sampled sufficiently but then exploit after that). To further diagnose whether dynamically modifying the softmax temperature alone is sufficient to improve robustness within an opponency model, we simulated a control variant in which DA levels were used to dynamically increase both $\beta_G$ and $\beta_N$ together, independent of the sign of $\rho$ ($\beta$ modulation model, see Appendix 1 'Comparison to softmax temperature modulation'). OpAL* outperformed the $\beta$ modulation in rich environments and was able to more rapidly learn in lean environments. These simulations show that while dynamic changes in softmax temperature may be sufficient to improve performance in one environment, the dynamic shift from one specialized actor to another is integral to flexibility across both environments.

To summarize, agents that prioritize learning accurate action values (including 'standard' RL) make qualitative predictions that performance should be significantly hindered in lean compared to rich environments. OpAL* shows substantially improved performance in lean environments due to its opponent and nonlinear properties, especially when DA is modulated dynamically to capitalize on these properties. These are testable predictions. In line with these qualitative patterns, rodents

showed equally robust learning in rich environments (90% vs. 50% bandit task) compared to lean environments (50% vs. 10% bandit task) in *Hamid et al., 2016* (see Figure 1d of that paper). We explore more detailed simulations of empirical data across species below.

## Advantages in lean environment are not seen in other opponent BG models lacking Hebbian nonlinearity

Notably, opponency alone is not sufficient to remediate this divergence in rich and lean performance. Indeed, we also analyzed an alternative model of D1/D2 opponency presented by *Möller and Bogacz, 2019*. This model does not include the Hebbian term, but does include a different nonlinearity which allows the $G$ and $N$ weights to converge to the mean expected payoffs and costs in the environment. This property serves as a useful comparison: once costs and benefits for each action are known, an agent should be able to choose its policy to maximize reward (and/or manage risk). However, similar to Q-learning, the convergence to expected payoffs and costs in this model is only guaranteed in the limit after repeatedly selecting the same action, and is subject to the same convergence impediments when faced with stochastic action selection. Moreover, this control model serves as another test for the utility of the Hebbian term and the resulting convexity of OpAL* G/N weights as described below. We tested this model with a two-armed bandit with the same reward contingencies as those in the rich and lean environments, but incorporating an explicit cost for incorrect choices (lmag = –1) to better align with the model's scope. (See Supplemental note 7 in Appendix 2). These simulations revealed similar properties to Q-learning: shallowing of action-gap and value ranking curves between rich and lean environments, and slowed convergence relative to full information models, again showing that they are due to the dependence on action selection (*Figure 8—figure supplement 3*).

Notably, for the models proposed by *Möller and Bogacz, 2019*, the G weights show stronger discrimination between actions in reward lean environments, whereas the N weights show stronger discrimination in rich environments (*Figure 8—figure supplement 3C*), the opposite of OpAL*. If this model were to vary its dopamine states similarly to OpAL* in order to amplify the contribution of the more informative actor, it would require adjusting DA in the opposite direction, with higher dopamine in reward lean environments and lower dopamine in reward-rich environments at choice, contrary to what has been found empirically (*Mohebi et al., 2019*; *Hamid et al., 2016*). *Moeller et al., 2021* found that human participants do show higher risk-taking for richer reward contexts and lower risk-taking for leaner reward contexts, in line with OpAL* predictions. When they allow stimulus onset to induce an RPE, the *Möller and Bogacz, 2019* model also accounts for this same empirical risk-taking pattern. However, as shown above, it would still show impeded learning in lean environments for discrimination bandit tasks as simulated in this article. Furthermore, in contrast to OpAL*, the nonlinearity used in their model induces concavity rather than convexity in actor weights, and thereby predicts the incorrect pattern of findings for the impact of DA manipulations on discrimination learning and choices amongst high and low rewarding options in Parkinson's patients (see Figure 10 in *Mikhael and Bogacz, 2016*). Many studies have replicated the pattern predicted by OpAL*, whereby PD patients off medication better discriminate between lean options, whereas on medication they better discriminate between rewarding options (*Frank et al., 2007b*; *Frank et al., 2004*; *McCoy et al., 2019*; *Kobza et al., 2012*; *Weismüller et al., 2018*; *Smittenaar et al., 2012*; *Shiner et al., 2012*). These observations further emphasize the need for the three-factor Hebbian nonlinearity for OpAL*'s normative properties but also its accordance with empirical data.

## OpAL* captures alterations in learning and choice preference across species

While all analyses thus far focused on normative advantages, the OpAL* model was motivated by biological data regarding the role of dopamine in modulating striatal contributions to cost/benefit decision-making. We thus sought to examine whether empirical effects of DA and environmental richness on risky choice could be captured by OpAL* and thereby viewed as a by-product of an adaptive mechanism. We focused on qualitative phenomena in empirical data sets that are diagnostic of OpAL* properties (and which should not be overly specific to parameter settings) and that could not be explained individually or holistically by other models. In particular, we consider impacts of optogenetic and drug manipulations of dopamine and striatal circuitry in rodents and humans. We further show

that OpAL* can capture economic choice patterns involving manipulation of environmental reward statistics rather than DA.

## OpAL* accounts for counterintuitive human choice preferences for loss-avoiding options over those that produce net gains

As noted in the above 'Mechanism' section, instead of converging to veridical values, OpAL*'s G/N weights serve to quickly rank the relative value of options, optimizing the policy across environments with varying reward statistics. Importantly, OpAL* retains the ranked values for a given environment (*Figures 1B and 4*), affording transitive choice amongst them (as in the probabilistic selection task and impacts of DA manipulations simulated previously by *Collins and Frank, 2014*).

Nevertheless, given that its actor weights are governed by history of critic RPEs, OpAL* does predict that the relative value of other options can influence how an action is learned, which may produce counterintuitive behavior when the overall reward richness of a context changes. Previous research supports this notion (*Klein et al., 2017*; *Palminteri et al., 2015*; *Gold et al., 2012*; *Geana et al., 2022*). For example, participants in *Palminteri et al., 2015* learned to select between fixed pairs of stimuli with different reward probabilities (*Figure 9*). For some stimuli, participants could maximize rewards, whereas for others they could simply avoid losses. In a post-learning transfer phase, participants were given the option to express preferences among novel combinations of stimuli, participants counterintuitively preferred to choose an option that had mostly avoided a loss but still infrequently produced a loss 25% of the time (L25) over an option that had strictly positive value (infrequent gain; G25). Moreover, this seemingly irrational pattern was observed only when participants were given full information during learning about both the outcome of their action and that of the action they had not chosen. As *Palminteri et al., 2015* describe, these results can only be captured if participants learn not the absolute values of options, but instead the relative value of options within their context. To determine whether these patterns can be captured also by OpAL*, we simulated the same task contingencies and allowed the critic to reflect the context (state) value (*Figure 9*). Because full information was given, we allowed the critic to learn quickly here, such that a less frequent loss (L25) stimulus mostly induces a positive prediction error in its relative context (by avoiding a loss 75% of the time), and conversely the less frequent gain (G25) mostly elicits a negative RPE in its context; full information amplifies this effect as the critic has a more accurate estimation of context value. While these qualitative results do not rely on the dopamine modulation component of OpAL*, they highlight the policy optimization that DA leverages. However, OpAL* does raise a new counterintuitive prediction: if participants were in a rich environment or administered DA enhancing drugs prior to the transfer phase, preference for a loss avoider (L25) should *increase* as this would promote the contribution of G weights which dominate in the loss avoider.

## Striatal D2 MSN activity and reward history alter risky choice in rodents

Perhaps the most germane empirical study to OpAL since the original model was developed is that of *Zalocusky et al., 2016*, who studied rodent risky choice as it is altered by reward history, dopamine manipulation, and striatal activity. Rats repeatedly chose between a certain option with a small reward or a gamble for larger reward whose expected value matched that of the certain option. Following unsuccessful gambles, they observed increased activity in D2-expressing MSNs in ventral striatum during subsequent decision periods. (See *Figure 10* for illustration of these results.) Recall that in OpAL*, reward history alters DA levels, which in turn modulate activity in striatal MSNs and accordingly cost/benefit choice. In this case, a reduced recent reward history should reduce striatal DA, elevate D2 MSN activity, and thus promote choices that avoid costs. Indeed, Zalocusky et al. observed that animals were more likely to make a 'safe' choice when D2 MSNs were stimulated during the choice period, and that endogenously, such safe choices were related to increased D2 activity and enhanced following unfavorable outcomes. Together, these results suggest an trial-to-trial adaptation of choice (rather than learning) driven by changes in D2 activity, akin to OpAL* mechanisms. Furthermore, such optogenetic stimulation effects were only seen in animals with a baseline preference for risk-seeking; risk-averse animals exhibited no change in behavior with the phasic manipulation.

Note first that these patterns of results are inconsistent with classical models in which striatal D2 activity is related only to motor suppression; here the impact of D2 activity is not to suppress actions

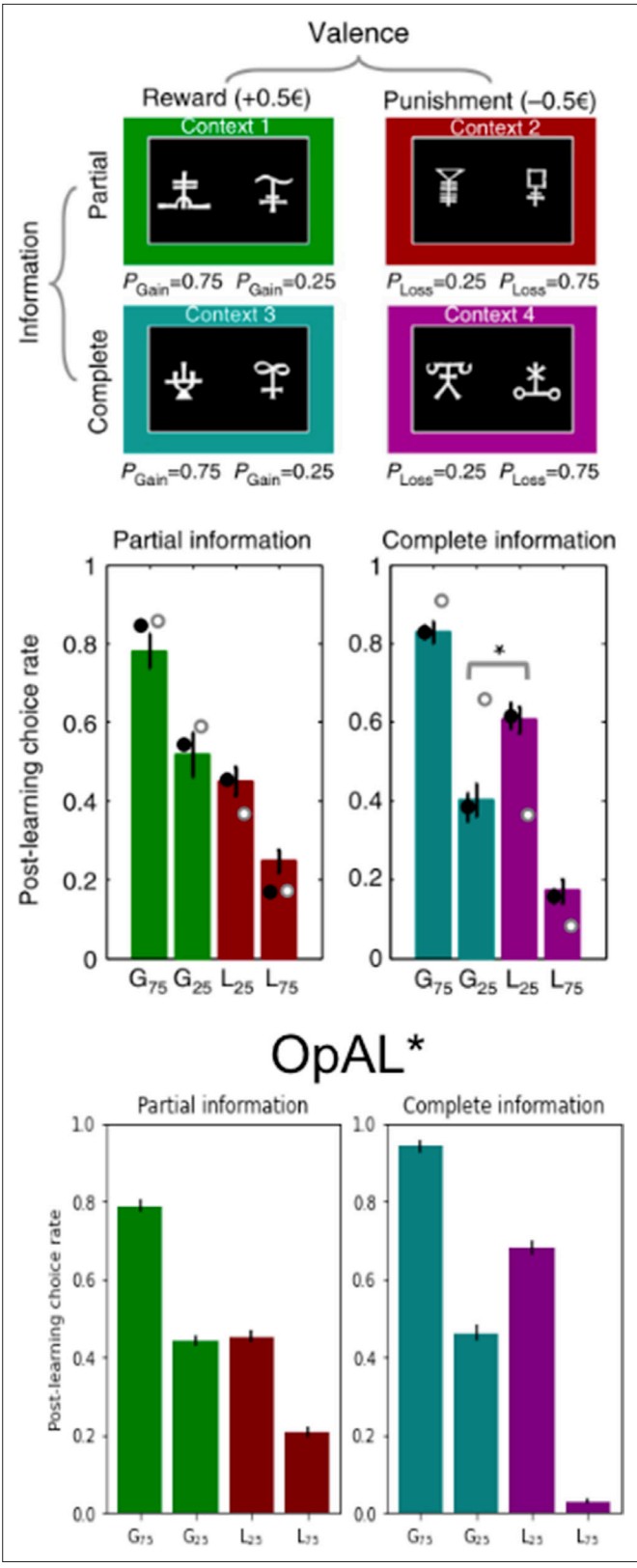

**Figure 9.** OpAL* captures counterintuitive and context-dependent human choice preference in *Palminteri et al., 2015*. In *Palminteri et al., 2015*, participants learned to select between two stimuli to maximize their rewards. Participants learned about eight stimuli in total, where two stimuli were always paired together during the initial learning phase for a total of four 'contexts.' Contexts varied in valence – either rewarding or punishing – and varied

*Figure 9 continued on next page*

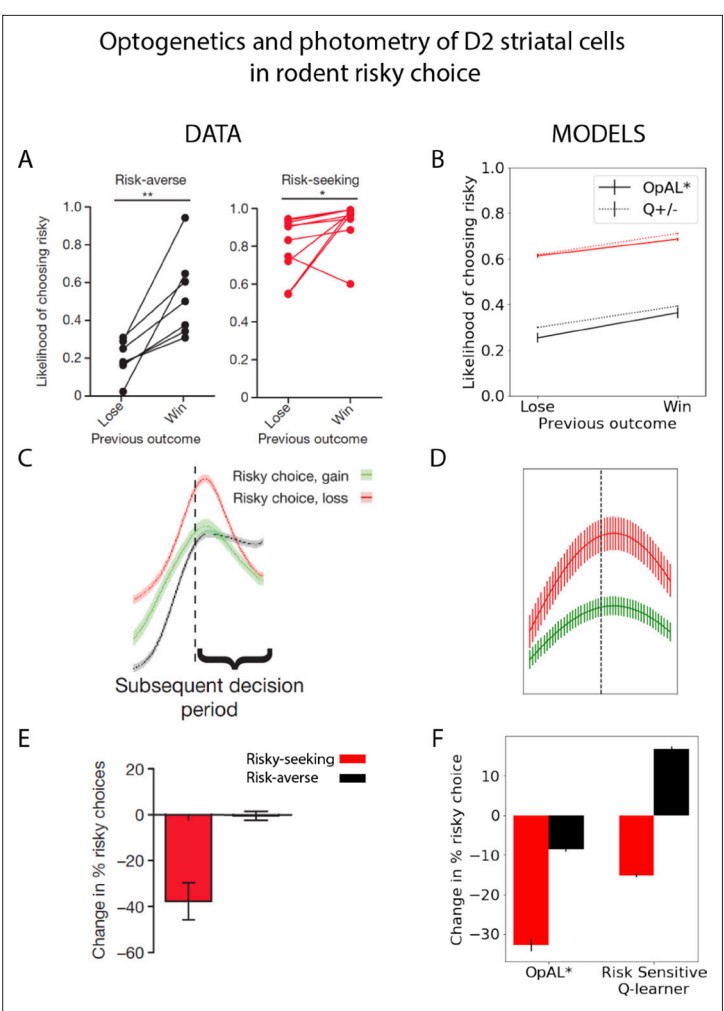

**Figure 10.** Striatal D2 medium spiny neuron (MSN) activity and reward history alter risky choice in rodents. Rodents repeatedly selected between a certain option with low magnitude of reward and a gamble with higher payout when successful. Left column: reproduced from *Zalocusky et al., 2016*. Right column: model simulations with OpAL* and risk-sensitive RL (RSRL). (**A, B**) Both risk-averse and risk-seeking animals are more likely to avoid a gamble after a gamble 'loss' (failure to obtain the large reward). Both OpAL* and RSRL, a standard Q-learner with different learning rates for positive and negative prediction errors, can capture this trend, via changes in either choice function (D1 vs. D2 MSN contributions) or learning rates, respectively. Error bars for OpAL* show standard error of the mean. (**C, D**) D2 MSN activity, measured via photometry, is larger after a gamble loss (red) than a gamble win (green) during the subsequent decision period. This pattern is reproduced in OpAL*, whereby D2 MSN activity is influenced by the product of the $N$ weights and the adaptive $\beta_n$, which amplifies D2 MSN activity when dopamine levels are low. The simulation peak represents the average of this product after a loss or after a win, which is carried over to subsequent choices; error bars reflect SEM across simulations and dynamics before and after peak were generated by convolving the signal with a sinusoidal kernel for illustrative purposes. (**E, F**) Optogenetic stimulation of D2 MSNs during the choice period induces risk-aversion selectively in risk-seeking rats. OpAL* captures this preferential effect by magnifying the effective D2 MSN activity and inducing avoidance primarily in risk-seeking agents. In contrast, RSRL predicts opposite patterns in risk-seeking and risk-averse animals. Error bars for simulations show standard error of the mean. Parameters OpAL*: $\beta = 1.5, \alpha = 1., T = 20, k = 1.1, \phi = 1.0$. Baseline $\rho$ risk-seeking (0.85) and risk-averse (–0.75). Parameters RSRL: risk-seeking $\alpha_+ = 0.3, \alpha_- = 0.1$; risk-averse $\alpha_+ = 0.1, \alpha_- = 0.3, \beta = 1.5$. Since optogenetic effects were evident primarily during the choice period, we modeled this by changing the choice function in both models: in OpAL, trial-wise $\rho$ values were decreased by 1.0 to mimic increased D2 MSN activity/decreased DA. In RSRL, the choice function was altered by reducing $\beta$ (to 0.01), leading to opposite directional effects in risk-seeking and risk-averse agents. Agents selected between a certain option and a 50/50 gamble with twice the payout for 100 trials.

*Figure 10 continued on next page*

in information – either only feedback of the selected action was provided (partial) or counterfactual information was also provided (complete). In the gain contexts, one action had a probability of receiving reward 75% of the time or nothing (G75%); the other received reward 25% of the time or nothing (G25%). In the loss contexts, the preferred action received a punishment 25% of the time or nothing (L25%); the other received a loss 75% of the time or nothing (L75%). In a post-learning transfer phase, participants were presented with stimuli every pair-wise combination regardless of context and were asked to choose which stimuli they preferred. For items learned under complete information, participants counterintuitively preferred the more frequent loss avoider (L25%) to the less frequent rewarding stimulus (G25%). While a standard Q-learner would predict preference according to expected value as is seen in the partial information results, a model which incorporates the relative value of an action to the state value into its prediction error would predict the counterintuitive results as *Palminteri et al., 2015* propose. OpAL* can similarly capture these results by allowing the critic to quickly reflect the context value. Like in the model proposed by *Palminteri et al., 2015*, complete information allows the critic to converge more quickly to the state value and produce positive prediction errors for the frequent loss avoider (L25%) and negative prediction errors for the infrequent reward earner (G25%). Error bars for OpAL* show standard error of the mean. Top images (paradigm and empirical findings) reproduced from Figure 1a of *Palminteri et al., 2015*.

altogether but instead to bias choices toward safe options. Instead, these results are consistent with OpAL* in which D2 activity is related to promoting actions with the lowest perceived cost. Indeed, we found that this pattern of results align with the predictions of OpAL* but not alternative risk-sensitive models (see below).

As in previous sections, we encode gamble outcomes relative to the certain option: $R_{mag} = +1$ if gamble was won or $L_{mag} = -1$. For OpAL*, the critic and actors operated as in the section 'OpAL* adaptively modulates risk-taking.' G/N actors then tracked the value of selecting the gamble using the prediction error generated by the critic. As before, the probability of accepting the gamble was selected using the softmax choice function.

To simulate risk-seeking and risk-averse rats, we modified the baseline DA levels ($\rho$), holding all other parameters constant. Risk-seeking rats were modeled by higher levels of baseline $\rho$ relative to those of simulations for risk-averse rats. To model phasic optogenetic stimulation, $\rho$ values were decreased by a constant amount from this baseline.

We contrasted OpAL* to alternative models in which risky choice could be adapted. A popular model of dynamics in risky choice is called 'risk-sensitive RL,' in which an agent learns at different rates from positive and

$$Q(t+1) = Q(t) + \alpha_+ * PE, \text{ if } PE >= 0,$$
$$Q(t+1) = Q(t) + \alpha_- * PE, \text{ if } PE < 0$$

where actions are selected using softmax function over Q values. If $\alpha_+ < \alpha_-$, an agent is more sensitive to risks in its environment. This formulation has been useful for characterizing asymmetric impacts of dopamine bursts and dips (*Frank et al., 2007a*; *Niv et al., 2012*), but focuses on learning rather than changes in choice functions. Because the effective manipulations on risky choice were made during the choice period rather than outcome, learning rate manipulations alone could not capture the effects. However, it is possible that DA or D2 manipulations can affect choice in simple RL models via simple changes to the overall softmax temperature, as assumed by many models (*FitzGerald et al., 2015*; *Cinotti et al., 2019*; *Eisenegger et al., 2014*; *Lee et al., 2015*; *Humphries et al., 2012*). We thus allowed the RSRL model to exhibit changes in risky choice by manipulating softmax gain accordingly, whereby D2 stimulation would mimic low DA levels and hence lower gain.

We found that both OpAL* and RSRL accounted for the decrease in gamble choices after gamble losses relative to wins, but generated opposing predictions for decision-period manipulation of D2-expressing neurons. While OpAL* predicts a decrease in riskiness in both risk-seeking and risk-averse rats (but more strongly in risk-seeking rats), RSRL predicts a decrease in riskiness in risk-seeking rats but an *increase* in riskiness in risk-averse rats. The reason for this effect is simply that a change in

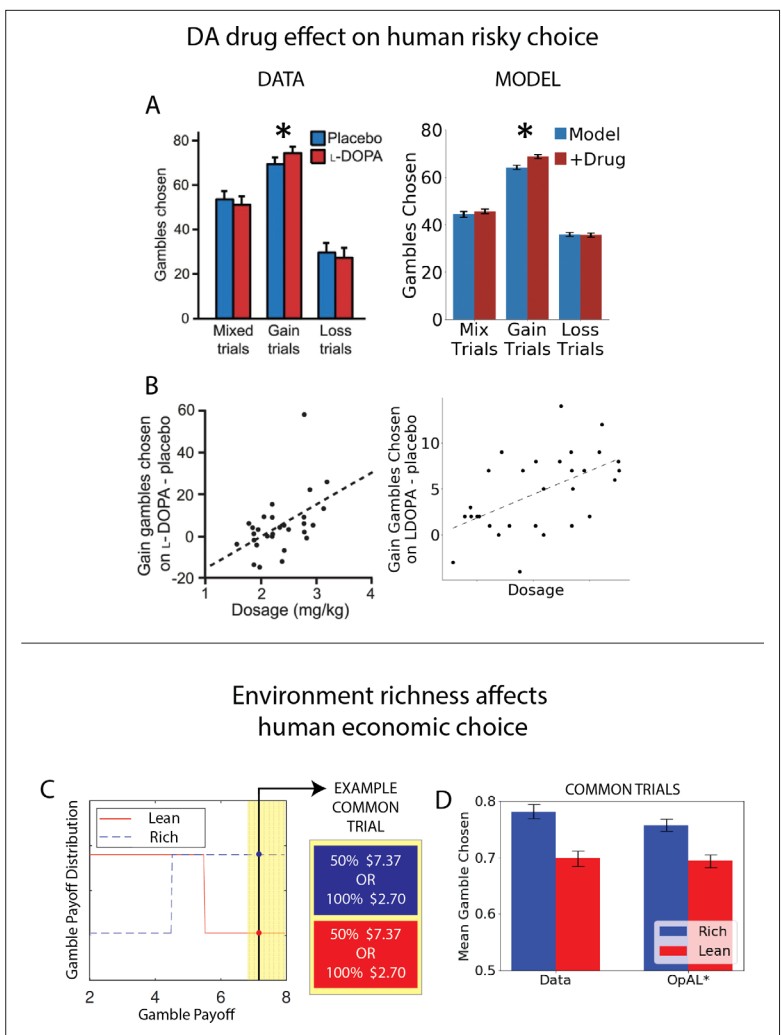

**Figure 11.** OpAL* captures human risk-taking patterns and their manipulation by drug and environmental context. (**A, B**) Dopamine (DA) drug effects on risky decision-making and individual differences therein. OpAL* captures behavioral risk patterns of healthy participants on and off L-DOPA, a drug which boosts presynaptic DA. (**A**) L-DOPA administration selectively increased risky choice in gain trials, where choice was between a sure reward and a 50% gamble for a larger reward, as compared to loss trials (sure loss vs. gamble to avoid loss) or mix trials (in which gambles could result in gains or losses). Error bars for OpAL* show standard error of the mean. Left: modified figures from **Rutledge et al., 2015**. (**B**) These effects were larger for subjects with higher effective drug doses, Spearman's $\rho = 0.47, p < 0.01$. Left: modified figures from **Rutledge et al., 2015**. Right: OpAL* simulations reproduce these selective effects. Spearman's $\rho = .50, p < .01$ To model individual differences in effective drug levels, for each pair of model on and off drug, $d$ was drawn from a normal distribution centered at 0.5 with variance 0.25. Parameters: $\beta = 1.5, k = 1$. (**C, D**) Risky decisions are sensitive to environmental richness In contrast to other empirical results discussed where dopamine pathways were directly manipulated, **Frydman and Jin, 2021**, manipulated reward statistics of the payoffs in the environment, as in our normative simulations. Participants chose between a certain reward and a 50% gamble over two blocks. The distribution of payoffs in each block was either Rich (higher frequency of large magnitudes) or Lean (higher frequency of small magnitudes). Crucially, each block contained predetermined 'common trials' where the payoff of both the gamble and certain option was fixed (e.g., an offer 50% $7.13 vs. 100% $2.70 was presented in *both* the Rich and Lean block). The key finding was that participants were more likely to gamble on these common trials when presented in the Rich context. OpAL* reproduces this pattern due to adaptive $\rho$ increasing DA levels and risk-tasking in the Rich block. Error bars show standard error of the mean. Parameters: $\alpha = 1., T = 10, \beta = 0.5, \phi = 1.0, k = 0.09$. Figure C modified from **Frydman and Jin, 2021**.

softmax gain leads to reduced exploitation, and thus drives both groups toward random selection. Thus the pattern of choice data is aligned with OpAL* but not with RSRL, or with classical models in which D2 activity inhibits choice altogether. These opposing predictions result from the *architecture* of OpAL* inspired by the biology– including opponency, Hebbian learning, and dynamic DA – rather than specific parameter values. Furthermore, OpAL* also captures the predicted relative activation of D2-expressing cells during the choice period following losses due to changing DA levels ($\beta_n(t)$) and the learned cost of the gamble ($N(t)$), in line with Zalocusky's photometry data.

## DA drug effects on risky decision-making and individual differences therein

We next focus on a human risky decision-making paradigm manipulating DA levels (*Rutledge et al., 2015*). Participants were presented with interleaving trials of gain gambles (certain gain vs. potential greater gain or 0), loss gambles (certain loss vs. potential greater loss or 0), and mixed gambles (certain no reward vs. potential gain or potential loss). All gambles were successful with 50% probability. The study tested the effects of levodopa (L-DOPA), a drug which boosts dopamine release, on risky decision-making. The main impact of L-DOPA was to selectively amplify gambling on gain (but not loss or mixed) trials (*Figure 11A*, left). This study also found that individual differences in this impact of drug on gambling correlated with effective drug dosage (*Figure 11B*, left). The authors reported that the risk-seeking behavior with DA drugs was best described in terms of changes in a Pavlovian approach parameter. Here, we wished to see if the mechanisms introduced above within OpAL* with endogenous changes in dopaminergic state could replicate the pattern of results, thereby providing a normative interpretation.

We simulated 300 trials (100 gain gambles, 100 loss gambles, and 100 mixed gambles, randomly interleaved, as described in *Rutledge et al., 2015*). The probability of gambling was determined as described above in the normative risky choice section, with gambles accepted as the benefits outweigh the costs relative to the ST. $G$ and $N$ actor values were explicitly set on each trial according to the instructed gamble and encoded relative to the certain option as in the section 'OpAL* adaptively modulates risk-taking.' This reduced the free parameters of OpAL* (no annealing or actor learning rate needed) while retaining its core features of DA reweighting the contributions of opponent representations during choice according to context.

While values and probabilities were explicitly instructed in the experiment, subjects nevertheless experienced the outcomes of each gamble. The OpAL* model assumes that they thus track the average value of offers across trials, such that a gain trial would elicit a positive dopamine deflection, given that its expected value is larger than that for mixed and loss trials. (As the authors note in discussing their findings, 'In this task design, even the worst gain trial is better than the average trial and so likely inspires dopamine release.') We thus modeled the relative DA-state $\rho$ proportional to the expected value of the current gamble offer, approximating how 'rich' or 'lean' the current offer was relative to all offers in the game. (See Supplemental note 8 in Appendix 2). (We formulate $\rho$ proportional to value here, to be consistent with simulations in the above sections, but very similar results were obtained in a separate set of simulations in which $\rho$ was modulated by RPE.)

$$\rho(t) = .5 \times (\text{certain outcome}) + .5 \times EV(\text{gamble}) \tag{23}$$

To model L-DOPA, we hypothesized that it would boost positive RPEs via enhancement of evoked (phasic) DA release, as observed in vivo across species (*Voon et al., 2010*; *Pessiglione et al., 2006*; *Qi et al., 2016*; *Harun et al., 2016*). We assumed that L-DOPA amplified endogenous phasic release, which occurs when offers are better than usual (positive RPE). The effect dosage level was represented by $d$ when the gamble had a positive value, as shown below.

$$\rho'(t) \quad = \rho(t)(1 + d) \tag{24}$$

$$d \quad \geq 0 \tag{25}$$

As hypothesized, OpAL* captured the selective effects of L-DOPA on gambling in gain trials. It also captured the overall proportion of gambles chosen for different trial types (*Figure 11A*), as well as the correlation between effective dosage and difference in gambling on and off drug (*Figure 11B*). (See Supplemental note 9 in Appendix 2). Furthermore, the Pavlovian model presented in *Rutledge*

*et al., 2015* would predict that gambling would occur for positive RPEs even if the potential benefit of the gamble was not as high as the sure thing; OpAL* would only predict increased gambling if the benefits are greater than the sure thing.

Here, we have extended OpAL to account for risky decision-making by dynamically changing dopamine levels at choice proportional to the value of the current state/gamble offer. This accounted for findings of increase attractiveness of high-value risky options with the administration of L-DOPA (*Figure 11A*). The model also accounted for individual differences of risk due to effective L-DOPA dosages (*Figure 11B*). As highlighted in the previous section, these effects can normatively be explained as behavioral changes reflecting changes of inferred richness of current state. These results also suggest that individual differences in risk preference and sensitivity may be due to learned statistics of the world, casting these individual differences as deriving from an adaptive mechanism to an animal's or human's experience niche.

## Risky decisions are sensitive to environmental richness: Concordance with efficient coding models of economic choice

Thus far we have focused on data that are informative about the biological mechanisms (striatal opponency and DA modulation thereof) by which OpAL* supports adaptive behavior. But OpAL* also makes straightforward economic choice predictions that do not require biological manipulations. In particular, one way of conceptualizing OpAL* is that it serves as an efficient coding mechanism by amplifying the actor that maximally discriminates between reward values in the current environment. If choice patterns concord with this scheme, one should be able to manipulate the environment and influence choice patterns. For example, consider a gamble in which the benefits outweighs the costs. OpAL* predicts that decision makers should more consistently opt to take this gamble when it is presented in the context of a rich environment. Indeed, this is precisely what was found by economist researchers, who also considered such patterns to be indicative of efficient coding (*Frydman and Jin, 2021*).

In this study, participants were presented with a series of trials where they selected between a gamble with a varying magnitude X with 50% probability and a certain option with varying magnitude C. The task featured two conditions, which we refer to as Rich and Lean. The range (minimum and maximum) of Xs and Cs were equated across the two conditions, but high-magnitude Xs and Cs were more frequent in the Rich environment, whereas low-magnitude Xs and Cs were more frequent in the Lean environment. The distribution of C was set to 0.5*X so that the expected values of the risky lottery and certain option were on average equated. Critically, there were a few carefully selected 'common trials' that repeated the exact same high payoff gambles (with identical X and C) across blocks (*Figure 11C*). The authors reported that participants were more likely to gamble on common trials in Rich environments than Lean environments. This is in line with their economic efficient-coding model, which predicts subjects allocate more resources to accurately perceive higher payoffs in the Rich condition where higher payoffs are more frequent (and therefore gamble more on common trials which are high payoff).

To simulate this dataset with OpAL* (*Figure 11D*), we assumed that the critic state value would reflect the statistics of the environment. We first set the baseline expectation to reflect the expected value of a uniform prior over the gamble magnitudes and certain magnitudes in the experiment, which serves as a prior for environment richness. $\rho$ was modulated by the learned average gamble offer in the environment relative to this baseline. (See Supplemental note 10 in Appendix 2). As in our earlier risky choice simulations, gambles were encoded relative to the certain option and G/N values were explicitly set according to the instructed gamble, omitting the need again for annealing and actor learning rate while preserving the core dynamics of the full OpAL*. As found empirically and in the authors' efficient coding model (*Frydman and Jin, 2021*), OpAL* predicts increased gambling on common trials in the Rich block relative to the Lean block. According to OpAL*, this result reflects adaptively modulated DA levels in the Rich environment, which emphasized the benefits of the gamble during decision-making. As will be discussed below, OpAL*'s amplification of one striatal subpopulation over another itself can be considered a form of efficient coding, offering a direct mechanistic explanation for recent findings in economic theory. Finally, note that such findings could not be captured by an alternative model in which risky choice is driven by surprise or novelty. Note that for both rich and lean blocks, common trials had larger than usual magnitudes of payoffs. While these payoffs deviated

from expectation to a larger degree in the lean block, this should produce a larger RPE (and presumably phasic dopamine signal). Given that increased DA in traditional RL models promotes exploitation (*Humphries et al., 2012*), this account (like the RSRL model above) would predict the opposite pattern than that seen empirically, in this case driving more risky choices in the lean block.

## Discussion

Taken together, our simulations provide a normative account for opponency within the BG and its modulation by DA. In particular, we suggest that nonlinear Hebbian mechanisms give rise to convexity in the learned D1 and D2 actor weights at different ends of the reward spectrum, which can be differentially leveraged to adapt decision-making. To do so, OpAL* alters its dopaminergic state as a function of environmental richness, so as to best discern between the costs or benefits of available options. Conjecturing that such a mechanism is most profitable when the reward statistics of the environment are unknown, we posited and found that the online adaptation robustly outperforms traditional RL and alternative BG models across environment types when sampling across a wide range of plausible parameters. These advantages grow monotonically with the complexity of the environment (number of alternative actions to choose from). Moreover, the unity of all three key features of OpAL* (opponency, three-factor Hebbian nonlinearity, and dynamic DA modulation) offered particularly unique advantages in sparse reward environments, mitigating against a particularly pernicious explore exploit dilemma that arises in such environments by amplifying the action gap in reward sparse environments. Finally, we showed how such a mechanism can adapt risky decision-making according to environmental richness, capturing the impact of DA manipulations and individual differences thereof.

This article intersects with theoretical (*Niv et al., 2007*) and empirical work (*Hamid et al., 2016*; *Mohebi et al., 2019*) showing that changes in dopaminergic states locally within striatum reflect reward expectations and impact motivation and vigor. However, this body of literature does not consider how increases or decreases of dopamine affect the decision itself, only its latency or speed. Instead, OpAL/OpAL* can capture both shifts in vigor and cost–benefit choice as seen empirically with drug manipulations across species (*Cousins et al., 1996*; *Salamone et al., 2005*; *Treadway et al., 2012*; *Westbrook et al., 2020*) and more precise optogenetic manipulations of DA and activity of D1 and D2 MSNs (*Doi et al., 2020*; *Bolkan et al., 2021*; *Zalocusky et al., 2016*; *Tai et al., 2012*; *Yartsev et al., 2018*). Notably, OpAL* suggests that in sparse reward environments, it is *adaptive* to lower dopaminergic levels and not merely avoiding action altogether (as in classical notions of the direct indirect pathways). Rather, lower dopamine helps to choose actions that minimize cost (by discriminating between D2 MSN populations). In physical effort decision tasks, DA depletion does not simply induce more noise or reduced effort overall, but selectively promotes actions that minimize effort when the benefits of exerting effort are relatively low (*Cousins et al., 1996*). For example, while a healthy rat will choose to climb a barrier to obtain four pellets instead of selecting two pellets that do not require physical effort, a dopamine-depleted animal will opt for the two-pellet option. However, in the absence of the two-pellet option, both healthy and dopamine-depleted animals will select to climb the barrier to collect their reward. While OpAL* naturally accounts for such findings, other models often suggest that lowered DA levels would simply produce more randomness and imprecision, as captured by a reduced softmax gain (*FitzGerald et al., 2015*; *Cinotti et al., 2019*; *Eisenegger et al., 2014*; *Lee et al., 2015*). Importantly, empirical evidence for this reduced gain account in low DA situations focused exclusively on reward rich situations (i.e., available options were likely to be rewarding); in these cases, OpAL* also predicts more noise. But as noted above, low dopaminergic states may not always be maladaptive. Indeed, they may be useful in environments with sparse rewards, allowing an agent to adaptively navigate exploration and exploitation and to avoid the most costly options.

The work described here builds off a preliminary suggestion in *Collins and Frank, 2014* that opponency in OpAL confers advantages over standard RL models across rich environments and lean environments. In particular, when parameters were optimized for each model, the optimal parameters for standard RL diverged across environments, whereas OpAL could maximize rewards across environments with a single set of parameters; biological agents have indeed demonstrated similar learning speeds between lean and rich environments, demonstrating such cross-environment flexibility (*Hamid et al., 2016*). However, this previous work applied to a balanced OpAL model and did not consider how an agent might adaptively modulate dopaminergic state to differentially weigh costs vs. benefits

of alternative decisions. Here, we showed that such advantages are robust across a wide range of parameters, that they are amplified in OpAL* by leveraging dynamic DA modulation, and that they grow with the complexity of the environment (number of alternative actions). Importantly, such benefits of OpAL* capitalize on the nonlinear and opponency convexity induced by Hebbian plasticity within D1 and D2 pathways (*Figure 2*).

These findings contrast with Q-learning agents and with other theoretical models of striatal opponency which omit the Hebbian term but leverage alternate nonlinearities so that D1 and D2 weights converge to the veridical benefits and costs of an action (*Möller and Bogacz, 2019*). This result is somewhat counterintuitive: if an agent knows the actual costs and benefits of each action, they could simply choose the one that maximizes net return. Critically, however, the agents that based choice on expected values exhibited misestimation errors in those values due to delayed convergence, especially in environments with sparse reward and the agent has to select between multiple actions. As a result, the agent stochastically switches between actions until each of their benefits and costs (or Q values) are known, leading to reduced 'action-gaps.' Meanwhile, the actors of OpAL* discriminate between the optimal and suboptimal actions well before the critic converges, and thus can rapidly optimize the policy. This difference is related to the recent predominance of policy gradient methods over value-based algorithms in the deep RL literature, particularly in environments with sparse reward and large action spaces. Work has also shown that striatal signals conform more with policy update methods rather than action values (*Li and Daw, 2011*). Notably, *G. Bellemare et al., 2015* showed that large action gaps are helpful for mitigating against estimation errors in the critic in deep RL settings, implying that an OpAL* like approach might be useful in those contexts as well (but this remains to be tested).

Clearly there is also a normative component to an agent's ability to properly estimate the true expected values (or costs and benefits) of its actions, which would be needed to more robustly exhibit transitive preferences between options that it has never confronted together. Nevertheless, empirically, violations of such transitive preferences are observed specifically when choosing between actions that had been experienced in the context of expected loss versus expected gains (*Gold et al., 2012*; *Palminteri et al., 2015*; *Geana et al., 2022*). Our model captured this pattern because its policy had been optimized to prefer a frequent loss avoider in the loss context, and to avoid an infrequent winner in a gain context (thus our account is similar in spirit to the model proposed by *Palminteri et al., 2015*, who used a Q-learning framework but wherein values are learned not in absolute terms, but relative to that of the other options). Finally, as noted above, while we do not leverage this mechanism, in principle an agent could combine the advantages of both frameworks by using OpAL* actors to optimize the policy during learning but then use the critic's Q values, once converged to select actions based on expected values. Such a pattern would be expected in hybrid Q-learning/actor-critic frameworks (*Gold et al., 2012*; *Geana et al., 2022*), and would naturally arise in an OpAL* agent that allowed its actor weights to decay (i.e., without annealing). Future research should thus test whether transitive violations such as those described by *Palminteri et al., 2015* would continue to be observed had the participants been given more extended training.

It is notable that the advantages exhibited by OpAL* depended on the nonlinear Hebbian mechanism. While the Hebbian term was originally motivated by the biology of three-factor plasticity as implemented in the neural network version, it is also needed to capture findings in which D2 MSNs become increasingly potentiated as a result of pathological DA depletion or DA blockade, leading to aberrant behavioral learning and progression of Parkinsonism (*Wiecki et al., 2009*; *Beeler et al., 2012*). Ironically, it is this same Hebbian-induced nonlinearity that affords adaptive performance in OpAL* when DA is not depleted or manipulated exogenously. As shown in the 'Mechanism' section, nonlinear accumulation allows the actor weight updates to be more sensitive to the probabilistic history of outcomes, preventing the agent from switching back to suboptimal actions. Finally, this adaptive role for activity-dependent Hebbian plasticity beyond standard learning algorithms is complementary to recent observations that such mechanisms can be leveraged to improve beyond gradient descent in neural networks (*Scott and Frank, 2021*). While the computations are leveraged for different purposes (roughly, choice vs. credit assignment) and in different architectures, both findings accord with the notion that mechanisms typically thought to merely approximate adaptive functions inspired by artificial intelligence may in fact confer benefits for biological agents.

Lastly, while many studies have documented *that* DA manipulations affect risky and effort-based decision-making across species, our results offer a normative explanation for such findings. In this perspective, the brain treats increases or decreases in dopamine as signaling presence in a richer or leaner state. Changes in behavior reflect an adaption to this perceived, artificial environmental change. Hence, a dopamine-depleted animal (or increased activity of D2 MSNs in *Zalocusky et al., 2016*) would focus on costs of actions, whereas dopamine increases would increase attractiveness of risky actions (*Rutledge et al., 2015*). We reasoned that the well-known impact of exogenous DA modulation on risky decision-making (*St Onge and Floresco, 2009*; *Zalocusky et al., 2016*; *Rutledge et al., 2015*) may be a by-product of this endogenous adaptive mechanism, showing that OpAL* can be used to modulate appropriately when it is worth taking a risk (*Figure 7*). We then demonstrated how behavioral effects of D2-receptor activity and manipulation (*Zalocusky et al., 2016*) reflect unique predictions of OpAL*, including outcome-dependent risk-avoidance paired with increase of D2 activity following a loss (*Figure 10A–D*). In conjunction, optogenetic stimulation of D2-expressing neurons induced decrease in risky choice in risk-seeking rodents in line with OpAL* predictions (*Figure 10E and F*). Furthermore, we showed that OpAL* can be used to capture changes in risk-taking by dopamine-enhancing medication in healthy human participants (*Figure 11A and B*). Our simulations highlighted how individual changes in risk preference may emerge from OpAL*'s adaptive mechanism. While some studies have shown that in unique circumstances increased dopamine may result in preference for a low-risk but low-reward option (*Mikhael and Gershman, 2022*; *St Onge et al., 2010*), these results rely on sequential effects but nonetheless they may be explainable by OpAL*'s sensitivity to environmental reward statistics. Furthermore, we focused on adaptive decision-making on the time scale of a single task in this article, and it is plausible that such an adaptive mechanism may account for larger individual differences across longer time horizons. For example, increased risk-taking has been well documented in adolescents and some evidence suggests that dopaminergic levels may peak during adolescents, attributing to this trend (see *Wahlstrom et al., 2010* for a full review). Speculatively, this may itself be an adaptive mechanism, where higher DA may allow more emphasis on potential benefits of risky but developmentally beneficial actions, such as exploring outside of parent's home to find a mate.

OpAL*'s separation and selective amplification of $G$ and $N$ actors also is reminiscent of efficient coding principles in sensory processing, which theorizes that neurons maximize information capacity by minimizing redundancy in neural representations (*Barlow, 2012*; *Laughlin, 1981*; *Chalk et al., 2018*). Efficient coding also suggests that resources should be reallocated according to features in an environment which occur more frequently (*Simoncelli and Olshausen, 2001*). In OpAL*, positive prediction errors are more abundant than negative in reward-rich environments and the $G$ actor strengthens disproportionately as this asymmetry grows. Conversely, negative prediction errors are more frequent in reward-lean environments and the $N$ actor specializes in this asymmetry. Changes in dopaminergic state, which modifies the contribution of $G$ and $N$ actors, therefore reallocate decision-making resources according to the relative frequency of positive and negative prediction errors in the environment. Recent behavioral work has applied an efficient coding framework to risky choice paradigms, showing participants are riskier in environments which have an increased frequency of large gamble payoffs (*Frydman and Jin, 2021*). Our model provides a mechanistic account of such findings that generalizes to broader behavioral implications. Moreover, while the authors did not test this pattern, OpAL* predicts that if common trials were administered to include unfavorable gambles (gambles whose expected values are less than a certain option), people would more reliably select the certain outcome in the lean environment.

## Limitations and future directions

A limitation of the DA modulation mechanism is that its performance advantages depend on relatively accurate estimates of environmental richness. Indeed, performance can suffer with incorrect estimation of the environment richness (*Appendix 1—figure 1*). Thus, it is essential in OpAL* that DA modulation is dynamic across trials so as to reflect sufficient reward history before modulating opponency. As such, while we systematically characterized the advantage of dynamic DA modulation in OpAL* over the balanced OpAL model ($\rho = 0$) across environments, this advantage should hold over any OpAL model with a fixed asymmetry (see *Figure 2*). For robust advantages, the critic estimation of environmental richness must be relatively confident before modulating DA. In the simulations

presented, we utilized a Bayesian meta-critic to explicitly track such uncertainty, and only increasing or decreasing DA when the estimate was sufficiently confident. Interestingly, this mechanism provides an intermediate strategy between directed and random exploration (*Wilson et al., 2014*), but at the level of actor (rather than action) selection. In OpAL*, such a strategy amounts to random exploration across both actors until the critic uncertainty is sufficiently reduced, at which point OpAL* exploits the actor most specialized to the task. Future directions will investigate how this strategy may itself be adapted as a function of the environment statistics and may offer potential predictions for understanding individual differences and/or clinical conditions. For example, given inappropriate dopaminergic state is most detrimental to sparse reward environments, an agent which prioritizes avoidance of costs such as those prevalent in sparse reward environments (such as in OCD or in early life stress) may benefit from more caution before changing dopaminergic state (i.e., have a higher threshold for DA modulation and exploiting knowledge) or take longer to integrate information to increase precision of estimates (i.e., lower learning rate).

There are several future directions to this work. For example, while OpAL* optimizes a single DA signal toward the actor most specialized to rich or lean environments, recent work also suggests that DA signals are not uniform across striatum (*Hamid et al., 2021*). Indeed, this work showed that DA signals can be tailored to striatal subregions specialized for a given task, keeping with a 'mixture of experts' model to support credit assignment. Future work should thus consider how the DA signals can be simultaneously adapted to the benefits and costs of alternative actions within subregions that are most suited to govern behavior. Moreover, while we addressed the impact of complexity within the action space, an alternative notion of complexity and sparsity yet to be explored is the length of sequential actions needed to achieve reward. Increasing the distance from initial choice to reward, a problem faced by modern deep RL algorithms (*Hare, 2019*), may also benefit from integrating OpAL*-like opponency and choice modulation into larger architectures, given the improved action gaps that facilitate performance in such settings (*G. Bellemare et al., 2015*). Finally, while our work focuses on asymmetries afforded in the choice function, DA manipulations can also induce asymmetries in learning rates from positive and negative RPEs (*Frank et al., 2007a*; *Niv et al., 2012*; *Collins and Frank, 2014*), which can, under some circumstances, be dissociated from choice effects (*Collins and Frank, 2014*). However, it is certainly possible that asymmetries in learning rates can also be optimized as a function of the environment. Indeed, larger learning rates for positive than negative RPEs are beneficial in lean environments (and vice versa) by amplifying the less frequent signal (Cazé and *Cazé and van der Meer, 2013*). Such effects are not mutually exclusive with those described here, but note that they do not address the issue highlighted above with respect to exploration exploitation dilemmas that arise in lean environments, and do not capture the various findings (reviewed above) in which DA manipulations affect performance and choice in the absence of outcomes.

## Materials and methods

### Parameter grid search

For OpAL* variants, we ran a grid sweep over a parameter space with $\alpha_c \in .025, .05, .1$, $\alpha_G = \alpha_N \in [.05, 1]$ with step size of 0.05 and $\beta \in [1, 10]$ with step size of 0.5. To equate the model complexity, the annealing parameter ($T = 10$), the strength of modulation ($k = 20$), and the confidence needed before modulation ($\phi = 1.0$) were fixed to the specified values across models. These were determined by coarser grid searches of the parameter space for reasonable performance of control models. For each parameter combination, we matched the starting random seed for three models – OpAL*, OpAL* with $\rho = 0$ (OpAL+), and OpAL* with no three-factor Hebbian term (No Hebb). For each parameter setting for each model type, we calculated the average softmax probability of selecting the best option (80% in rich environments or 30% in lean environments) across 1000 simulations for 1000 trials. We then took the AUC of this averaged learning curve for different time horizons (100, 250, 500, and 1000 trials) and took the difference between the AUCs of OpAL* and OpAL* with $\rho = 0$ or OpAL* No Hebb of matched parameters. We conducted a one-sample *t*-test on these differences, where a difference of zero was the null hypothesis.

We conducted the same set of analyses with the learning curves for the actual rewards received and received mirror results. We therefore only report the analysis according to the probability of selecting an action, which is a finer grain measure of average performance.

For Q-learner, we ran a grid sweep over a parameter space with learning rate $\alpha \in [.05, 1]$ with step size of 0.05 and softmax temperature $\beta \in [2, 100]$ with step size of 2.

For UCB, the exploration parameter $c$ (see next section) was searched over the space $c \in [0, 2]$ with increments of .01.

## Upper Confidence Bound

To implement UCB (*Sutton and Barto, 2018*; *Auer et al., 2002*), rather than calculating a Q-value incrementally, we used the sample mean of receiving reward for each action. The algorithm began by selecting each action once in a random order, thus amounting to full information for one trial of each action. The agent then greedily selected the action $a$ with the largest mean combined with an exploration factor, determined by the hyperparameter $c$. In order to calculated the choice $A_t$ for each trial, $t$, we used the following:

$$N_t(a) = \sum_{i=0}^{t-1} \mathbb{1}(A_i == a) \qquad \text{Number of times action a was selected}$$

$$Q_t(a) = \frac{\sum_{i=0}^{t-1} \mathbb{1}(A_i == a)R_i}{N_i(a)}$$

$$A_t = \arg\max_a \left[ Q_t(a) + c\sqrt{\frac{ln(t)}{N_t(a)}} \right]$$

During optimization, the hyperparameter, $c$, was optimized across environments using a grid search where $c \in [0, 2]$ with increments of 0.01.

## Möller and Bogacz 2019 model

The Möller and Bogacz model (*Möller and Bogacz, 2019*) offers another computational account of how benefits and costs may be encoded in the D1/D2 striatal subpopulations. First note that this model defines benefits and costs as the *absolute magnitude* of positive and negative outcome for each action. In contrast, benefits and costs as represented in OpAL/OpAL* are relative metrics (accordingly, for gamble simulations, an outcome of 0 is encoded as a cost relative to the sure thing, similar to other models of reference dependence; see also simulations of *Palminteri et al., 2015* above in the case of full information). Second, both OpAL and Möller and Bogacz's model have nonlinearities in the learning rule (otherwise, as seen in our No Hebb model, the two pathways are redundant). However, rather than using Hebbian plasticity, Möller and Bogacz transform the prediction error itself (such that the impact of negative prediction errors is smaller in the G actor, and vice versa, parameterized by $\epsilon$) and impose a weak decay ($\lambda$), as expressed below. Similar to OpAL, dopamine levels, D, modulate the contribution of D1/D2 to choice.

$$\Delta G(s, a) \quad = \alpha f_\epsilon(\delta) - \lambda G(s, a) \tag{26}$$

$$\Delta N(s, a) \quad = \alpha f_{-\epsilon}(\delta) - \lambda N(s, a) \tag{27}$$

$$f_\epsilon \quad = \begin{cases} \delta & \text{for} \quad \delta > 0 \\ \epsilon\delta & \text{for} \quad \delta < 0 \end{cases} \tag{28}$$

$$Q(s, a) = DG(s, a) - (1 - D)N(s, a) \tag{29}$$

$$D = .5 \tag{30}$$

This learning rule allows the G and N weights to converge to the expected payoffs and costs of alternative actions with sufficient learning. However, as noted above and shown in *Figure 8—figure supplement 3*, just like in Q-learning, convergence can be impeded in lean environments with stochastic action selection, leading to slowed acquisition of an effective policy.

To select between actions, we used a softmax policy. While *Möller and Bogacz, 2019* explicitly do not use a softmax function in their simulations, they did so only because they were simulating behaviors in which an action may not be selected at all (i.e., they did not subject their agent to choose between different actions). In contrast, for all of our experiments, our agents must select an action

each trial. We therefore generate a choice as follows using the softmax function by using the value of the action, $V(a)$. **Figure 8—figure supplement 3** uses parameters reported in Figure 5c of **Möller and Bogacz, 2019**, where the authors demonstrate that G and N weights should appropriately converge in a simulation analogous to a one-armed bandit with reward probability of 50%. We selected beta using a grid search from 10 to 100 in steps of 10 and found comparable results.

We also explored optimizing the Möller et al. model over all four free parameters – $\alpha, \epsilon, \lambda, \beta$ – for cross-environment performance. Parameters were found using scipy.optimize.differential_evolution routine, optimizing for the best average softmax probability selecting the best option over 1000 simulations across rich and lean. For convergence to expected payoffs and costs, **Möller and Bogacz, 2019** demonstrate approximate constraints and relationships that the parameters should adhere to, specifically the decay parameter ($\lambda$) must be close to 0 and smaller than the learning rate and the nonlinearity parameter ($\epsilon$) must be approximately 1. We found for optimization over shorter time horizons (e.g., less than 250 trials), the optimized value for epsilon was closer to zero than 1 (<0.1), but for 500 trials its estimated value was more appropriate (>0.9) though the relative relationship of lambda and alpha did not hold. Nonetheless, as in Q-learning, the action gap was consistently divergent between the rich and lean environments in these different iterations.

$$V(a) = \tfrac{1}{2}(G - N) \tag{31}$$

$$p(a) = \frac{e^{\beta V(a)}}{\sum_{i \in A} e^{\beta V(i)}} \tag{32}$$

## Code
Code repository available at https://github.com/amjaskir/opal-star, (copy archived at **Jaskir, 2023**).

## Acknowledgements

AJ was partly supported by NIMH training grant T32MH115895 (PIs: Frank, Badre, Moore). The project was also supported by NIMH R01 MH084840-08A1 and NIMH P50 MH119467-01. Computing hardware was supported by NIH Office of the Director grant S10OD025181. We thank Pete Hitchcock and Lucas Lehnert for comments.

## Additional information

### Competing interests
Michael J Frank: Senior editor, eLife. The other author declares that no competing interests exist.

### Funding

| Funder | Grant reference number | Author |
| --- | --- | --- |
| National Institute of Mental Health | P50MH119467 | Michael J Frank |
| National Institute of Mental Health | R01 MH084840 | Michael J Frank |
| National Institutes of Health | S10OD025181 | Michael J Frank |

The funders had no role in study design, data collection and interpretation, or the decision to submit the work for publication.

### Author contributions
Alana Jaskir, Conceptualization, Software, Formal analysis, Validation, Investigation, Visualization, Writing – original draft, Writing – review and editing; Michael J Frank, Conceptualization, Formal analysis, Supervision, Funding acquisition, Validation, Project administration, Writing – review and editing

### Author ORCIDs
Alana Jaskir http://orcid.org/0000-0003-3538-0827

Michael J Frank http://orcid.org/0000-0001-8451-0523

**Decision letter and Author response**
Decision letter https://doi.org/10.7554/eLife.85107.sa1
Author response https://doi.org/10.7554/eLife.85107.sa2

## Additional files

### Supplementary files
• MDAR checklist

### Data availability
The current manuscript is a computational study, so no data have been generated for this manuscript. Simulation code is available on the authors' GitHub repositories https://github.com/amjaskir/opal-star, (copy archived at *Jaskir, 2023*).

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

# Appendix 1

## Incorrect modulation impairs performance

As noted in the main text, it is important that the critic estimate of environmental richness is reasonably accurate (on the correct side of 0.5) for OpAL* to confer advantages. Indeed, pathological behavior arises if DA states are altered in opposing direction to environmental richness. In *Appendix 1—figure 1*, we see the effect of flipping the sign of OpAL*'s calculation of dopaminergic state (*Equation 14*). For this demonstration, if the critic of OpAL* estimated that it was in a rich environment (positive value of $\rho$, high dopaminergic state), it would emphasize the N instead of G actor (as if it were in a lean environment). We see that the lean environment shows high sensitivity to incorrect modulation. The rich environment shows greater robustness but nonetheless has decreased performance in comparison to the standard simulations. This result confirms that the direction of modulation in OpAL* is important, and moreover that it is particularly important to have lower DA in lean environments.

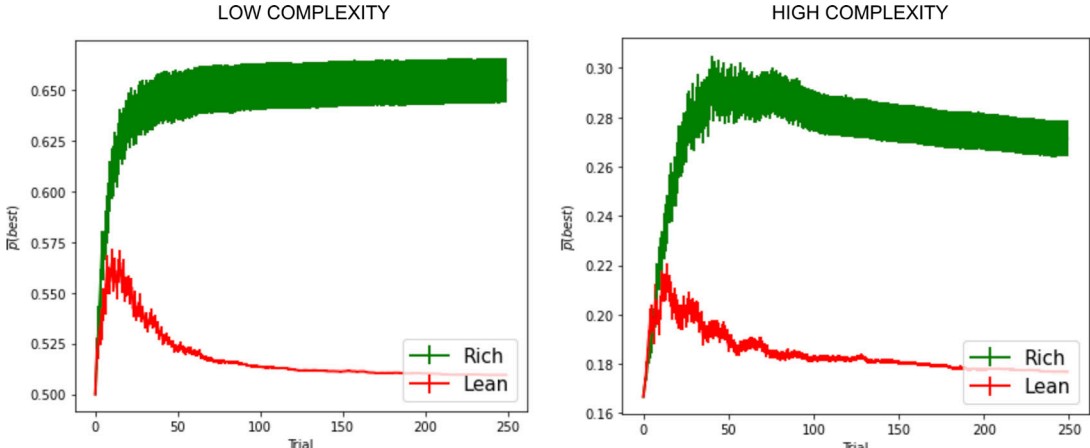

**Appendix 1—figure 1.** Effects of dopaminergic states that inaccurately reflect environmental richness. Parameters optimized the average area under the curve (AUC) across both rich *and* lean environments according to the grid search. See 'Parameter grid' search for details.

## Comparison to softmax temperature modulation

As noted in the main text, OpAL* confers larger benefits in lean environments, in part by mitigating against an exploration/exploitation dilemma. In particular, during early learning, OpAL* relies on both actors equally and thereby distributes its policy more randomly, but after it estimates the richness of the environment, it exploits the more specialized actor. To evaluate whether similar benefits could be mimicked by simply increasing softmax gain over trials (transitioning from exploration to exploitation), we considered an OpAL* variant that symmetrically increased the softmax temperature according equally across the G and the N actor. As the richness (or leanness) of the environment grew, the agent would progressively exploit both actors equally using the same Bayesian meta-critic as in OpAL*.

$$\beta_g = \beta \max(0, 1 + |\rho_t|) \tag{33}$$

$$\beta_n = \beta \max(0, 1 + |\rho_t|) \tag{34}$$

Given the difference in exploration–exploitation demands across rich and lean environments, we compared the average AUCs of OpAL* and beta-modulation (B-Mod). Overall we found that OpAL* exhibited improved maximal cross-environment robustness and specifically improved maximal performance in the rich environment. Thus, global changes in explore–exploit the softmax temperature alone are insufficient to capture the full performance benefit in lean environments induced by dopaminergic modulation in OpAL*, which capitalizes on specialized learned representations across actors.

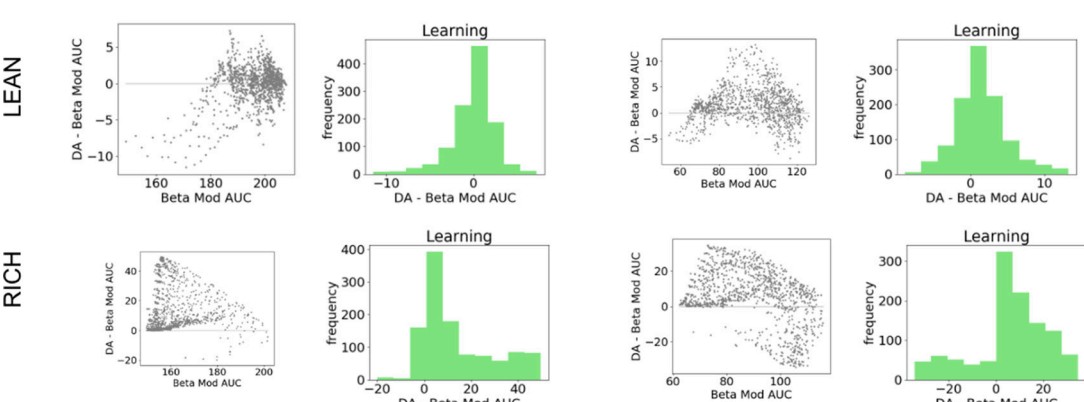

**Appendix 1—figure 2.** Comparison of OpAL* to dynamic modulation of softmax temperature (bmod). Figure shows average areas under the curve (AUCs) of models for fixed parameter in both lean and rich environments for varying complexity. Histograms show differences in AUC of paired parameters. Across horizons and complexity levels, OpAL* significantly outperformed Bmod in the rich environment ($p's < 1e^-10$). As time horizons increase, Bmod outperforms begins to outperform OpAL* in lower complexity levels for lean environments (levels 2 and 3 for 500 trials, all complexity levels for 1000 trials [$p < 1e^-7$]), though with smaller effect sizes than OpAL*'s relative performance in rich.

## Addressing *Möller and Bogacz, 2019*

We incorporated normalization and weight decay for the actors to address weaknesses of the original OpAL model raised by *Möller and Bogacz, 2019*. The (valid) critique outlined by *Möller and Bogacz, 2019* is that its three-factor Hebbian update gives rise to unstable actor dynamics, specifically after the critic converges. They demonstrated that when OpAL is sequentially presented with a reward of 2 followed by a cost of –1, the dynamics of G and N rapidly converge to 0 (*Appendix 1—figure 3*, left). As described in their text (*Equations 39–41*), stable oscillations in reward prediction errors cause G and N weights to decay towards zero. This decay is indeed a characteristic of the standard OpAL model without annealing: once the critic begins to converge, RPEs for disconfirmatory outcomes induce larger changes in the weights due to higher order terms (see expanded weight update *Equation 22* for intuition). (Arguably, this decay is akin to an advantage-learning action value curve, whereby once the critic begins to converge, the 'advantage' of the option [difference between the action value and the average value of the environment] decreases overtime [*Dayan and Balleine, 2002*]. While the current section addresses the decay via annealing, we do not view the existence of decay itself to be necessarily problematic. First, in neural network versions of our and other BG models, striatal action selection contributes primarily to early learning; once a policy is repeated sufficiently [e.g., when convergence is more likely], the cortex can directly select an action in a stimulus–response fashion [*Frank and Claus, 2006*; *Ratcliff and Frank, 2012*; see also *Ashby et al., 2007*]. Alternatively, although we do not explore this here, it is also possible that the decay is a feature rather than a bug: since the weights decline once the critic has converged, the decay itself is an indicator that the learned Q value from the critic is well estimated, and the agent could potentially use that Q value directly for action selection itself, using a hybrid Q learner/actor-critic model. In that case, the actor weights would dominate the policy early during learning but once they decay the hybrid action selection mechanism would agent could rely on [the now converged] Q values. Such hybrid models have been used to simulate choice preferences in tasks similar to that of *Palminteri et al., 2015*; *Gold et al., 2012*; *Geana et al., 2022*.)

The rapid decay evident in *Appendix 1—figure 3*, left, was constructed to highlight a particularly pernicious example of this issue. The following simulations suggest that the introduction of larger reward magnitudes, rather than the oscillating PEs, has driven such expedited instability. Larger reward magnitudes yield larger reward prediction error signals, which in turn yield larger G/N values as evident by *Equations 3 and 4*, which, through the Hebbian positive feedback cycle, further increase effective learning rate. One simple correction is to simply rescale and shrink the magnitudes by some constant ($0 < c < 1$); this slows decay in this example (simulations not shown). Note also that *Möller and Bogacz, 2019* simulations used a relatively large critic learning rate (0.3), which speeds

convergence and exacerbates these effects. Adaptive behavior in OpAL* involves a relatively low critic learning rate, as highlighted above (due to contributions of the higher order terms in the G/N recursive updates that drive decay). Simply decreasing the critic learning rate (e.g., 0.05) thus also dampens the decay (not shown).

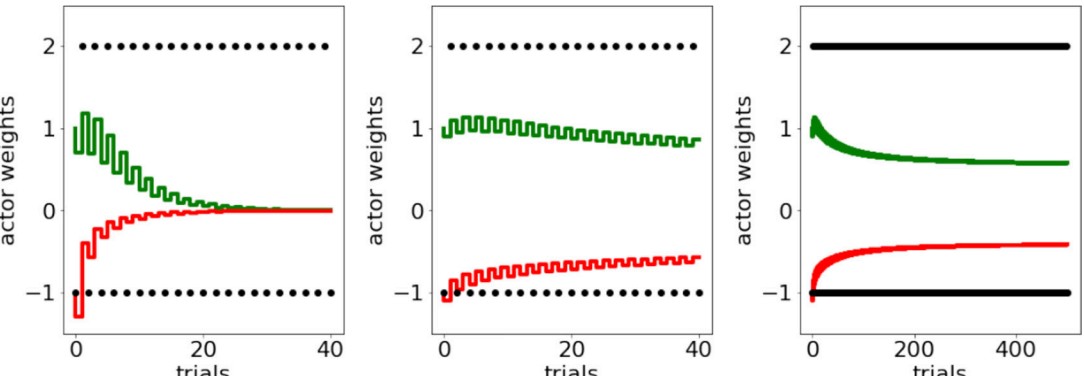

**Appendix 1—figure 3.** Scenario presented in *Möller and Bogacz, 2019*, where oscillating payout of +2 and loss of –1 induces accelerated decay of G and N weights to zero, caused both by large reward magnitudes and quick convergence of the critic. To protect against such decay, we introduce normalization of reward prediction errors and learning rate annealing (middle and right).

Nevertheless, we sought to more robustly address these issues because the above modifications only prolong the decay. We introduced two modifications in OpAL* to address these concerns. First, prediction errors used to update G and N actors (*Equations 18 and 19*) are normalized by the range of known reward magnitudes in the environment (*Equation 21*). Importantly, OpAL* is not provided any reward statistics beyond the range of reward feedback, and in theory this value could be adjusted as the agent learns, reflecting how dopamine neurons rapidly adapt to the range of reward values in the environment (*Tobler et al., 2005*).

*Appendix 1—figure 3*, center, shows the effect of normalization for the example in question. We see that the rapid decay is substantially decreased, and simulating into a farther time horizon of 100 trials shows a trend toward, but not final convergence at, zero (*Appendix 1—figure 3*, right). (Note that OpAL* behaves well for several hundred trials in the experiments we simulated in this article.) While there remains a general decay over time, as previously stated, the behavior is reminiscent of advantage learning curves, which have the positive feature that such decay can encourage the agent to explore after many trials in the event the world has changed. Furthermore, it is plausible that other learning mechanisms, such as more habitual stimulus–response learning, also contribute to choice after many learning trials (*Frank and Claus, 2006*). Thus striatal weight decay, which has been documented empirically (*Yttri and Dudman, 2016*), may not be detrimental for procedural performance. Normalizing, therefore, addresses one factor (large RPE magnitudes) contributing to the rapid decay in early trials demonstrated by *Möller and Bogacz, 2019* while still preserving core OpAL dynamics, which allow it to capture a range of biological phenomenon as well as hypotheses for advantages of dopaminergic states presented in this article.

Secondly, to address the original issue raised by *Möller and Bogacz, 2019* that OpAL weights decay with oscillating prediction errors, we introduced annealing of the actor learning rate. This is a common addition to reinforcement learning algorithms where the learning rate is large in early stages of learning to avoid local minimums and slowly decreases with time to protect values in later stages of learning from rapid updating. (To allow for change points in reward statistics, other mechanisms capturing the effects of cholinergic interneurons have been shown to be useful in BG networks and OpAL variants; *Franklin and Frank, 2015*). *Appendix 1—figure 3*, right, shows that while actor weights still decrease with the addition of annealing, they no longer converge to zero and lose all prior learning as demonstrated in *Möller and Bogacz, 2019*. Ordinal rankings of G and N weights for various probability of rewards are also preserved after extended learning (1000 trials, *Appendix 1—figure 4*). Since fixed annealing with time would hinder an agent's ability to respond to sudden changes in the environment, we modulate annealing according to the uncertainty generated by the Bayesian meta-critic. This allows OpAL* to adequately respond to sudden changes

in environmental statistics (e.g., when the rewarding option changes). Further augmenting OpAL* with decay of actor weights as proposed by *Franklin and Frank, 2015*, thought to be implemented by cholinergic neurons, also allows OpAL* to remain flexible to extremely volatile environments, such as when reward rate varies trial-by-trial according to a random walk (*Appendix 1—figure 5*), but optimizing for these types of scenarios is beyond the scope of this article.

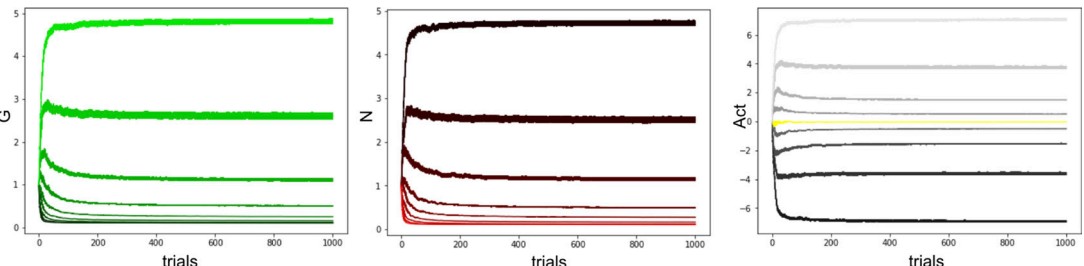

**Appendix 1—figure 4.** Our proposed annealing preserves ordinal rankings for extended learning. Curves averaged over 100 simulations.

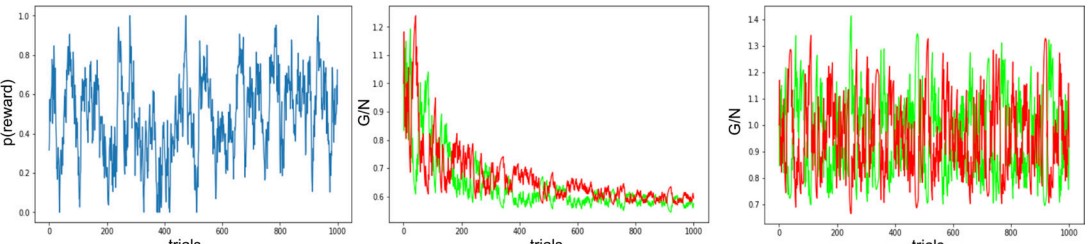

**Appendix 1—figure 5.** Combination of decay and annealing can also improve OpAL* responses to highly volatile environments, such as in a random walk (left, probability of reward). Middle: OpAL* annealing slows convergence of G weights (green) and N weights (red) to zero but the model still suffers from decay. Right: OpAL* with decay and Bayesian annealing allow flexibility, though the optimization of this combined mechanism for variable environments is beyond the scope of this article.

Notably, annealing intensifies a common tradeoff of increased stability for decreased flexibility. While lowering the learning rate protects the actor weights from converging to zero and allows the model to retain a useful policy, the actor weights become insensitive to changes in reward contingencies when the variance of the meta-critic is sufficiently small. In this article, our scope focus on stationary reward environments (where rewards do not change) and this flexibility is not required. However, in environments where reward consistently drifts, OpAL* can track such rapid reward fluctuations with less annealing, but suffers from gradual decay as a result (*Appendix 1—figure 5*), albeit less so than the OpAL model without annealing (not shown). However, it is possible to further improve OpAL*'s flexibility by simply incorporating decay of actor weights as proposed by *Franklin and Frank, 2015*, who introduced this mechanism to approximate the impact of cholinergic neurons and which specifically addressed the flexibility stability dilemma. One can see that a simple version of this decay does improve flexibility (*Appendix 1—figure 6*, right, and *Appendix 1—figure 5*, right). Assessing whether OpAL* advantage could be further improved when optimizing for weight decay in conjunction with annealing parameters (held fixed in this paper) is left for future work, though we did verify that including decay for the optimized parameters presented in *Figure 6A* demonstrated similar performance advantages for OpAL* while enhancing flexibility. Furthermore, the effect of learning rate on stability in drifting reward environments should also be explored; the higher order terms present implicitly in the G/N weights (*Equation 22*) imply that lower actor learning rates throughout learning will support enhanced stability because larger learning rates increase contribution of the higher order terms which drive decay – even though larger learning rates may be optimal in short time horizons and in stable environments such as those presented. The general stability–flexibility tradeoff is an issue studied beyond BG-circuitry (*Nassar et al., 2012*; *Iglesias et al., 2013*) and can never fully be eliminated, though approaches such as those we outline here may help improve OpAL*'s flexibility across a broader range of environments.

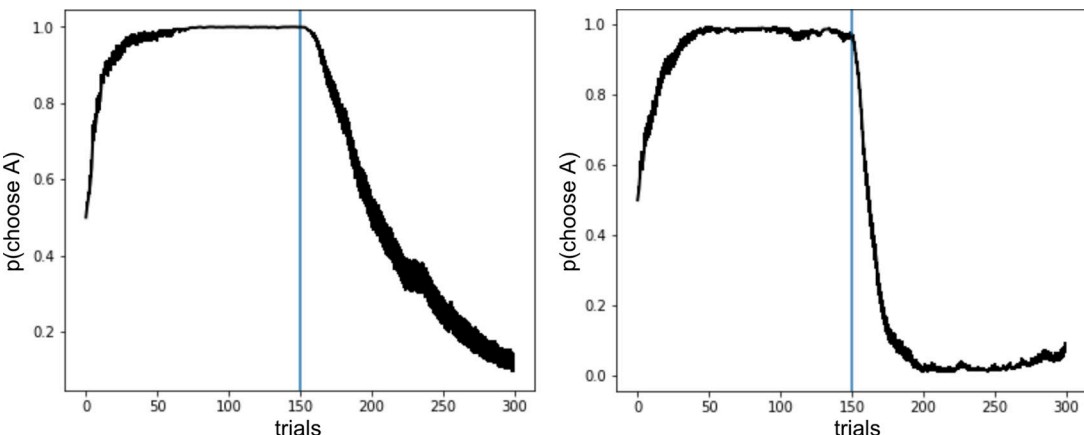

**Appendix 1—figure 6.** Left: Bayesian annealing allows OpAL* to remain flexible when the rewarding option suddenly changes. Two-armed bandit task where option A is rewarded 80% of the time and option B is reward 50% of the time. After 150 trials, the reward contingencies reverse and option A is the least rewarding. Right: augmenting OpAL* further with decay as proposed by *Franklin and Frank, 2015* increases adaptiveness to a switch point.

## Derivation of OpAL actor weights as a function of RPE history

The actor weight updates are recursive, whereby the G and N weights are updated not only as a function of RPE on trial i ($\delta_i$) but also as a function of the previous actor weights ($G_{i-1}$ and $N_{i-1}$). Consider $t$ successive trials of selecting action $a$. Expanding *Equation 3* using an initial $G_0 = 1$ (as used in all simulations so that the first update reduces to a standard RL algorithm, with weight changes directly proportional to the first RPE), we have

$$
\begin{aligned}
G_t(a) \;&= G_0 + G_0\alpha_G\delta_1 + G_1\alpha_G\delta_2 + G_2\alpha_G\delta_3 + G_3\alpha_G\delta_4 + \ldots \\
G_t(a) \;&= 1 + \alpha_G\delta_1 + (1 + \alpha_G\delta_1)\alpha_G\delta_2 + [(1 + \alpha_G\delta_1) + (1 + \alpha_G\delta_1)\alpha_G\delta_2]\alpha_G\delta_3 \\
&\quad + [(1 + \alpha_G\delta_1) + (1 + \alpha_G\delta_1)\alpha_G\delta_2 + [(1 + \alpha_G\delta_1) + (1 + \alpha_G\delta_1)\alpha_G\delta_2]\alpha_G\delta_3]\alpha_G\delta_4 \\
&\quad + \ldots \\
&= 1 + \alpha_G\delta_1 + \alpha_G\delta_2 + \alpha_G^2\delta_1\delta_2 + \alpha_G\delta_3 + \alpha_G^2\delta_1\delta_3 + \alpha_G^2\delta_2\delta_3 \\
&\quad + \alpha_G^3\delta_1\delta_2\delta_3 + \alpha_G\delta_4 + \alpha_G^2\delta_1\delta_4 + \alpha_G^2\delta_2\delta_4 + \alpha_G^2\delta_3\delta_4 + + \alpha_G^3\delta_1\delta_3\delta_4 \\
&\quad + \alpha_G^3\delta_2\delta_3\delta_4 + \alpha_G^4\delta_1\delta_2\delta_3\delta_4 + \ldots
\end{aligned}
$$

Collecting all the terms and rearranging:

$$
G_t(a) = (1 + \sum_{i=1}^{t}\alpha_G\delta_i + \sum_{i=1}^{t-1}\sum_{j=i+1}^{t}\alpha_G^2(\delta_i\delta_j) + \sum_{i=1}^{t-2}\sum_{j=i+1}^{t-1}\sum_{k=j+1}^{t}\alpha_N^3(\delta_i\delta_j\delta_k) + \ldots + \prod_{i=1}^{t}\alpha_G^t\delta_i)
$$

$$(35)$$

and similarly, the update $\Delta G_t(a)$ can be written as:

$$
\Delta G_t(a) = \alpha_G\delta_t(1 + \sum_{i=1}^{t-1}\alpha_G\delta_i + \sum_{i=1}^{t-2}\sum_{j=i+1}^{t-1}\alpha_G^2(\delta_i\delta_j) + \sum_{i=1}^{t-3}\sum_{j=i+1}^{t-2}\sum_{k=j+1}^{t-1}\alpha_G^3(\delta_i\delta_j\delta_k) + \ldots + \prod_{i=1}^{t-1}\alpha_G^{t-1}\delta_i)
$$

$$(36)$$

The same equations apply for the $N$ weights, replacing each $\delta_i$ with $-\delta_i$. The higher order terms in the above equations introduce distortions in the G and N weights such that they grow nonlinearly as a function of the consistency of prior RPEs (e.g., when most pairs $\delta_i\delta_j$ are of the same sign they have greater impact on the weight update and the ultimate weight itself than the sum of their individual contributions). Note also that these terms introduce decay in the weights after the critic has converged (as long as there remain both positive and negative RPEs that sum to zero), but OpAL* stabilizes the weights via annealing.

# Appendix 2

## Supplemental note 1

For clarity, 'benefits' and 'costs' are evaluations relative to the critic's expectation. The exact numeric value is not interpretable. Rather, high benefits ($G$) convey that an action is better than expected more often; high costs ($N$) convey that an action more often disappoints relative to the critic's expectations.

## Supplemental note 2

One can adjust DA without the conservative inference process but there is a cost to misestimation of environmental richness that can arise due to stochasticity in any given environment, which can lead to reliance on the wrong actor; see *Appendix 1—figure 1*. Although we focus on the Bayesian implementation here, other heuristics for achieving the same desideratum can be applied, for example, waiting a fixed number of trials before changing the dopaminergic state by integrating information from the standard RL critic to estimate context value. However, using a beta distribution (whose mean implicitly incorporates uncertainty) and explicitly adapting according to the distributions' standard deviation isolates whether any differences in performance between OpAL* and a baseline model with fixed dopaminergic states were a result of dopamine modulation rather than an ineffective use of the meta-critic (e.g., waiting too few trials) or a suboptimal meta-critic (e.g., poorly tuned learning rate for RL version).

## Supplemental note 3

The meta-critic provides a proxy for the agent's uncertainty about the task which can be used to uniformly anneal learning rates across actions. Other implementations are possible, however, for example, the critic itself could be Bayesian and have access to uncertainty within individual state-action values to guide annealing. We chose the current implementation for simplicity.

## Supplemental note 4

In our simulations, the OpAL+ model includes the annealing and normalization additions as discussed in the section 'OpAL*.' While these features were not present in the original version presented in *Collins and Frank, 2014*, we found that they are necessary to address pathological behavior as discussed in the section 'OpAL*' and in Appendix 1 ('Addressing'; *Möller and Bogacz, 2019*). The crucial distinction we emphasize between OpAL+ and OpAL* is the non-dynamic versus dynamic adaptation of DA, respectively.

## Supplemental note 5

As such, relatively larger leads to better discrimination among highly rewarded options, and relatively larger leads to better discrimination among lean options, consistent with the many effects of DA manipulation on asymmetric learning in the literature; *Collins and Frank, 2014*, see *Figure 8—figure supplement 4*.

## Supplemental note 6

It is worth noting that in more standard actor-critic models, slower actor learning relative to critic learning is often preferable in order for the critic to properly evaluate the value of a stable policy (*Castro and Meir, 2010*). It is therefore also noteworthy that OpAL* stabilizes its policy via annealing actor learning rates as function of uncertainty, so that when the critic value does converge, it is based on a stable policy. The prediction errors generated by the critic could then in principle be used in temporal-difference sequential decision settings just as in standard actor-critic models.

## Supplemental note 7

The G weights of this model are designed to converge to the expected payoffs of an action, while the N weights are designed to converge to the expected costs. With lmag = 0, actions have no explicit expected costs and in theory reduces to a nonopponent mechanism. To best equate scope and explore the role of different [Hebbian vs. non-Hebbian] nonlinearities in combination with opponency, we explicitly included a cost in these simulations. Given the normalization of reward values used in OpAL*, we expect it would perform similarly for lmag = -1 as it does for lmag = 0;

the actor weights and performance OpAL* depend on the experienced consistency of positive and negative outcomes, not their absolute magnitudes. See *Equation 22*.

## Supplemental note 8

As gamble offers were explicit, removing uncertainty in trial richness, we omitted the parameter which modulated DA levels by degree of certainty in environmental richness, further reducing model complexity. OpAL*'s ability to capture shifting patterns of risky choice should thus be viewed as a by-product of interacting opponent, nonlinear, and dynamic DA mechanisms rather than a result of high degrees of freedom.

## Supplemental note 9

For clarification, *Rutledge et al., 2015* highlighted that the drug effects appear 'value-independent,' whereas here we explicitly are changing risk sensitivity according to the interaction between drug and offer value. It is important to note, however, that their definition of value differs than that used to modulate dopaminergic state in these simulations. In *Rutledge et al., 2015*, value is defined as the advantage of the gamble, i.e., the difference between the expected value of the gamble and the sure reward. Here, we considered value to be the combined overall value of the offer presented, such that positive RPEs exist when values are greater than expected, and are in turn was modified by drug dosage. It is this component that captures the selective increase in gambling in gain trials. Note that the model does predict that such gambles would be yet more likely when the potential benefit of gambling is larger [i.e., when gains are particularly large] – but that this effect would also be present off drug. It is also possible that the value-independence in *Rutledge et al., 2015* resulted from a ceiling effect for gambling in higher gain trials.

## Supplemental note 10

This reference-dependent modulation is analogous to our learning experiments, in which the implicit baseline used a mean reward probability of 50%, and where environments with higher estimated reward probabilities were considered 'rich' and those below 50% were considered 'lean.' One could more generally apply the terms 'rich' and 'lean' to any values which deviate from a determined baseline, where $\bar{R}$ represents the estimated richness of the current environment and $\boldsymbol{B}$ represents the mean of an uninformative prior over the expected outcomes, $\rho(t) \propto \bar{R}(t) - B$. $\rho \geq 0$ would be considered 'rich'; $\rho < 0$ would be considered 'lean.' Indeed, previous work has suggested that of a single environment may be encoded by tonic levels of dopamine, inducing changes in vigor of actions (*Niv et al., 2007*), but does not model changes in the choices themselves as we do here. A similar approach is used in average reward RL. Rather than maximizing the total cumulative reward, average reward RL additionally optimizes the average reward per timestep. Reward prediction errors are therefore computed relative to the long-term average reward per time step ($\bar{r}$), resulting in $\delta(t) = r(t) - \bar{r} - V(t)$. $\rho$ as operationalized in OpAL* resembles a prediction error at the task/environment level, though may additionally be influenced by trial-by-trial prediction errors when trials are sufficiently distinct as in the interleaved gambles in *Rutledge et al., 2015*.

