## [Editor Report]

This paper provides a formal analysis of the normative advantage of the opponent pathways of the basal ganglia circuit for cost-benefit decision-making. Specifically, a previously introduced Hebbian nonlinearity is combined with reward-based DA modulation to optimize exploration across lean and rich environments, and across a range of pharmacological and contextual manipulations. The scope of the model, its biological plausibility, and its normative and descriptive aspects are likely to have a significant impact.

---

## [Decision Letter]

**Decision letter after peer review:**

[Editors’ note: the authors submitted for reconsideration following the decision after peer review. What follows is the decision letter after the first round of review.]

Thank you for submitting the paper "On the normative advantages of dopamine and striatal opponency for learning and choice" for consideration by *eLife*. Your article has been reviewed by 3 peer reviewers, including Mimi Liljeholm as the Reviewing Editor and Reviewer #, and the evaluation has been overseen by a Senior Editor.

Comments to the Authors:

We are sorry to say that, after consultation with the reviewers, we have decided that this work will not be considered further for publication by *eLife*.

Specifically, concerns about the stability and adaptivity of the model, based on actual simulations, were judged serious enough to preclude an invitation to revise. However, if you feel confident that you can compellingly address all issues raised by the reviewers, and the matter of environmental dynamics in particular, we would be happy to consider a resubmission.

*Reviewer #1 (Recommendations for the authors):*

The authors aim to demonstrate the normative advantage of opponent basal ganglia pathways, using dynamic dopamine (DA) modulation by environmental reward statistics. The finding that well-known neurophysiological and psychopharmacological mechanisms yield rational decision-making in ecologically plausible environments is a major strength of the paper. The methods are rigorous, and the advantage of the model is reliable across a range of parameters and paradigms. More could be done to explain the optimality of certain model predictions, to map the correspondence between model predictions and animal behavior, and to articulate the conceptual relationship between relevant constructs, such as exploration, discrimination, and risk-taking. Nevertheless, the scope of the model, its biological plausibility, and its normative and descriptive aspects suggest that it may have a significant impact.

1. More could be done to detail the, sometimes counterintuitive, optimality of certain model predictions. For example, while it is straightforward that you need to explore to discover better options, the need to sample sub-optimal options so that you know to avoid them makes less sense: if you don't sample them, you are already avoiding them, and when you do choose them, you learn the values – why is it adventitious to do the sampling early in the process as opposed to later? Another example is the claimed optimality of avoiding gambles in lean environments, despite the expected value of the gamble being greater than that of the ST.

2. It would be helpful if there was a description, early in the Introduction, and perhaps again in the Discussion, of how the relevant constructs (i.e., exploration, discrimination, and risk-taking), are related conceptually – note that exploration can be characterized as risk-taking and that the decision to gamble may reflect either exploration or value-based discrimination.

3. In Figure 10a, it is unclear why L-DOPA does not boost G weights and reduce N weights, yielding a greater probability of gambling, on loss trials as it does on gain trials?

4. In Equations 1-3, why not show the updating of Act values, or, if identical to V(a), be explicit about that. Also, Equation 14 is referred to as a prediction error in the text, but the notation indicates an action value [i.e., V(a)].

5. It is argued that OpAL* eliminates the need for a priori knowledge about environmental reward statistics, but DA-modulation depends on confident estimates of the values (i.e., reward probabilities) of all available actions – the implications of this reliance on reward estimates seem under-explored in simulations.

6. It is unclear from the curves in Figure 7 whether the models reach equilibrium by 100 trials. Please show asymptotic performance.

7. Why use only forced-choice simulations? It is hard to imagine a real-world scenario in which "no action" is not an alternative. Failure to include such an option detracts from the ecological validity of the experiments.

*Reviewer #2 (Recommendations for the authors):*

I really enjoyed this paper that significantly advance our understanding of the basal ganglia circuit. I have some suggestions for additional simulations that can allow establishing to which extent previous findings in reinforcement learning can be accounted for by the OpAL(*) models.

A key concept of the model is its sensitivity to environmental richness (i.e., whether the agent is in a lean or in a rich environment). The concept behind the "rho" variable is very closely related to that of "state value" or "reference point" as it has been applied to reinforcement learning and valuation since Palminteri et al. (2015). The key demonstration of a crucial role of overall "environmental richness" in learning and valuation came from a task, which, explicitly coupling features of the Frank (2004) and Pessiglione (2006) tasks, showed that transfer learning performance is highly context-dependent (see Palminteri and Lebreton, 2021; Daw and Hunter, 2021 for review of these results). My question here is whether or not the OpAL(*) models are sufficient to generate such context-dependent preferences. Does this (very robust) behavioral effect naturally emerge from the model? Or do we need to specify an additional process to explain this? Can you simulate Palminteri et al. (2015) using the OpAL(*) models and show what are the performance in the learning and transfer phase?

A robust and replicable finding in human (and nonhuman) reinforcement learning is that when fitting the model with two learning rates for positive or negative prediction errors (starting from Frank et al. 2007, supplementary materials, continuing in Lefebvre et al. 2017; Gagne et al. 2020; Farashahi et al. 2019; Ohta et al. 2021; Chambon et al. 2020 – the rare occurrences of the opposite pattern are generally explained by wrongly initializing the Q-values at pessimistic values). The optimality of this bias has been sometimes investigated (see Cazé and Van den Merr 2013; Lefebvre et al. 2022). My question here is whether and/or under which circumstances do the OpAL(*) models explain the ubiquity of this pattern? If you simulate OpAL(*) models and then fit the asymmetric model, would you (quasi-)systematically retrieve the asymmetric pattern?

Finally, the actor prediction errors are normalized. I can see the logic of this. I was nonetheless puzzled by the functional form of the normalization that is not really a range normalization (Lmag is lacking at the numerator). Consider for example Bavard et al. (2021), which propose a full range normalization rule. What is the rationale for this particular form of normalization? Are Lmag and Rmag learned (via δ rule or other, see Bavard et al.) or specified in advance?

*Reviewer #3 (Recommendations for the authors):*

This manuscript proposes a refined version of the OpAL model, which aims to address the numerical instability problems present in the previous version and demonstrate the normative advantages of the model in learning tasks. Unfortunately, the instability problems still persist in the proposed model, and the normative comparison is unconvincing due to the limited range of scenarios and tasks investigated.

Comments

1. It has been previously pointed out that the OpAL model is numerically unstable: all synaptic weights converge to 0 with learning, and as a result, the model has a tendency to make random choices with practice. The authors claim that the proposed model "addresses limitations of the original OpAL", but in fact, it does not – it just postpones their effect by introducing learning rates that decay with time, which ensures that the weights simply stop changing as learning progresses. Consequently, there are several problems with the proposed argumentation in the paper:

a. The paper does not adequately describe the scale of the stability problem faced by the original OpAL model. A cartoon in Figure 1 falsely suggests that the weights in the OpAL model converge to stable values with learning, by contrast even for this scenario all weights decay to 0. The appendix attached below this review contains Matlab code simulating the OpAL model, and the first figure generated by this code shows OpAL model simulations corresponding to Figure 1. Running the code clearly shows that all weights decay to 0 in this simulation. Therefore, weight evolution in the OpAL model in Figure 1, needs to show the actual simulation of the OpAL model rather than cartoons with qualitatively different behaviour of the weights.

b. Similarly, a reader may get a false impression that the instability problems only occur in the scenario in Figure 13, because the manuscript states in lines 1104-1105 that the OpAL model "in carefully constructed situations, gives rise to unstable actor dynamics". However, as pointed out in my previous comment, these problems are ubiquitous, and in fact, Mikhael and Bogacz (2016) also illustrated this problem in Figure 9 of their paper for randomly generated rewards. The reason why Moeller and Bogacz (2019) considered the scenario in Figure 13 is that for this case one can easily show analytically that the weights in the OpAL model converge to 0.

c. To solve the stability problem, the authors modify the model such that the learning slows with time. However, they admit in l.1130: that the model loses its memory with time, and they propose that the habit system may take over learning and rescue animal bahaviour. However, animals with lesions of dorso-lateral striatum (which is known to underlie habitual behaviour) still can perform learned tasks (Yin HH, Knowlton BJ, Balleine BW. 2004. Lesions of dorsolateral striatum preserve outcome expectancy but disrupt habit formation in instrumental learning. European Journal of Neuroscience 19:181-189.) which is not consistent with the decay of memory of goal directed system.

d. The reduction in learning rate with time introduced in OpAL* only makes sense in stable environments. In changing environments decaying learning rate does not make sense, because the animal would be unable to adapt to changing rewards. However, the second figure generated by the code in the Appendix shows that the weights also decrease to 0 in environments where the reward probability constantly fluctuates according to a random walk (in this simulation the rewards are binary, so the normalization of prediction error introduced in OpAL* does not make any difference). In summary, the learning rules of the OpAL model decay to 0 even if the reward probability is constantly changing, so for this case the model is unable to track reward probability no matter if the decrease of learning rate is introduced or not.

2. The manuscript shows that the OpAL* model can achieve higher rewards than alternative models in simple tasks with constant reward probabilities. However, there are severe problems with the arguments in the paper.

a. All the simulations are run with a small number of trials (i.e. 100). Beyond this number of trials the OpAL model suffers from decay of weights (see my previous comment), so it is questionable whether it would have performance advantages. Therefore, it is critically important that the performance comparison includes simulations with more (e.g. 1000) trials.

b. The simple tasks simulated in the paper has been a focus of much research. There is known optimal learning algorithm (Gittins index) and several well performing ones (e.g. Thomson sampling, Upper confidence bound algorithms). I feel that the normative comparison needs to include a comparison with some of these known solutions.

c. The reason why the OpAL* achieves best performance in the chosen scenario is not well explained. Lines 339-343 present the key of the mechanism: "opponency allows the non-dominant (here, *G*) actor to contribute early during learning (before *N* weights accumulate), thereby flattening initial discrimination and enhancing exploration. Second, the Hebbian nonlinearity ensures that negative experiences induce disproportional distortions in *N* weights for the most suboptimal actions after they have been explored (Figure 6a), thereby allowing the agent to more robustly avoid them (Figures 6b and 6c)." However, the cited figures do not illustrate this mechanism, and to show this, it would be better to show how the weights change during learning.

d. The performance of model is compared in two ways: (i) for the best parameters for each model and (ii) for an average over the range of parameters. I think that method (i) is valid, but method (ii) may introduce a bias if the range is chosen such that the optimal range for one model overlaps with the tested range for one model more than for another. Therefore, I feel that method (ii) should not be included in the manuscript.

3. The manuscript also describes comparison of performance of OpAL* with the model by Moeller and Bogacz (2019), however the manuscript includes incorrect statements about the latter model, and the way it is parameters are chosen does not reflect the conditions in simulated scenarios.

a. The manuscript states in l. 392: "However, in actuality, the convergence to expected payoffs and costs in this model depends on having a constrained relationship between parameters optimized by a priori access to the distributions of rewards in the environment." This is not true. For a given set of parameters, the convergence to payoffs and costs is guaranteed for any reward distribution – derivation of the condition for encoding payoff and costs in Eq 22 and 23 in Moeller and Bogacz does not make any assumptions about knowledge of reward distribution. Please remove this sentence.

b. The condition on parameters derived by Moeller and Bogacz are only necessary to learn payoffs and costs if every trial includes a cost (e.g. an effort to make an action) and a payoff (e.g. the outcome of the action). This is not the case in the presented simulations, so the model parameters do not even need to satisfy the conditions of Moeller and Bogacz.

c. The model is simulated with constant "tonic dopamine" level, and by mathematical construction, in this case the models described by Moeller and Bogacz is mathematically equivalent to Q-learning with decay controlled by parameter λ, and will reduce to the standard Q-learning for λ = 0. Since the Authors simulated Q-learning, it is not even clear if it is necessary to simulate Moeller and Bogacz model because by its definition it will have identical (with λ=0) or very similar performance.

d. The manuscript states in l. 415: "Finally, the model in Möller and Bogacz (2019) demonstrated poor across-environment performance, performing only slightly above chance in the rich environment. Results are not shown for this model" – one should not make such statements without actually presenting evidence for them.

Appendix – code simulating OpAL model

function run_OpAL

figure(1)

trials = 2000;

pr = [0.9 0.8 0.7; 0.1 0.2 0.3];

labels = {'rich', 'lean'};

for env = 1:2

subplot (2,1,env);

for action = 1:3

p = zeros (1,trials)+pr(env,action);

[G,N] = opal (p);

plot (G, 'g');

hold on

plot (N, 'r');

end

xlabel ('Trials');

title (labels{env});

legend ('G','N');

end

figure(2)

trials = 1000;

p = zeros (1,trials)+0.1;

for t = 1:trials^-1^

p(t+1) = p(t) – 0.1*(p(t)-0.5) + 0.1*randn;

if p(t+1) > 1

p(t+1) = 1;

elseif p(t+1) < 0

p(t+1) = 0;

end

end

[G,N] = opal (p);

subplot (2,1,1);

plot (p);

ylabel ('Reward probability')

xlabel ('Trials');

subplot (2,1,2);

plot (G, 'g');

hold on

plot (N, 'r');

legend ('G','N');

xlabel ('Trials');

end

function [G,N] = opal (p)

trials = length(p);

V = zeros (1,trials)+0.5;

G = zeros (1,trials)+0.1;

N = zeros (1,trials)+0.1;

α = 0.2;

for t = 1:trials^-1^

r = (rand

δ = r – V(t);

V(t+1) = V(t) + α*δ;

G(t+1) = G(t) + G(t)*α*δ;

N(t+1) = N(t) – N(t)*α*δ;

end

end

Thank you for resubmitting your work entitled "On the normative advantages of dopamine and striatal opponency for learning and choice" for further consideration by *eLife*. Your revised article has been evaluated by Joshua Gold (Senior Editor) and a Reviewing Editor.

The manuscript has been improved but there are some remaining issues that need to be addressed, as outlined below:

Essential revisions:

Reviewer 3 still has significant concerns about the stability of the model, and about the mechanisms accounting for its superior performance. Please make sure to fully address all requests for clarification detailed below.

*Reviewer #1 (Recommendations for the authors):*

The authors have successfully addressed my concerns.

*Reviewer #2 (Recommendations for the authors):*

The authors successfully addressed my suggestions.

*Reviewer #3 (Recommendations for the authors):*

This manuscript proposes a refined version of the OpAL model, which aims to address the numerical instability problems present in the previous version, and demonstrate normative advantages of the model in tasks involving balancing exploration and exploitation.

I thank the Authors for replying to my review. I feel that the manuscript has been significantly improved, and the comparison of performance with UCB is particularly interesting. Nevertheless, there are still issues that need to be addressed. In particular, it is not clear from the analysis why OpAL* can outperform UCB, and multiple statements about the stability of the model are still misleading.

Comments:

1. It needs to be further clarified why OpAL* outperforms UCB and Q-learning. I have to admit that I am surprised by the higher performance of OpAL* over UCB, because UCB is not an easy algorithm to outperform. Unlike Q-learning, UCB has perfect memory of all rewards and does not forget them. Then I realized that OpAL* also has such perfect memory, as its Bayesian critic counts the rewarded and unrewarded trials for each option. I feel this is the main reason why OpAL* outperform Q-learning. I suggest explaining this important property of OpAL* in the text, and it would be good to test it, e.g. by replacing Bayesian critic by a normal forgetful critic, and testing "Contribution of perfect memory". Please also discuss if such perfect memory is biologically realistic – how could it be implemented in biological neural network?

However, the most surprising result is that OpAL* outperforms UCB. This cannot be explained by most of the discussion of Mechanism section focussing on the gaps between weights for different options, because such gaps are only important if an algorithm choses actions stochastically based on weights, while UCB is practically a deterministic algorithm (beyond initial few trials it will deterministically chose an option). The only mechanism which can explain outperforming UCB is explained in a paragraph starting in line 574. Please investigate further, and provide a clear explanation for how it is possible for OpAL* to outperform UCB.

2. The stability of OpAL* model needs to be honestly presented. The main issue pointed by Mikhael and Bogacz (2016) and then by Moeller and Bogacz (2019) is that the weights in OpAL will asymptotically converge to 0. It is still evident from simulations of the OpAL* in the manuscript and in the response letter that on average the weights decay with trials. There is no demonstration that OpAL* can prevent its weights from converging to 0 eventually, and at the same time can respond to changing rewards. In the revised version, such demonstration has been added in Figures 15c right and 15d right, but it is not possible to understand how these simulations were performed from the paper, and it seems that they are for a model with "weight decay", which is not described in the manuscript. It is not clear if the model with weight decay has the advantages of OpAL* in exploration/exploitation that are the focus of the manuscript. Therefore, I feel that the manuscript has to be modified in one of two ways: The Authors may change the model in the paper to one with weight decay and analyse its performance in exploration/exploitation task. Alternatively, if the current OpAL* model remains the focus of the paper, it needs to be honestly admitted that the OpAL* model suffers from the problem that weights will eventually converge to 0 or the model will stop adapting to changes in the rewards.

Specific comments:

Equation 15 – How is std X estimated? Is it computed from the analytic expression for β distribution? If so, this expression is complex (involving division, square, etc.), so please comment on how such computation could be made by biological networks of neurons.

Equation 22 – Please explain parameter T. As above, please explain how X is computed in the simulation.

Line 293: "These modifications improve the robustness of OpAL* and ensure that the actor weights are well-behaved" – I do not agree with this statement, because in the OpAL* model the weights still converge to 0 unless they are prevented from convergence to 0 by making the model non-adaptive. Please replace "well behaved" in the cited statement, by a more specific description.

Line 499: "Indeed, algorithms like Q-learning and UCB converge well when an option is well-sampled, but the speed and accuracy of this convergence is affected by stochastic sampling". This argument does not apply to UCB, which is deterministic, and hence the sampling in UCB is NOT stochastic.

Line 582 – this again does not seem to apply to UCB, which is deterministic.

Figure 6a is very interesting, but I have a few suggestions to make it more informative. I simulated UCB on the problems in Figure 6a and verified that it indeed gives similar performance to that visualized in this figure, but found that with 1000 iterations there is still substantial variability in the results, and sometimes you get non-monotonic changes in accuracy as shown in blue curve in Figure 6a left, which disappear if the number of iterations is increased. Hence, I suggest to increase the number of iterations (repetitions) in Figure 6a to 10,000 to get less noisy curves.

In Figure 6a the gap between UCB and OpAL* is higher in the right panel but it is not clear if this is due to change in richness or in the number of options. It would be helpful to add two more panels ([0.2 0.3] and [0.8 0.7 0.7 0.7 0.7 0.7]). Also for an easier replicability, it would be good to list the parameters used in simulations of each model.

l. 1478: "Normalizing, therefore, addresses the valid concerns of Möller and Bogacz (2019) while still preserving core OpAL dynamics". I do not agree – the key concern of Möller and Bogacz (2019) was that the weights decay to 0 eventually. The normalization does not fix the problem. In the simulations I did in the previous round of the review, rewards were 0 and 1, so the normalization had no effect.

Equation 30 – I guess the last 2 "cases" were included accidentally (they are a typo), and should be removed.

l.1112: UCB – Please provide the reference for the paper describing UCB. In this algorithm, the model is initialized to selecting each action once before relying on estimated values. Do you do such initial selection? If so, please say this.

Throughout – In many places in the manuscript there are references to appendix, but the appendix is long, and it takes time to find information. Hence whenever you point the reader to the appendix, please point to a specific section or figure.

---

## [Author Response]

[Editors’ note: the authors resubmitted a revised version of the paper for consideration. What follows is the authors’ response to the first round of review.]

Comments to the Authors:We are sorry to say that, after consultation with the reviewers, we have decided that this work will not be considered further for publication by eLife.Specifically, concerns about the stability and adaptivity of the model, based on actual simulations, were judged serious enough to preclude an invitation to revise. However, if you feel confident that you can compellingly address all issues raised by the reviewers, and the matter of environmental dynamics in particular, we would be happy to consider a resubmission.

We provide a full point-by-point response below to all Reviewer comments (including adding several new simulations, new account of empirical data, and a favorable comparison to additional RL models with more adaptive exploration mechanisms such as Upper Confidence Bound). But because concerns of stability (i.e., that actor weights decayed to zero with more training) and adaptivity (the ability to flexibly switch policies) were the basis for the decision, this letter focuses first on these issues. We then highlight that, in response to all reviewers, we have now more fully exposed the mechanism by which OpAL* advantages are borne out, and in so doing provide a crisper analysis of how it relates to a broader class of RL algorithms. Addressing the reviews also allowed us to clarify the sense in which our model is normative – in terms of policy optimization – which is different in scope to models that aim to learn the expected values. As we point out below, this comparison is akin to Q learning vs policy gradient or other direct policy search methods in modern RL.

This stability concern regarding the original model was articulated in the work by Mikhael and Bogacz and Möller and Bogacz, who addressed it by removing the Hebbian term from weight updates in the opponent actors, but adding in other nonlinearities in learning from positive and negative RPEs. Instead, we sought to retain the Hebbian nonlinearity, given its biologically motivation and the several modeling studies showing that it is needed to capture choice performance, motor learning, and plasticity studies across species (Beeler et al. 2012, Cockburn et al. 2014, Wiecki et al. 2009), which we now further emphasize and discuss. But moreover, our main goals were to explore why this seemingly pathological mechanism might be normative in terms of an efficient coding mechanism for choice, especially when DA is properly regulated as a function of reward history. Our main results were (and are) that this combination of mechanisms is more robust and adaptive than not including either of them on their own (or neither of them), including when comparing to standard RL and other algorithms.

To begin, we note that the main stability concerns highlighted by R3 were based on simulations provided in MATLAB code that did *not* reflect the proposed model in our original submission, but rather the original 2014 OpAL model with particularly pathological parameters that exacerbate these effects. Our newer model did adjust some mechanisms to substantively ameliorate, if not completely eliminate, these stability issues, as we show in more detail below, and as we have now further improved in response to these concerns.

Stability

First, as Reviewer 3 correctly points out, initialization of G and N weights affects performance. The standard for OpAL is to initialize such values at 1, which reduces the first trial to a standard TD update rule; any future changes in learning are a function of the whole reward history. Simulations provided by Reviewer 3 initialize this value at.1, leading to rapid decay. Second, the adaptive growth dynamics of the G/N actors in OpAL(*) rely on slow critic convergence relative to actor learning, as we now elaborate in our Mechanism section “Opponency and Hebbian nonlinearity allows OpAL* to optimize action gaps”. The simulations provided by Reviewer 3 include a high learning rate (.2) shared by both the actor and the critic. If we simply reduce actor learning rate to.1 and critic learning rate to.05, the decay is much reduced and the actor weights discriminate between options after extensive learning even within the original OpAL model. We show the results of these simulations in Author response image 1–Author response image 3 for completeness, but the actual paper focuses instead on the new model (which is more stable).

**Author response image 1. sa2fig1:** Initial simulations from with Reviewer 3’s code and parameters (more positive G(N) values correspond to actions with higher reward probability in rich(lean)).

**Author response image 2. sa2fig2:** Modified learning rates.

**Author response image 3. sa2fig3:** Modified learning rates and actor initialization.

We show these more systematically across multiple simulations in response to R3 comments below. In any case, we agree that the decay in the original model needed to be addressed, especially given that a main theme of our paper is that the model should be robust across a range of parameters. In our first submission, we included annealing and normalization to address this. Reviewer 3 noted that our new model “just postpones their effect by introducing learning rates that decay with time, which ensures that the weights simply stop changing as learning progresses.” But (beside the more central contribution on adaptive modulation / efficient coding), the model in the original submission also introduced a Bayesian critic with which to compute RPEs, and a soft annealing of learning rates with time. Simulations from this model *do* show that the (i) weights no longer converge to zero with extensive training, (ii) that the rank order of reward probabilities is still preserved asymptotically, both of which address the limitations highlighted by Mikhael/Moller/Bogacz that resulted from prolonged experience, while retaining the Hebbian nonlinearity (again, needed for the model’s adaptive contributions and for capturing various empirical data). See Author response image 4, which averages G/N learning curves over 1000 trials (as requested) from the model of original submission (actor learning rate = .5, annealing parameter T = 10):

We also confirmed that although the learning rates annealed with time, it is not the case that the weights “simply stop changing” altogether – indeed, while the Möller example produced weights that converge to 0 within 20 trials, our simulations in the original manuscript showed that learning continued to improve over 100 trials; we had noted parenthetically that similar advantages hold for multiple hundreds of trials, but we now show this explicitly.

**Author response image 4. sa2fig4:** 

We also confirmed that our updated OpAL* annealing mechanism in the new submission also stabilizes actor weights and preferences over prolonged periods. Simulations in the main text were previously for 100 trials but because multiple reviewers were interested in longer asymptotic behavior we have now presented those simulations for 250 trials. See Appendix 1—figure 4 average learning curves for the updated model over 1000 trials, which includes bayesian annealing (critic learning rate = .05, actor learning rate = .5, annealing parameter T = 10). Panels show G weights, N weights, and net Actor values for the same parametrically varying reward probabilities as above.

Adaptivity

Nevertheless, we take the reviewer’s point that a simple reduction of learning rate as a function of time is too limiting, and a constant progressive annealing mechanism would eventually impede adaptivity to change points. Indeed, we have previously focused on this stability/flexibility tradeoff in the basal ganglia, showing that it can be optimized by incorporating the role of striatal cholinergic signaling, which gates plasticity induced by dopaminergic RPEs (Franklin and Frank, 2015, *eLife*; see https://www.nature.com/articles/s41467-022-28950-0 for recent evidence of this mechanism). Our model showed that this mechanism can be adaptive when cholinergic signaling is itself modulated by uncertainty across the population of striatal MSNs, stabilizing learning when task contingencies are stationary but increasing adaptivity when they change, mimicking a Bayesian learner. Accordingly, we have now adjusted the OpAL* annealing mechanism to better approximate this mechanism: rather than annealing with time, we now anneal as a function of uncertainty within the Bayesian “meta-critic” (which evaluates the reward statistics of the policy/environment). As the meta-critic becomes more certain about the reward statistics, actor learning rate declines, consistent with Bayesian learning. This annealing is also more adaptive in that when task statistics change, the uncertainty rises and learning rate increases again (see Franklin and Frank for a more thorough investigation).

To demonstrate the impact of this new annealing mechanism on adaptivity, consider the following switch points in reward contingencies. In Author response image 5, action A is rewarded 80% of the time and action B is rewarded 50% of the time. After 150 trials the reward contingencies flip. Plots show the softmax probability of selecting action A. On the left is our initial submission with fixed annealing. To the right is our new submission with Bayesian annealing.

**Author response image 5. sa2fig5:** 

This flexibility can be further augmented by incorporating decay in the actor weights as proposed by Franklin and Frank, also allowing OpAL* to handle highly volatile environments such as random walks in reward probability. Author response image 6, the left panel shows random fluctuations in reward contingencies assigned to an action; middle and right the G/N weights of OpAL and OpAL* (with bayesian annealing and decay), respectively. OpAL* actor G/N weights closely follow the random reward schedule. These parameters could be further optimized, but we wished to keep the model in the current paper as simple as possible to equate parameters with competitor models, and switch points are not the focus here. Moreover it is known that more rapid switching involves other brain mechanisms not modeled here, such as prefrontal cortex and hippocampus.

**Author response image 6. sa2fig6:** 

We hope these simulations and discussion clarifies that these issues are addressed.

Finally, while not the main basis for the original *eLife* decision, multiple Reviewers noted that it would be important to provide a clearer explanation of the mechanism for how OpAL* improves performance relative to Q learning and the Moeller and Bogacz model (as well as others). We have now added a much more detailed consideration of this issue, conceptually, analytically, and via simulations, which we unpack here before turning to the point-by-point rebuttal.

The key issue is that the two classes of models have different normative aims. Möller’s model proves that G and N weights can converge to the expected payoffs and costs of an action, much like a Q value converges to the net expected return. But in both cases, this proof is asymptotic, with no guarantees on how long it takes for such convergence to occur, which can be prolonged when an agent is forced to select among multiple actions to learn their expected outcomes. In RL terms, as a modified actor critic, the OpAL* model is more like a *policy gradient* mechanism that directly optimizes its policy to maximize the return, while obtaining a rapid estimate of the relative *ranking* among alternative actions. Indeed, actor-critics are policy gradient algorithms and neuromodulated activity-dependent plasticity rules such as REINFORCE also implement policy gradients. OpAL* explicitly leverages activity-dependent plasticity together with opponency, and capitalizes on the dynamic DA modulation thereof. In contrast, Q learning and Moller/Bogacz models obtain long-run estimates of the expected values of these actions. It is widely known that policy gradient methods in RL are more suited for environments with large action spaces, consistent with our finding that OpAL* advantages grow with the number of actions, and that Q learning models suffer from reduced “action-gaps” (differences in Q values between the best and next best option) in these environments.

To unpack this more clearly, we now show that both Q learning and Moller’s model exhibit delays in convergence when selecting among multiple actions, and reduced action gaps, particularly in environments with sparse reward – specifically the setting in which OpAL* advantages are most prominent. We also further elaborate the mechanism by which OpAL* actor weights can quickly discriminate between actions before the Q learning critic converges. We do this (1) via additional simulations directly comparing the evolution of OpAL* weights to those of alternatives for a fixed policy and sequence of outcomes; and (2) analytically we unpack how the OpAL* learning rule allows weights to be updated as a function of the entire history of previous RPEs and is thus less sensitive to spurious outcomes. We also now (3) show that this same learning rule accounts for counterintuitive choice preferences in a new simulated task, as requested by R2.

However, we do not challenge the notion that it would be useful to obtain estimates of expected costs and benefits explicitly, and it remains possible that the Moller and Bogacz mechanism, or some adaptation thereof, may apply to some aspects of BG function. Indeed we also note that OpAL* still retains access to a Q learning critic which could be leveraged.

In sum, the central message of our paper is that although the Hebbian mechanism within OpAL* does not produce asymptotic estimates of ground truth expected action cost and benefits, it (i) allows for more rapid assessment of the relative rankings of costs and benefits, (ii) affords an adaptive choice algorithm that is robust to task contingencies, (iii) capitalizes on knowledge about the inferred richness of the environment, (iv) produces benefits that scale monotonically with number of actions in the task, and (v) can account for several datasets across species that can also rationalize the findings that are pathological on the extreme, in a way that is not accommodated by other models.

Reviewer #1 (Recommendations for the authors):The authors aim to demonstrate the normative advantage of opponent basal ganglia pathways, using dynamic dopamine (DA) modulation by environmental reward statistics. The finding that well-known neurophysiological and psychopharmacological mechanisms yield rational decision-making in ecologically plausible environments is a major strength of the paper. The methods are rigorous, and the advantage of the model is reliable across a range of parameters and paradigms. More could be done to explain the optimality of certain model predictions, to map the correspondence between model predictions and animal behavior, and to articulate the conceptual relationship between relevant constructs, such as exploration, discrimination, and risk-taking. Nevertheless, the scope of the model, its biological plausibility, and its normative and descriptive aspects suggest that it may have a significant impact.1. More could be done to detail the, sometimes counterintuitive, optimality of certain model predictions. For example, while it is straightforward that you need to explore to discover better options, the need to sample sub-optimal options so that you know to avoid them makes less sense: if you don't sample them, you are already avoiding them, and when you do choose them, you learn the values – why is it adventitious to do the sampling early in the process as opposed to later? Another example is the claimed optimality of avoiding gambles in lean environments, despite the expected value of the gamble being greater than that of the ST.

Thank you for this comment and opportunity to further explain the adaptive nature of the choice mechanism for learning. We agree the mechanism is counterintuitive and we have expanded the Mechanism section to more clearly expose it. We focus here on the sparse reward (lean) conditions, in which the advantages of OpAL* are more clear. The key issue is that in “vanilla” RL, when the optimal action produces few rewards, its value declines, causing the agent to switch away from it toward sub-optimal actions. The agent will eventually choose the sub-optimal actions enough to learn they are even worse and switch back to the optimal action. The net result is more stochastic switching between actions during early learning, especially in lean environments. This effect is amplified as the number of actions increases. Thus it is indeed true that “when you do choose the [suboptimal actions], you learn the values”, but learning the values of all suboptimal actions requires repeated choices of each of these actions, leading to suboptimal performance.

In OpAL*, once the meta-critic detects it is in a sparse reward environment, dynamic DA modulation allows it to rely preferentially on its N actor. Initial exploration of sub-optimal actions early allows the Hebbian nonlinearity to amplify those weights sufficiently so that they are avoided robustly, producing less switching (retaining the ability to switch if change points are encountered). Thus while a Q learner may, after much more experience, produce more robust estimates of action values for all actions, OpAL* instead optimizes how quickly the agent can *rank* the best action over the others.

To help communicate this counterintuitive point, we have now expanded our mechanism section and have included exemplar simulations and analysis. These show that Q learning (and related agents that rely on expected values of alternative actions) show impeded convergence of those values when making choices, unless they receive full feedback not only on the rewards of the actions they chose but also those they didn’t choose; (ii) this results in a reduced “action-gap” (difference between Q values for optimal and second best option), specifically in sparse reward environments; (iii) this is accompanied by more stochastic switching. We then show (analytically and via simulations) that OpAL* circumvents this issue because its nonlinear update rule allows the actor weights to be sensitive to not just the most recent RPE, but the history of such RPEs, thereby accentuating its ability to discriminate between different reward probabilities in either the high (G) or low (N) range. The dynamic DA modulation based on the meta-critic can then guide action selection according to the actor most specialized for the environment (efficient coding).

2. It would be helpful if there was a description, early in the Introduction, and perhaps again in the Discussion, of how the relevant constructs (i.e., exploration, discrimination, and risk-taking), are related conceptually – note that exploration can be characterized as risk-taking and that the decision to gamble may reflect either exploration or value-based discrimination.

We agree that these terms require clarifying as they can carry multiple meanings. We have included a description of the scope and definitions of terms that we explore in the introduction.

3. In Figure 10a, it is unclear why L-DOPA does not boost G weights and reduce N weights, yielding a greater probability of gambling, on loss trials as it does on gain trials?

In our simulations, L-DOPA amplifies endogenous changes in dopamine: because it enhances presynaptic DA synthesis, we modeled it as boosting the phasic signal that would have occurred naturally (see also Collins and Frank 2014). According to OpAL*, only gambles whose net expected values are larger than average will induce an increase in DA levels during choice, and it is these transient increases that are then further amplified by L-DOPA, leading to the selective change in gambling in gain gambles.

4. In Equations 1-3, why not show the updating of Act values, or, if identical to V(a), be explicit about that. Also, Equation 14 is referred to as a prediction error in the text, but the notation indicates an action value [i.e., V(a)].

Thank you for this feedback. The definition for Act was defined later in the section, which is an unhelpful ordering. We therefore have switched the order of this section for clarity and introduce the softmax equation only after properly defining Act.

5. It is argued that OpAL* eliminates the need for a priori knowledge about environmental reward statistics, but DA-modulation depends on confident estimates of the values (i.e., reward probabilities) of all available actions – the implications of this reliance on reward estimates seem under-explored in simulations.

The critical distinction is that OpAL* does not need advance knowledge about reward statistics to select its hyper parameters – the DA modulation only occurs once the agent has accrued enough information about reward statistics in the current environment, at which point it can leverage the nonlinearity in the G/N weights to improve its policy. Furthermore, to clarify, the confidence estimates of reward statistics needed for DA modulation concern the *overall* environmental richness (in the “meta-critic”), NOT the accuracy of action values. In this sense, the explore-exploit dilemma here is one in which uncertainty is used to first guide more stochastic action selection according to both actors, and then transition to exploiting the *actor* more adept for the task, without specifying which *action* should be selected per se. In our simulations, for simplicity we fixed the degree of confidence required before modulating DA (phi) to be 1.0 (i.e. one standard deviation). In theory this parameter could also be optimized to improve performance further, but that would make the agent more complex and more difficult to compare with alternatives.

6. It is unclear from the curves in Figure 7 whether the models reach equilibrium by 100 trials. Please show asymptotic performance.

We now have added simulations up to 1000 trials, with a minimum of 250 trials, showing asymptotic performance. All performance advantages hold in these new simulations.

7. Why use only forced-choice simulations? It is hard to imagine a real-world scenario in which "no action" is not an alternative. Failure to include such an option detracts from the ecological validity of the experiments.

Previous work with the OpAL model explored how the model may account for “no action” scenarios in effort based decisions, by including a threshold to compare Act, i.e. benefits must outweigh costs by a sufficient amount for actions to be emitted. Specifically, Collins and Frank demonstrated OpAL’s architecture captures impacts of DA depletion on effort base choice (e.g., work by Salamone) and number of actions emitted, and how they interact with learning. This model also simulated the impacts of DA depletion on learned catalepsy (avoidance of actions that increase with time in DA depletion), as in Beeler el al, 2012 and Wiecki et al., 2009. (Notably those effects depend on the Hebbian term to manifest, which is pathological when DA is depleted; our point here is that this same mechanism is adaptive under healthy endogenous DA). While we do not reproduce those effects here, we do explore gambling simulations which are formulated such that the model is learning whether or not to select a gamble, which is encoded relative to the sure thing. That is, the agent is learning whether to take an action (e.g. select the gamble) or not take an action (and receive some default). The default can be defined as any desired threshold of value below which alternatives are not selected.

Reviewer #2 (Recommendations for the authors):I really enjoyed this paper that significantly advance our understanding of the basal ganglia circuit. I have some suggestions for additional simulations that can allow establishing to which extent previous findings in reinforcement learning can be accounted for by the OpAL(*) models.A key concept of the model is its sensitivity to environmental richness (i.e., whether the agent is in a lean or in a rich environment). The concept behind the "rho" variable is very closely related to that of "state value" or "reference point" as it has been applied to reinforcement learning and valuation since Palminteri et al. (2015). The key demonstration of a crucial role of overall "environmental richness" in learning and valuation came from a task, which, explicitly coupling features of the Frank (2004) and Pessiglione (2006) tasks, showed that transfer learning performance is highly context-dependent (see Palminteri and Lebreton, 2021; Daw and Hunter, 2021 for review of these results). My question here is whether or not the OpAL(*) models are sufficient to generate such context-dependent preferences. Does this (very robust) behavioral effect naturally emerge from the model? Or do we need to specify an additional process to explain this? Can you simulate Palminteri et al. (2015) using the OpAL(*) models and show what are the performance in the learning and transfer phase?

We are delighted to hear the paper was enjoyable and we thank the reviewer for their feedback. We agree that the connection of our work to the reference point literature cited is quite clear. We have now included simulations of the Palminteri et al. (2015) task as an additional example of counter-intuitive experimental data, which OpAL* can indeed capture. This is accomplished when the critic considers context-value rather than action value, so that the less frequently rewarded stimuli (gain 25) induce negative RPEs relative to what could have been obtained had the other action been selected (this is justified given that full information was provided to participants in this condition). As such the N weights grow for Gain 25, and conversely the G weights grow for the more frequent loss avoider (loss 25). In principle, one could also directly use the meta-critic tracking state value as formulated currently to then influence the Q-learning critic’s, akin to the model proposed by Palminteri. We considered the context-value critic as a more parsimonious model within the OpAL* framework, and therefore now feature it in our Results section.

A robust and replicable finding in human (and nonhuman) reinforcement learning is that when fitting the model with two learning rates for positive or negative prediction errors (starting from Frank et al. 2007, supplementary materials, continuing in Lefebvre et al. 2017; Gagne et al. 2020; Farashahi et al. 2019; Ohta et al. 2021; Chambon et al. 2020 – the rare occurrences of the opposite pattern are generally explained by wrongly initializing the Q-values at pessimistic values). The optimality of this bias has been sometimes investigated (see Cazé and Van den Merr 2013; Lefebvre et al. 2022). My question here is whether and/or under which circumstances do the OpAL(*) models explain the ubiquity of this pattern? If you simulate OpAL(*) models and then fit the asymmetric model, would you (quasi-)systematically retrieve the asymmetric pattern?

We agree that it is also worthwhile to consider the optimality of learning asymmetries, which could be manifest by changes in learning rates from positive vs negative RPES as in the above studies, or in OpAL* it could be considered as imbalances in learning rates within G and N actors. Our focus in the present manuscript is on how dynamic DA optimizes the policy during decision making, and show that this can impact risky choice and learning without requiring additional asymmetries in learning rates per se. Moreover, in the original OpAL paper, Collins and Frank showed that empirically one can reproduce empirical patterns resulting from DA manipulations on learning from positive vs negative RPEs in a more parsimonious and monotonic fashion when simulating asymmetries in alphaG vs alphaN instead of positive vs negative RPEs (see Figure 8 and Figure A1 in the appendix of that paper for comparison).

While we could additionally explore the optimality of asymmetries in learning rates, as per the above studies, this would again add further complexity and degrees of freedom. Moreover, the manuscript has become quite extensive (including many normative comparisons, simulations and analysis to unpack the mechanism, robustness to stability and adaptivity, and simulations of 6 different empirical datasets across species). We therefore prefer to maintain the focus here.

Finally, the actor prediction errors are normalized. I can see the logic of this. I was nonetheless puzzled by the functional form of the normalization that is not really a range normalization (Lmag is lacking at the numerator). Consider for example Bavard et al. (2021), which propose a full range normalization rule. What is the rationale for this particular form of normalization? Are Lmag and Rmag learned (via δ rule or other, see Bavard et al.) or specified in advance?

The Rmag and Lmag are specified in advance, but in theory can be learned via a δ rule. The specific formulation was designed to constrain prediction errors within the range of 1 and -1, which absolutely can be achieved with alternative formulations. Our results are not dependent on this exact formulation; indeed, the vast majority of our main results utilize a reward magnitude of 1 and a loss magnitude of 0, so normalization as proposed by Bavard et al. would generate the same results. In either case, the effect of normalization is negligible as it reduces to dividing by one. We also note that the PE normalization we used is that projected to the ACTORS, not that is used by the critic; that is, rather than normalizing the reward directly, we are modifying the RPE signal used to learn a policy. As we now further unpack in the Mechanism section, the normative contribution of OpAL* concerns the actor policy and not accurately estimating values. Because the actors also learn via a three factor hebbian term where high G/N weights cause excess variance, we therefore feel it reasonable to modify the RPE used by the actors only. For our simulations that do include various magnitudes (the gambling simulations), G and N weights are set explicitly and therefore no normalization applies. The sole exception is the simulations proposed by Moller and Bogacz (which prompted the inclusion of normalization) that demonstrate rapid decay with rewards of 2 and losses of -1. Here, again, either normalization formulation is sufficient and helps stabilize the G and N weights. As our primary focus was to characterize flexibility of OpAL*, we believe the proposed normalization is sufficient to demonstrate how normalization in general may be applied to stabilize OpAL* weights but we leave the exact formulation (and the learning of its Rmag and Lmag parameters) for future work.

Reviewer #3 (Recommendations for the authors):This manuscript proposes a refined version of the OpAL model, which aims to address the numerical instability problems present in the previous version and demonstrate the normative advantages of the model in learning tasks. Unfortunately, the instability problems still persist in the proposed model, and the normative comparison is unconvincing due to the limited range of scenarios and tasks investigated.Comments1. It has been previously pointed out that the OpAL model is numerically unstable: all synaptic weights converge to 0 with learning, and as a result, the model has a tendency to make random choices with practice. The authors claim that the proposed model "addresses limitations of the original OpAL", but in fact, it does not – it just postpones their effect by introducing learning rates that decay with time, which ensures that the weights simply stop changing as learning progresses. Consequently, there are several problems with the proposed argumentation in the paper:

Please see introduction to this response, where we have extensively considered these issues. While we did not fully agree with the characterization, we do appreciate that the reviewer forced us to confront these issues more thoroughly and improve our model using Bayesian annealing.

a. The paper does not adequately describe the scale of the stability problem faced by the original OpAL model. A cartoon in Figure 1 falsely suggests that the weights in the OpAL model converge to stable values with learning, by contrast even for this scenario all weights decay to 0. The appendix attached below this review contains Matlab code simulating the OpAL model, and the first figure generated by this code shows OpAL model simulations corresponding to Figure 1. Running the code clearly shows that all weights decay to 0 in this simulation. Therefore, weight evolution in the OpAL model in Figure 1, needs to show the actual simulation of the OpAL model rather than cartoons with qualitatively different behaviour of the weights.

Figure 1 came from the original OpAL paper and actual simulations (using low fixed learning rates and initializing G/N weights to 1). The decay to 0 would happen within that model with much longer time horizons. We now show explicitly that weights do not decay to 0 (even in the original OpAL model, but especially with the new model). As noted above the simulations provided here (which were the basis of rejecting the paper) were from the previous model, not from the new one! Much time was invested in adapting the model and simulating the various scenarios described in this paper and we feel this comment was overly dismissive, particularly as the normative advantages we showed clearly would not have held had the weights decayed to zero so quickly. Nevertheless, again as noted in the introduction we have now improved the annealing mechanism further so that it is sensitive to uncertainty and retains adaptivity, while still providing stable ranked estimates of actor weights.

b. Similarly, a reader may get a false impression that the instability problems only occur in the scenario in Figure 13, because the manuscript states in lines 1104-1105 that the OpAL model "in carefully constructed situations, gives rise to unstable actor dynamics". However, as pointed out in my previous comment, these problems are ubiquitous, and in fact, Mikhael and Bogacz (2016) also illustrated this problem in Figure 9 of their paper for randomly generated rewards. The reason why Moeller and Bogacz (2019) considered the scenario in Figure 13 is that for this case one can easily show analytically that the weights in the OpAL model converge to 0.

The Reviewer is correct that the stability issue from the original OpAL model goes beyond that of the Moeller and Bogacz scenario; we meant that this situation was carefully constructed to show a particularly pernicious version of it, given that the decay happened within 20 trials of experience. This is why we addressed that particularly problematic scenario with the new model and showed that OpAL* preserves rankings of G/N weights for 1000 trials (as per Appendix 1—figure 4 we now adapt it further to address your other comments). In more canonical reward probability learning scenarios, even the original OpAL doesn’t decay until much later (unless fast learning rates or initial weight values < 1 are used). Moreover, in Author response image 7 you can see that despite the decay, even at 1000 trials the rankings of the actor values are preserved, with the dominant actor still showing a large action gap for high (G) or low (N) reward probabilities.

**Author response image 7. sa2fig7:** 

Original OpAL model, average G/N/Act over 100 simulations, critic learning rate = .05, actor learning rate = .1

c. To solve the stability problem, the authors modify the model such that the learning slows with time. However, they admit in l.1130: that the model loses its memory with time, and they propose that the habit system may take over learning and rescue animal bahaviour. However, animals with lesions of dorso-lateral striatum (which is known to underlie habitual behaviour) still can perform learned tasks (Yin HH, Knowlton BJ, Balleine BW. 2004. Lesions of dorsolateral striatum preserve outcome expectancy but disrupt habit formation in instrumental learning. European Journal of Neuroscience 19:181-189.) which is not consistent with the decay of memory of goal directed system.

As noted above our newer model addresses this issue via Bayesian annealing. Moreover, although some researchers consider the DLS to be the seat of habitual behavior, several neural network models suggest that striatum is only needed to *learn* the habit, but that automatized behavior is supported directly by Hebbian stimulus-response learning from sensory to motor cortex. Please see Beiser Hua and Houk 1996, Ashby et al. (“SPEED model” focusing on just this), as well as neural net models preceding OpAL by our group, Frank 2005, Frank and Claus 2006, Ratcliff and Frank 2012 (see http://ski.clps.brown.edu/BGmodel_movies.html for an animation of this; search ‘habit’). The finding that animals with DLS lesions still perform learned tasks is precisely what is expected from these models, which propose that the BG is only needed for gating motor responses that the cortex can’t resolve without a disinhibitory boost. Indeed Parkinson’s patients can famously execute very well learned skills but not novel ones, and cats with striatal lesions only have trouble converting a learned reward value into action but not executing familiar actions. Nevertheless, as we now unpack below and in the manuscript, it is also possible that the agent could make use of the critic value for action selection once it is converged, even if the actor weights were to decay (we do not leverage this mechanism in the manuscript).

d. The reduction in learning rate with time introduced in OpAL* only makes sense in stable environments. In changing environments decaying learning rate does not make sense, because the animal would be unable to adapt to changing rewards. However, the second figure generated by the code in the Appendix shows that the weights also decrease to 0 in environments where the reward probability constantly fluctuates according to a random walk (in this simulation the rewards are binary, so the normalization of prediction error introduced in OpAL* does not make any difference). In summary, the learning rules of the OpAL model decay to 0 even if the reward probability is constantly changing, so for this case the model is unable to track reward probability no matter if the decrease of learning rate is introduced or not.

Thank you for this point, which motivated us to improve our model. As mentioned in the introduction to this letter, OpAL can account for this varying reward giving a slower critic and actor learning rates, though exhibits mild decay. But we agree that annealing will indeed decrease an agent’s responsiveness to change points in task contingencies, and as such, we now prescribe our annealing as a function of uncertainty of the meta-critic to remain sensitive to switch points in the environment (see more details above). We also highlight above in the Adaptivity section how further augmenting OpAL* with weight decay, as proposed by Franklin and Frank, allows OpAL* to flexibly track highly volatile environments such as random walks of rewards. We again note that changing the parameters of the original OpAL model such that the critic learning rate and actor learning rates are smaller (0.05 and.1, respectively) allows even the original OpAL to adequately follow random walks in rewards with milder decay.

2. The manuscript shows that the OpAL* model can achieve higher rewards than alternative models in simple tasks with constant reward probabilities. However, there are severe problems with the arguments in the paper.a. All the simulations are run with a small number of trials (i.e. 100). Beyond this number of trials the OpAL model suffers from decay of weights (see my previous comment), so it is questionable whether it would have performance advantages. Therefore, it is critically important that the performance comparison includes simulations with more (e.g. 1000) trials.

As requested, we have now modified our explorations so that the minimum number of trials used is 250, but we also show advantages for up to 1000 trials. Moreover, while we focus on these simple reward probability tasks for the normative advantages, the general issue we highlight with sparse reward and large action spaces is appreciated in the RL literature, and thus the key points that we address should in principle scale to those situations involving model-free RL. We also simulate several other task set ups to compare our model to empirical data, and we do not know of other models that can capture this range of data.

b. The simple tasks simulated in the paper has been a focus of much research. There is known optimal learning algorithm (Gittins index) and several well performing ones (e.g. Thomson sampling, Upper confidence bound algorithms). I feel that the normative comparison needs to include a comparison with some of these known solutions.

Because the Gittins index does not scale to other scenarios, and requires explicit computation not typically considered in the realm of model-free RL, we focus instead on the upper confidence bound heuristic that is featured in both biological and machine learning, and which does not require a posterior distribution to sample from. We now include UCB in our models for comparison and demonstrate that OpAL* exceeds or achieves comparable performance for various time horizons, including 1000 trials in the most difficult environment we explored (lean with 6 options). We unpack this in the manuscript.

c. The reason why the OpAL* achieves best performance in the chosen scenario is not well explained. Lines 339-343 present the key of the mechanism: "opponency allows the non-dominant (here, G) actor to contribute early during learning (before N weights accumulate), thereby flattening initial discrimination and enhancing exploration. Second, the Hebbian nonlinearity ensures that negative experiences induce disproportional distortions in N weights for the most suboptimal actions after they have been explored (Figure 6a), thereby allowing the agent to more robustly avoid them (Figures 6b and 6c)." However, the cited figures do not illustrate this mechanism, and to show this, it would be better to show how the weights change during learning.

In response to your comments and that of R1, we have expanded our mechanism section, including explicit demonstration of how the N weights evolve over learning and compare relative to Q learner and a non hebbian alternative (equivalent of vanilla actor critic model). Please see details above and in the Mechanism section.

d. The performance of model is compared in two ways: (i) for the best parameters for each model and (ii) for an average over the range of parameters. I think that method (i) is valid, but method (ii) may introduce a bias if the range is chosen such that the optimal range for one model overlaps with the tested range for one model more than for another. Therefore, I feel that method (ii) should not be included in the manuscript.

A critical feature of OpAL* (and central point of the paper) is that it enhances performance across a range of hyper parameter settings, without requiring the model to know which parameters are optimal. We interpret this finding as evidence for OpAL*’s flexibility when the statistics of a novel environment are unknown or incorrectly assumed. Given the divergent explore/exploit trade-offs across environments (we use rich and lean here as extreme examples), OpAL*’s ability to adapt quickly and robustly is a key characteristic. Therefore, we felt that including performance over a range of parameters as well as the optimal performance is informative. Nevertheless, we ensured that the optimal cross-environment parameters for comparison models fell within the specified range and conducted additional analysis verifying that our conclusions held when only considering the top 10% performing parameters for each agent in each environment. That is, a Q learner may have high performance for one set of parameters in one environment, but when that parameter set is used in another environment performance suffers.

3. The manuscript also describes comparison of performance of OpAL* with the model by Moeller and Bogacz (2019), however the manuscript includes incorrect statements about the latter model, and the way it is parameters are chosen does not reflect the conditions in simulated scenarios.a. The manuscript states in l. 392: "However, in actuality, the convergence to expected payoffs and costs in this model depends on having a constrained relationship between parameters optimized by a priori access to the distributions of rewards in the environment." This is not true. For a given set of parameters, the convergence to payoffs and costs is guaranteed for any reward distribution – derivation of the condition for encoding payoff and costs in Eq 22 and 23 in Moeller and Bogacz does not make any assumptions about knowledge of reward distribution. Please remove this sentence.

We agree that we incorrectly summarized the limitation of Moeller and Bogacz, and we have removed this sentence (we did see performance advantages relative to that model and wrongly assumed that it must be due to parameter specification). As noted in the Introduction to this letter and response to R1, the Reviewers’ comments have motivated us to better unpack the source of the differences between models. We thus now focus on the difference in the scope / aims of these models in context of the larger RL literature regarding models that learn expected values vs those that optimize policies. Given the Moeller and Bogacz model falls in the former class, we show that it faces similar explore/exploit, convergence, and action-gap difficulties in the lean environment akin to Q-learning. We further demonstrated in this manuscript revision how stochastic sampling causes misestimations in convergence, akin to Q-learning, with differing patterns across rich and lean environments. We then expose how the OpAL* mechanism can speed learning in lean environments with multiple actions, via new simulations and analysis (see response to R1 and the new manuscript under “Mechanism”).

b. The condition on parameters derived by Moeller and Bogacz are only necessary to learn payoffs and costs if every trial includes a cost (e.g. an effort to make an action) and a payoff (e.g. the outcome of the action). This is not the case in the presented simulations, so the model parameters do not even need to satisfy the conditions of Moeller and Bogacz.

We now explicitly expose the Moeller and Bogacz model to tasks with explicit payoffs and costs, similar to the probabilistic one-armed bandit of Figure 5c of their paper.

c. The model is simulated with constant "tonic dopamine" level, and by mathematical construction, in this case the models described by Moeller and Bogacz is mathematically equivalent to Q-learning with decay controlled by parameter λ, and will reduce to the standard Q-learning for λ = 0. Since the Authors simulated Q-learning, it is not even clear if it is necessary to simulate Moeller and Bogacz model because by its definition it will have identical (with λ=0) or very similar performance.

We agree with the authors that there is overlap between the performance predictions of Q learning and the Moeller and Bogacz model, nonetheless we find it informative to include both in our paper to highly that OpAL*’s advantages are not only a product of opponency but also Hebbian nonlinearity. The Moeller and Bogacz model provides a useful comparison in this light.

d. The manuscript states in l. 415: "Finally, the model in Möller and Bogacz (2019) demonstrated poor across-environment performance, performing only slightly above chance in the rich environment. Results are not shown for this model" – one should not make such statements without actually presenting evidence for them.

We have since discovered in our initial submission that the value of β was not properly being applied in the softmax choice function of this model for certain simulations. Therefore, the model, particularly in high complexity environments, was performing near chance as it was not able to effectively exploit its learning. We have since corrected this bug, and removed this sentence, although the general result remains re: OpAL* advantages across environments.

[Editors’ note: what follows is the authors’ response to the second round of review.]

The manuscript has been improved but there are some remaining issues that need to be addressed, as outlined below:Reviewer #3 (Recommendations for the authors):This manuscript proposes a refined version of the OpAL model, which aims to address the numerical instability problems present in the previous version, and demonstrate normative advantages of the model in tasks involving balancing exploration and exploitation.I thank the Authors for replying to my review. I feel that the manuscript has been significantly improved, and the comparison of performance with UCB is particularly interesting. Nevertheless, there are still issues that need to be addressed. In particular, it is not clear from the analysis why OpAL* can outperform UCB, and multiple statements about the stability of the model are still misleading.Comments:1. It needs to be further clarified why OpAL* outperforms UCB and Q-learning. I have to admit that I am surprised by the higher performance of OpAL* over UCB, because UCB is not an easy algorithm to outperform. Unlike Q-learning, UCB has perfect memory of all rewards and does not forget them. Then I realized that OpAL* also has such perfect memory, as its Bayesian critic counts the rewarded and unrewarded trials for each option. I feel this is the main reason why OpAL* outperform Q-learning. I suggest explaining this important property of OpAL* in the text, and it would be good to test it, e.g. by replacing Bayesian critic by a normal forgetful critic, and testing "Contribution of perfect memory". Please also discuss if such perfect memory is biologically realistic – how could it be implemented in biological neural network?

Thanks for encouraging us to further clarify this issue. In fact, the revised OpAL* uses a standard critic with a fixed learning rate and does not have access to the full sequence of rewards. Only the “meta-critic”, which evaluates whether the environment as a whole is rich or lean, is Bayesian, providing it a measure of confidence (uncertainty) in the richness of the environment before it modulates dopamine – but crucially, it does not represent Bayesian values for each action and cannot use its estimation for action selection. Regarding biological mechanism, as we discuss in the manuscript (beginning line 236), we envision the meta-critic to involve prefrontal regions that represent such reward statistics / task state and in turn modulate striatal dopamine via projections to cholinergic interneurons. There are several computational papers that suggest full posterior distributions may be represented in biological neural networks (e.g., Ma et al. 2006 and 2008, Pecevski and Maass 2016, Deneve 2008, Deneve et al. 1999), and in this case the computational demands are simpler due to the use of a β distribution (which simply requires incrementing two counts, from which the mean and variance can be easily accessed). Moreover, while we use a Bayesian implementation of the meta-critic because it has a principled measure of certainty, this is likely not essential for the advantages we report: indeed, we only use the posterior distribution in a rudimentary way – we simply imposed an arbitrary threshold for the amount of uncertainty (precisely 1 standard deviation) tolerated before dopamine is modulated, and we did not adjust this hyperparameter. Thus many other non-Bayesian heuristic approximations would behave very similarly, including a traditional RL critic that simply evaluates whether its estimate of environmental richness is above or below 0.5 for multiple trials in a row before modulating dopamine. In sum, the perfect memory, at least in this context, is a red herring.

However, as we emphasized in the previous submission and the current one (Equation 24 in the Mechanism section), OpAL*’s actor learning rule implicitly does contain information about the entire history of reward (prediction errors) due to its recursive update rule. The analytic expansion showed that each weight update is influenced not only by the current RPE (as in standard RL) but also by each of the previous RPEs with equal weight (ie scaled by α) and their higher order interactions (scaled by α^2 and so on). This is what enables the nonlinear actors to accentuate differences in reward statistics within rich or lean environments and to exploit the best option (this is perhaps best demonstrated by the fixed policy simulations in Figure 8c).

However, the most surprising result is that OpAL* outperforms UCB. This cannot be explained by most of the discussion of Mechanism section focussing on the gaps between weights for different options, because such gaps are only important if an algorithm choses actions stochastically based on weights, while UCB is practically a deterministic algorithm (beyond initial few trials it will deterministically chose an option). The only mechanism which can explain outperforming UCB is explained in a paragraph starting in line 574. Please investigate further, and provide a clear explanation for how it is possible for OpAL* to outperform UCB.

We agree the advantage over UCB is striking, given its access to the sample mean of each option and a strategic exploration strategy. We have now further clarified (via simulations and text) why UCB still suffers in policy performance as the number of actions grows. While UCB is deterministic, the sample mean that it obtains for a given action shows impeded performance due to (i) partial sampling (first and third, column Figure 8—figure supplement 2) and (ii) the exploration bonus (second and fourth column, Figure 8—figure supplement 2). The partial sampling effect on convergence is similar to Q-learning, i.e, the precise expected value of an action is only accurate after it has been chosen sufficiently often, thus requiring the agent to choose all actions sufficiently often to converge. The exploration bonus is aimed at ensuring the agent revisits actions for which its values are not sufficiently well estimated, and in a partial sampling environment, this continues to impact UCB policy even at 1000 trials (RICH/LEAN Q+BONUS vs. RICH/LEAN, second and fourth column). Both the convergence and exploratory bonus issues are further amplified in higher complexity (with more actions). Thus while choice is deterministic on a single trial, the UCB agent still switches between actions across trials and thus vacillates in its policy.

In contrast, OpAL* progressively becomes more deterministic in its policy. The nonlinear update rule is sensitive to the full sequence of RPEs, and once it sufficiently discriminates between the options, it can exploit the best one, which progressively increases the action gap and dynamic dopamine modulation then further amplifies this action gap. This can be seen in the variance of the policy of UCB compared to OpAL* in Figure 8b.

Lastly, while we examine optimal learning curves to illustrate differences in performance across agents, our main focus is on the robustness of performance across parameter settings. UCB performance is highly sensitive to its exploration parameter, evidenced by the decreasing trend in its histograms (Figure 6b). To further illustrate this point, in (Author response image 8) we plot the UCB top-performing parameters by cross-environment performance. We see that UCB has a narrow range in which its performance is optimized and moreover, the best-performing exploration bonus parameter differs according to the complexity of the environment. In particular, as the number of actions grows, the UCB agent has to reduce its exploration bonus in order to achieve greater performance, because this bonus serves to encourage excessive exploration that impedes the policy.

**Author response image 8. sa2fig8:** 

We have updated our text to reflect this discussion and included the convergence figures in our supplementary section.

2. The stability of OpAL* model needs to be honestly presented. The main issue pointed by Mikhael and Bogacz (2016) and then by Moeller and Bogacz (2019) is that the weights in OpAL will asymptotically converge to 0. It is still evident from simulations of the OpAL* in the manuscript and in the response letter that on average the weights decay with trials. There is no demonstration that OpAL* can prevent its weights from converging to 0 eventually, and at the same time can respond to changing rewards. In the revised version, such demonstration has been added in Figures 15c right and 15d right, but it is not possible to understand how these simulations were performed from the paper, and it seems that they are for a model with "weight decay", which is not described in the manuscript. It is not clear if the model with weight decay has the advantages of OpAL* in exploration/exploitation that are the focus of the manuscript. Therefore, I feel that the manuscript has to be modified in one of two ways: The Authors may change the model in the paper to one with weight decay and analyse its performance in exploration/exploitation task. Alternatively, if the current OpAL* model remains the focus of the paper, it needs to be honestly admitted that the OpAL* model suffers from the problem that weights will eventually converge to 0 or the model will stop adapting to changes in the rewards.

We of course agree with the reviewer that (in general), increased stability can trade off with flexibility. This is not surprising as stability and flexibility naturally trade-off more broadly in the literature and many modeling and empirical papers focus on this trade-off. We now mention this explicitly in the main text and appendix.

“Of course, the stability-flexibility tradeoff remains, and the level of annealing could be further optimized dynamically, as shown specifically within OpAL (Franklin and Frank, 2015) and more broadly in the literature (Nassar et al., 2012; Iglesias et al., 2013). This stability-flexibility trade-off is particularly apparent in drifting reward environments (Appendix; Figure A3d), but optimization for this tradeoff is largely orthogonal to the focus of the present work.…”

Also see the full portion of the following section: “Notably, annealing intensifies a common trade-off of increased stability for decreased flexibility. While lowering the learning rate protects the actor weights from converging to zero and allows the model to retain a useful policy, the actor weights become insensitive to changes in reward contingencies when the variance of the meta-critic is sufficiently small. In this paper, our scope focus on stationary reward environments (where rewards do not change) and this flexibility is not required. However, in environments where reward consistently drifts, OpAL* suffers from gradual decay when annealing is lowered to track reward fluctuations (Figure A3d), though such decay is improved relative to the OpAL model without annealing (not shown)…”

Nevertheless, to address the point regarding annealing, note that the learning curves presented in Appendix 1—figure 4 show stabilization of OpAL* weights and corresponding Act curves for 1000 trials, as originally requested by the reviewer. While there is mild decay in the first 200 trials for some reward contingencies, the weights and policy stabilize and the ordinal rankings (which determine the policy) are robust. With the same parameter settings, we can extend this to 2000 trials and observe similar dynamics (G/N/Act) (not shown).

Without Bayesian annealing, in Author response image 9 we observe the dynamics which starkly show the detriment relative to annealing:

As we now mention in the revised paper, this annealing could be further optimized to be sensitive to volatility in the environment, as other models have done (including within OpAL). Nonetheless, Bayesian annealing used here does allow for some flexibility, as evidenced by the simple change-point scenario in Appendix 1—figure 6, left. Appendix 1—figure 6, left demonstrated that the model as formulated in the paper and used for all simulations (i.e. without decay) can indeed adapt to an environmental switch point. This is achievable even with more extensive pre-switch learning (Author response image 10) :

**Author response image 9. sa2fig9:** 

**Author response image 10. sa2fig10:** 

Regarding weight decay in Appendix 1—figure 5 right and Appendix 1—figure 6 right, this was meant to demonstrate that OpAL*’s adaptability to change points, which was raised in the first review, could be further enhanced with biologically plausible weight decay mechanisms that have been explored in Franklin and Frank within OpAL* but which are not the focus of this paper. While we don’t include weight decay in the main simulations for this paper, we merely chose to omit this mechanism to minimize the number of parameters explored and for comparison to other models. Adding such decay should not impede the advantages described in the paper in stationary environments since these advantages predominate during early learning; Indeed, we have now verified that if we simply add the decay to prior used above to the simulations that optimize learning curves in stationary environments, the advantages remained. Nevertheless, future work could assess whether OpAL* advantages could be further improved when optimizing parameters jointly with weight decay over long time horizons.

Finally, the reviewer notes that “it is still evident that on average the weights decay with trials”. As shown in (Author response image 11) this decay is minimal up to 2000 trials, although we agree that there are parameter regimes in which this may be more apparent (for example, lower actor learning rates are related to less decay), but as we also note above this can be mitigated in principle, and as we noted in the manuscript, the decay specifically occurs when the critic converges (at which point the Q value could be in principle used for action selection, but this transition is left for future work).

**Author response image 11. sa2fig11:** 

Perhaps the reviewer was referring to the drifting reward environment, in which case annealing helps but does not completely prevent decay (though the G/N ordering for OpAL* can still reasonably track the value of an action and reach a value of approximately.6). Author response image 11 is without annealing, Appendix 1—figure 5 is with annealing.

This simulation was added given the reviewer’s earlier comments, to show that this is not a fundamental limitation of OpAL* and that it can adapt to these volatile scenarios, and that again this can be further enhanced with decay to prior (Appendix 1—figure 5). But as noted, we feel that these volatile drifting-reward environments are beyond the main scope of this paper, which focuses on performance advantages in novel stationary environments by optimizing action gaps, and hence we consider these simulations as supplemental rather than central.

Nonetheless, we have adjusted the text to emphasize that additional mechanisms and considerations (e.g. decay, optimizing T) are required to optimize adaptability for drifting reward environments in the section "Normalization and annealing”.

Specific comments:Equation 15 – How is std X estimated? Is it computed from the analytic expression for β distribution? If so, this expression is complex (involving division, square, etc.), so please comment on how such computation could be made by biological networks of neurons.

Yes, this is calculated by taking the analytic expression for the β distribution, but as noted in response to point 1, several computational papers suggest summary statistics may be represented in biological neural networks (e.g., Ma et al. 2006 and 2008, Pecevski and Maass 2016, Deneve 2008, Deneve et al. 1999), and that only a rough proxy for this is needed for our purposes in any case. There is also evidence that orbitofrontal regions represent confidence and task states, and modulate striatal dopamine in proportion to reward history, via projections to cholinergic neurons (Kepecs et al., 2008, Stalnaker et al., 2016; Adrover et al. 2020; Mohebi et al. 2019), as cited in the manuscript. Our focus here is not on the mechanisms of such uncertainty computations but their downstream impacts on the striatal machinery for learning and choice, and moreover that the benefits of doing so require opponency and the hebbian nonlinearity according to OpAL*.

Equation 22 – Please explain parameter T. As above, please explain how X is computed in the simulation.

Thank you for asking; revisiting this section, there was not enough detail to make this clear. var(X) is the analytical variance of the β distribution of the meta-critic (X is defined in Equation 12). T is a hyper-parameter that determines how variance is converted to annealing. We have updated the text to clarify these definitions.

Line 293: "These modifications improve the robustness of OpAL* and ensure that the actor weights are well-behaved" – I do not agree with this statement, because in the OpAL* model the weights still converge to 0 unless they are prevented from convergence to 0 by making the model non-adaptive. Please replace "well behaved" in the cited statement, by a more specific description.

We have replaced “well behaved” with a more specific description:

“These modifications improve robustness of OpAL* and ensure that the actor weights are better behaved (avoiding convergence to zero and maintaining ordinal rankings in the resulting Act values for 1000 trials; Figure A3b)…”

Line 499: "Indeed, algorithms like Q-learning and UCB converge well when an option is well-sampled, but the speed and accuracy of this convergence is affected by stochastic sampling". This argument does not apply to UCB, which is deterministic, and hence the sampling in UCB is NOT stochastic.Line 582 – this again does not seem to apply to UCB, which is deterministic.

For both the above points, we have now clarified that the key issue is not whether the choice on a given trial is stochastic, but rather that the speed and accuracy of convergence for UCB is impeded with the requirement to choose among multiple actions (and switching between then across trials), yielding partial information on reward statistics across all options. Thus even though UCB chooses deterministically on a given trial, it still only chooses a single option on that trial and eventually will shift to other options and learn about those, depending on the value and the number of times it has selected those options, and the C parameter, as noted above. We now show explicitly (Supplementary Figure S4) that UCB suffers convergence when choosing between multiple actions.

Figure 6a is very interesting, but I have a few suggestions to make it more informative. I simulated UCB on the problems in Figure 6a and verified that it indeed gives similar performance to that visualized in this figure, but found that with 1000 iterations there is still substantial variability in the results, and sometimes you get non-monotonic changes in accuracy as shown in blue curve in Figure 6a left, which disappear if the number of iterations is increased. Hence, I suggest to increase the number of iterations (repetitions) in Figure 6a to 10,000 to get less noisy curves.

In response to a later comment, we have modified this figure to use the updated UCB algorithm (which explicitly samples each action once before exploring) and to reflect the optimal parameters for each environment. The learning curves presented are those that generated that data point on the rightmost tail on the AUC histograms for each agent. Because those histograms were based on 1000 simulations, we find it more parsimonious to show the 1000 simulations that went into these histograms, but we present the 10,000 simulations here to confirm comparable dynamics (Author response image 12) Results for extended time horizons again were comparable (Author response image 13).

**Author response image 12. sa2fig12:** 

**Author response image 13. sa2fig13:** 

In Figure 6a the gap between UCB and OpAL* is higher in the right panel but it is not clear if this is due to change in richness or in the number of options. It would be helpful to add two more panels ([0.2 0.3] and [0.8 0.7 0.7 0.7 0.7 0.7]). Also for an easier replicability, it would be good to list the parameters used in simulations of each model.

We have updated the figure so that the learning curves directly illustrate the data that underlies the AUC in the histograms for best performing parameters in each agent, and (as the reviewer suggested elsewhere) to use the modified UCB algorithm (i.e. sampling each option once on the first trial before exploring, giving it a further advantage). We keep just two illustrative examples of these learning curves in the figure because it is already multi-panel and they merely illustrate data that are already in the histograms. Nevertheless, we have now added the corresponding optimal learning curves for the other cases that the reviewer mentions, as well as other time horizons (1000 trials), in the supplementary figures. Code will be made available for the replicability of these learning curves.

l. 1478: "Normalizing, therefore, addresses the valid concerns of Möller and Bogacz (2019) while still preserving core OpAL dynamics". I do not agree – the key concern of Möller and Bogacz (2019) was that the weights decay to 0 eventually. The normalization does not fix the problem. In the simulations I did in the previous round of the review, rewards were 0 and 1, so the normalization had no effect.

We agree that normalization on its own does not prevent the decay to zero due to the critic convergence as analytically shown in Moller and Bogacz (2019). However, in the provided scenario, the decay is exacerbated by large reward magnitudes. In Author response image 14, the left figure is a replication of the figure from Moller and Bogacz (2019); the right shows the effect of normalization only (no annealing). Without annealing the weights WILL eventually decay as helpfully outlined by Moller and Bogacz; however, the rapid decay to zero in approximately 20-30 trials resulted in part from large RPE magnitudes (in the range of -3 to 3).

**Author response image 14. sa2fig14:** 

We have adjusted our wording in this section to better emphasize that the normalization serves to address issues caused by the magnitude of RPE. This is distinctive from the decay effects caused by the oscillations in the critic prediction error, for which we introduced annealing.

“Normalizing, therefore, addresses one factor (large RPE magnitudes) contributing to the rapid decay in early trials demonstrated by Möller and Bogacz (2019)…”

Equation 30 – I guess the last 2 "cases" were included accidentally (they are a typo), and should be removed.

Thank you, this has been adjusted.

l.1112: UCB – Please provide the reference for the paper describing UCB. In this algorithm, the model is initialized to selecting each action once before relying on estimated values. Do you do such initial selection? If so, please say this.

We had not originally included this initialization step, in part because this provides UCB with another advantage, in that it effectively gets full information for one trial, and our main intent was to demonstrate that partial information was more detrimental to value-based algorithms like Q learning and UCB than to OpAL*. However, upon reflection, we decided to implement this initialization step to better reflect how UCB is actually used. We have rerun our simulations with such initialization and found comparable results, albeit with some slight improvement to UCB performance given this additional information about each option. We have updated our figures and methods accordingly. References to Sutton and Barto and Auer et al. (2002) have been added.

Throughout – In many places in the manuscript there are references to appendix, but the appendix is long, and it takes time to find information. Hence whenever you point the reader to the appendix, please point to a specific section or figure.

We have now included links to the appropriate sections of the appendix and supplementary figures in the main text.